# An improved epigenetic counter to track mitotic age in normal and precancerous tissues

Tianyu Zhu[1,3], Huige Tong[1,3], Zhaozhen Du[1], Stephan Beck[2] & Andrew E. Teschendorff[1]✉

The cumulative number of stem cell divisions in a tissue, known as mitotic age, is thought to be a major determinant of cancer-risk. Somatic mutational and DNA methylation (DNAm) clocks are promising tools to molecularly track mitotic age, yet their relationship is underexplored and their potential for cancer risk prediction in normal tissues remains to be demonstrated. Here we build and validate an improved pan-tissue DNAm counter of total mitotic age called stemTOC. We demonstrate that stemTOC's mitotic age proxy increases with the tumor cell-of-origin fraction in each of 15 cancer-types, in precancerous lesions, and in normal tissues exposed to major cancer risk factors. Extensive benchmarking against 6 other mitotic counters shows that stemTOC compares favorably, specially in the preinvasive and normal-tissue contexts. By cross-correlating stemTOC to two clock-like somatic mutational signatures, we confirm the mitotic-like nature of only one of these. Our data points towards DNAm as a promising molecular substrate for detecting mitotic-age increases in normal tissues and precancerous lesions, and hence for developing cancer-risk prediction strategies.

A key priority area of precision and preventive oncology is to develop novel strategies for early detection and risk prediction[1,2]. Given the growing evidence that the risk of neoplastic transformation of any given tissue in any given individual is strongly influenced by the cumulative number of stem-cell divisions (aka mitotic age) that the underlying cell-of-origin has undergone[2–5], the ability to measure this mitotic age in human tissues could help address this clinical need. Underpinning the association of mitotic age with cancer risk is the gradual accumulation of molecular alterations following stem-cell division, eventually predisposing specific subclones to neoplastic transformation[6–12]. Conversely, these molecular alterations, if measured, could be used to estimate mitotic age[11,13–22].

Among the potential molecular alterations that could be used to measure mitotic age, somatic mutations and DNA methylation (DNAm)

have emerged as the most promising ones[11,14,15,23]. DNAm-based mitotic-like counters hypothesized to track the cumulative number of DNA methylation errors arising during cell division in both stem-cell and expanding progenitor cell populations have been proposed[16–18,24–29], yet confounders such as cell-type heterogeneity (CTH)[30] and chronological age[24] have cast doubt on the biological and clinical significance of these counters and their total mitotic age estimates (defined here as the cumulative number of divisions in both stem-cells and amplifying progenitor populations). Moreover, DNAm changes in normal and preneoplastic lesions often display a stochastic pattern[19,31,32], which can pose specific statistical challenges to estimating mitotic age. Finally, although the relation between clock-like DNAm and somatic mutational signatures has already undergone preliminary investigations[18,33], these studies have only explored this in the context

[1]CAS Key Laboratory of Computational Biology, CAS-MPG Partner Institute for Computational Biology, Shanghai Institute of Nutrition and Health, Shanghai Institute for Biological Sciences, University of Chinese Academy of Sciences, Chinese Academy of Sciences, 320 Yue Yang Road, Shanghai 200031, China. [2]Medical Genomics Group, UCL Cancer Institute, University College London, 72 Huntley Street, WC1E 6BT London, UK. [3]These authors contributed equally: Tianyu Zhu, Huige Tong. ✉e-mail: andrew@sinh.ac.cn

of B cell malignancies[33] or did not directly link total mitotic age estimates to the specific somatic mutational signatures from Alexandrov et al.[18,34]. Exploring this relationship across multiple cancer types is vital to better understand which molecular substrate may be more suitable for tracking mitotic age on the time scales in which precancerous lesions progress to invasive cancer[35,36].

Here we build and validate a pan-tissue epigenetic counter of total mitotic age called stemTOC (Stochastic Epigenetic Mitotic Timer of Cancer) that addresses the above challenges, and subsequently apply it in conjunction with a DNAm atlas and advanced cell-type deconvolution methods[37,38], to demonstrate that a sample's total mitotic age correlates with the fraction of its putative tumor cell-of-origin, thus establishing a direct link between mitotic age and tumor progression. We show that stemTOC can track the subtle increases in mitotic age of normal tissues exposed to cancer-risk factors, or at risk of cancer development. By cross-correlating stemTOC to clock-like mutational signatures, we find that the number of C > T mutations representing deamination of 5-methylcytosine is a marker of mitotic age. Collectively, our data point towards a potential advantage of DNAm changes over clock-like mutational signatures[11,34] as a means of tracking total mitotic age in normal tissues at cancer risk, whilst also adding substantial evidence to the hypothesis put forward by Tomasetti and Vogelstein[3], that the mitotic age of a tissue is a major cancer-risk factor.

## Results

### Construction of stemTOC
We aimed to build a mitotic counter that is, to the largest extent possible, unaffected by confounders such as CTH and chronological age, and which can also account for the potential stochasticity of DNAm changes. To reduce confounding by CTH it is critical to build a mitotic counter with CpGs that are not cell-type specific in an appropriate ground state[39]. To this end, we collected Illumina 450k/EPIC DNAm profiles from 86 fetal samples encompassing 13 fetal/neonatal tissue-types ("Methods" section), to select 30,257 promoter-associated CpGs that are constitutively unmethylated across all fetal/neonatal tissue-types (Fig. 1a). To avoid confounding by chronological age, we used the cell-line data from Endicott et al.[24] to require that CpGs, which undergo significant DNA hypermethylation (i.e. displaying gains of DNAm) with increased population doublings (PDs) in-vitro across a range of different normal cell lines, that they simultaneously do not undergo hypermethylation in these same cell lines when treated with a cell-cycle inhibitor (e.g mitomycin) or under reduced growth-promoting conditions (i.e. serum deprivation) ("Methods" section). A total of 6 cell lines were considered representing a variety of cell types including fibroblasts, smooth muscle, and endothelial cells[24]. Of the 30,257 CpGs, only 629 (denoted "vitro-mitCpGs", Fig. 1a) satisfied these requirements. In order to avoid confounding by cell-culture effects, we next required these CpGs to also undergo significant DNA hypermethylation with chronological age in three separate large in-vivo blood DNAm datasets (Fig. 1b). Since the cell-line data removes many CpGs that accumulate DNA hypermethylation purely because of "passage of time" (and hence chronological age), this additional requirement is designed to ascertain that these vitro-mitCpGs do display age-associated DNA hypermethylation in-vivo, in line with the expectation that in a tissue with a relatively high stem-cell division rate (i.e. blood), mitotic age and chronological age should be strongly correlated. Blood-tissue was chosen for another 2 reasons. First, the availability of many large whole-blood DNAm datasets ensures adequate power to detect DNAm changes associated with chronological and mitotic age. Second, for blood-tissue, we can adjust for CTH at high resolution (12 immune cell subtypes)[40,41], which further ensures that the observed DNAm changes are not due to shifts in underlying cell-type proportions[31]. At the end of this step, we thus obtained a reduced subset of 371 mitotic CpG candidates, which we call "vivo-mitCpGs" defining our stemTOC CpGs (Fig. 1b, Supplementary Data 1).

Next, we subjected these 371 stemTOC CpGs to further tests to verify their validity. First, using independent DNAm data from neonatal buccal swabs and cord blood we verified that our 371 stemTOC CpGs retain ultra-low DNAm levels, as required ("Methods" section, Supplementary Fig. 1). Second, we used 3 separate age-matched DNAm datasets of sorted cells[42–44] (including adult neutrophils, monocytes, naïve CD4+ T-cells, B cells, neurons, adipocytes, endothelial cells, lung-epithelial, colon-epithelial, hepatocytes, exocrine and endocrine pancreas), as well as the age-matched multi-tissue eGTEX dataset[45], to confirm that stemTOC CpGs are not cell-type specific markers of adult cell types ("Methods" section, Supplementary Figs. 2 and 3). Indeed, any cell-type specific DNAm differences in these aged cell and tissue-types were exclusively restricted to comparisons involving colon-epithelial cells, the tissue with the highest turnover rate, thus clearly indicating that these DNAm differences are likely due to differences in cell-type specific mitotic rates (Supplementary Figs. 2 and 3). Applying eFORGE2[46,47] to the 371 stemTOC CpGs, we observed strong and exclusive enrichment for DNase Hypersensitive Sites (DHSs) as defined in hESCs (Supplementary Fig. 4a). Among chromatin states, we observed strong enrichment for bivalent transcription start sites (TSS), which was strongest for those also defined in hESCs (Supplementary Fig. 4b). Correspondingly, we observed strong enrichment for both H3K27me3 and H3K4me3 marks, although notably stronger for the repressive H3K27me3 mark (Supplementary Fig. 4a), consistent with previous observations that H3K4me3 is moderately protective of age-associated DNAm accrual[39,48].

Based on the 371 vivo-mitCpGs, we finally derive a relative estimate of total mitotic age that explicitly accounts for the underlying stochasticity of mitotic DNAm changes in normal tissues[19,31,32] ("Methods" section, Fig. 1c). This stochasticity implies that the 371 vivo-mitCpGs could display marked variations in DNAm within a tissue and subject, owing to the presence of multiple subclones of varying carcinogenic potential. We reasoned that a specific CpG subset displaying the highest DNAm levels could mark the clone at the highest risk and that this subset may vary randomly between subjects[19,32,49]. We built a simple yet realistic simulation model ("Methods" section) to demonstrate how taking a specific upper quantile of the DNAm distribution over mitotic CpGs should yield an improved estimator of total mitotic age compared to taking an average DNAm over these same CpGs (Fig. 1c)[19,32]. In effect, taking an upper quantile can better capture the mitotic age of any dominant subclone within the complex subclonal mosaic characteristic of any aging tissue[19,32,50–56]. To identify a sensible upper-quantile threshold, we computed upper quantiles for the 371 vivo-mitCpGs in a real DNAm dataset encompassing 42 normal-breast samples adjacent to breast cancer ("normal-tissue at risk") and 50 age-matched normal-breast samples from healthy women ("normal healthy")[32] ("Methods" section, Fig. 1d). This revealed substantially larger effect sizes for higher upper-quantile values consistent with DNAm changes in the normal tissue at risk being highly stochastic. We decided on a 95% upper quantile threshold to maximize effect size without compromising sampling variability (Fig. 1d).

### Validation of stemTOC in normal tissues and sorted cell populations
One way to assess if stemTOC is measuring mitotic age in in-vivo human tissue samples is by cross-comparing correlation strengths of mitotic age estimates with chronological age across tissues characterized by very different stem-cell division rates. We reasoned that for tissues with a high stem-cell division rate, the total mitotic age will be strongly determined by the subject's chronological age, whilst for tissues with a low stem-cell division rate, or for tissues where the mitotic age is more strongly influenced by temporary turnover or other factors operating on shorter time scales (e.g. hormonal factors), the correlation between mitotic age and chronological age should be much weaker or non-evident. Applying stemTOC to the

# Construction of the stochastic epigenetic mitotic timer of cancer (stemTOC)

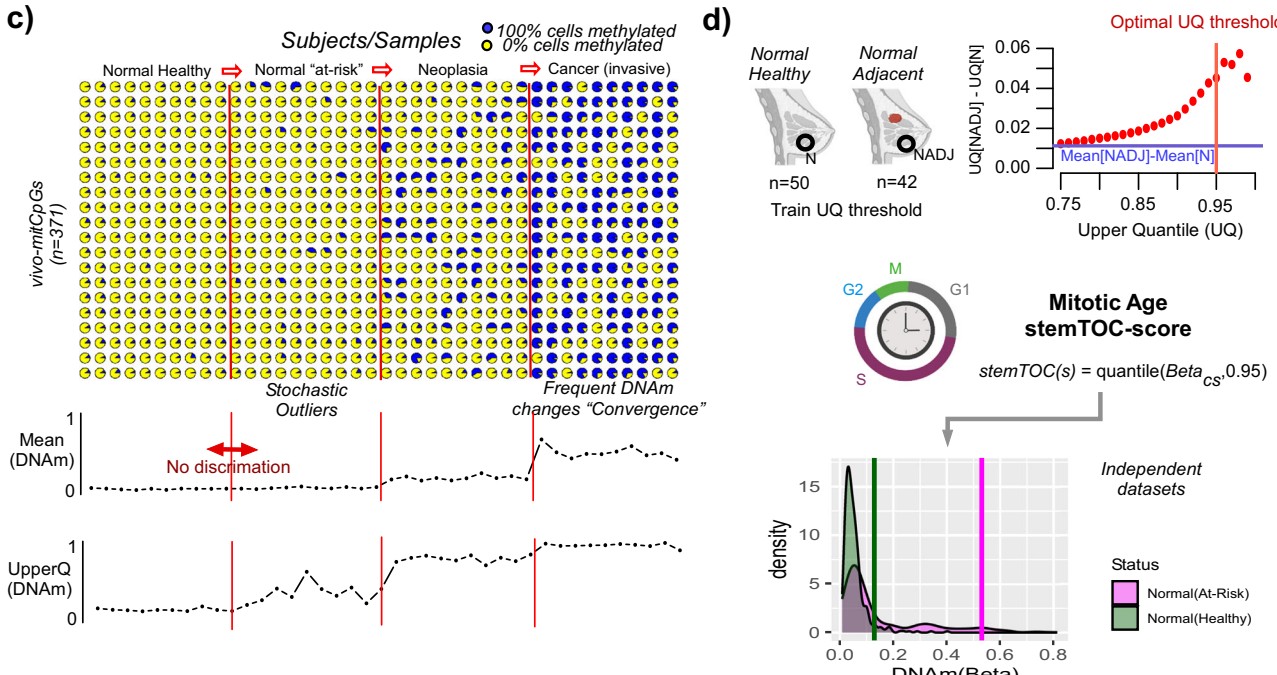

**Fig. 1 | Construction of stemTOC and estimation of mitotic age. a** We first identify CpGs ($n = 30,257$) mapping to within 200 bp upstream of the TSS of genes that are unmethylated (defined by DNAm beta value < 0.2) across 86 fetal-tissue samples from 13 different fetal-tissues (including neonatal cord blood). These are then filtered further by the requirements that they display hypermethylation as a function of population doublings (PDs) in 6 cell lines representing fibroblasts, endothelial, smooth muscle, and epithelial cell types. To avoid confounding by chronological age, we also demand that they don't display such hypermethylation when cell lines are deprived of growth-promoting serum or when treated with mitomycin (MMC, a cell-cycle inhibitor), resulting in 629 "vitro-mitCpGs". **b** To exclude cell-culture effects, CpGs displaying significant hypermethylation with chronological age, as assessed in 3 separate whole-blood cohorts and after adjusting for variations in 12 immune-cell type fractions, are selected. In the heat-maps, rows label samples, ordered by increasing age. Columns label CpGs ordered according to hierarchical clustering. **c** Simulation of DNAm changes at 20 CpGs during carcinogenic transformation (a total of 40 independent samples, 10 from

each disease stage). Initially, DNAm changes are inherently stochastic, and average DNAm over the CpGs may not discriminate normal healthy from normal "at-risk" tissue. Taking an upper 95% quantile of the CpG's DNAm values can discriminate normal from normal at-risk. **d** Top: Scatterplot depicts the difference in upper quantiles (y-axis) over the 371 vivo-mitCpGs between 42 normal samples adjacent to breast cancer (NADJ, "normal at-risk") and 50 age-matched healthy samples (N, "normal healthy"), against the upper quantile threshold (x-axis). Horizontal blue line indicates the difference in mean DNAm over the 371 CpGs between NADJ and N tissue. Vertical red line marks the upper-quantile threshold maximizing difference without compromising variability. A relative mitotic-age score (stemTOC) is obtained for any sample, by taking the 95% upper quantile (UQ) over the 371 DNAm beta-values corresponding to these vivo-mitCpGs. Bottom density plots display the distribution of the 371 DNAm values for one hypothetical normal healthy and one hypothetical normal at-risk sample. Vertical lines indicate the stemTOC-scores (95% upper quantiles defined over the 371 CpGs). Generated with Biorender.com.

normal-adjacent tissue data from the TCGA confirmed this: tissues with a high stem-cell division rate like the colon and rectum displayed strong correlations between mitotic and chronological age, whilst slower-dividing and hormone-sensitive tissues like breast and endometrium did not, despite these being appropriately powered (Fig. 2a, b, Supplementary Fig. 5, Supplementary Data 2). Similar results

were obtained when estimating mitotic age in the enhanced GTEX (eGTEX) DNAm dataset, encompassing 987 samples from 9 normal tissue-types (Supplementary Fig. 6, Supplementary Data 3). For instance, the colon displayed a clear correlation, whilst the ovary and breast did not (Supplementary Fig. 6). Applying stemTOC to large purified immune cell datasets from healthy individuals

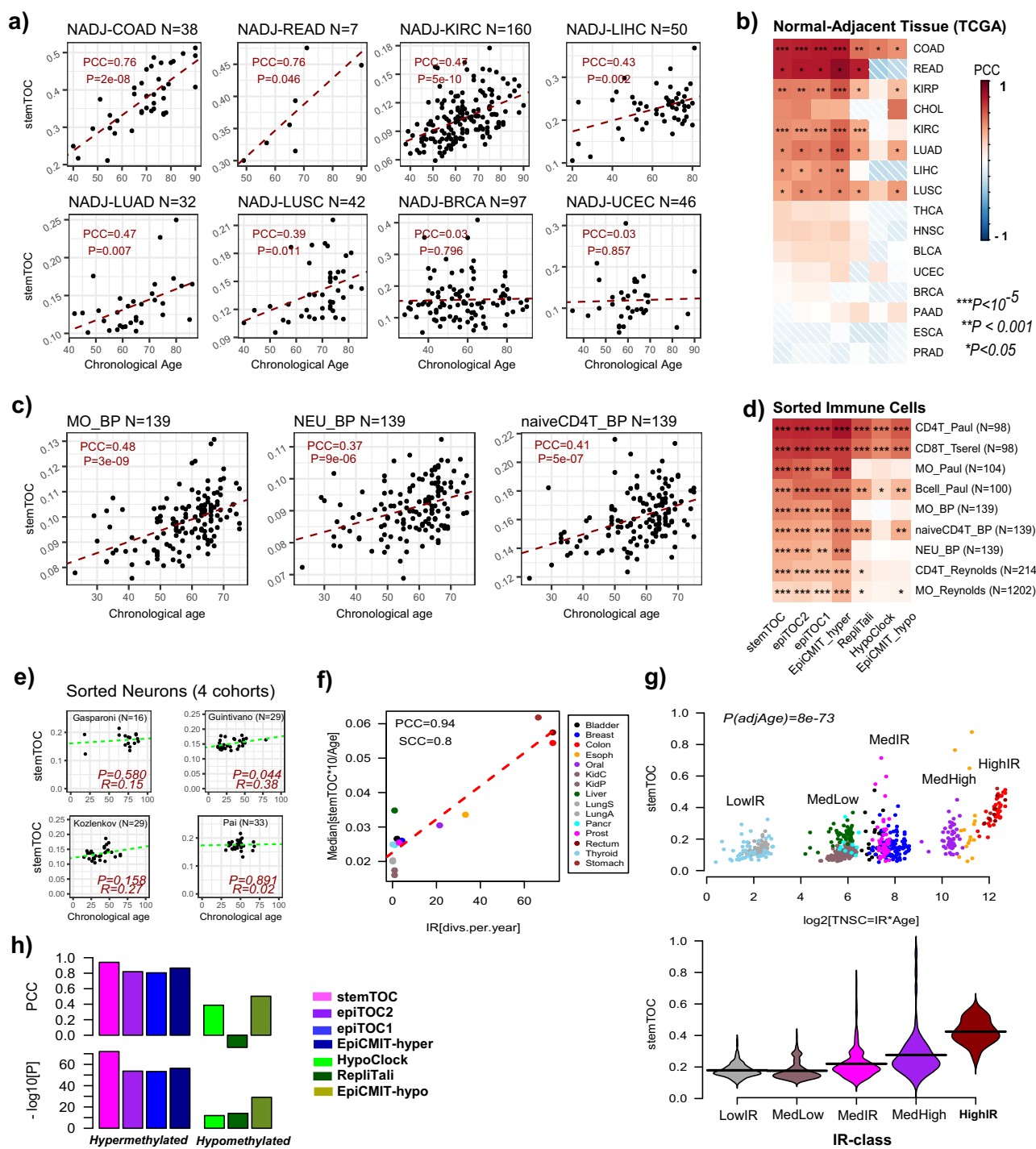

("Methods" section) also revealed relatively strong correlations with chronological age (Fig. 2c, d, Supplementary Data 4), in line with the hematopoietic system's relatively high stem-cell division rate[3,57]. Of note, the correlation of stemTOC with chronological age was also observed in a fairly large dataset of 139 naïve CD4+ T-cell samples (Fig. 2c, d), supporting the view that the observed correlation in sorted CD4+ T-cell datasets is not the result of age-associated shifts in the naïve to mature CD4+ T-cell fractions[41]. Finally, in sorted neurons, a predominantly post-mitotic population, the correlation was generally speaking not significant (Fig. 2e).

To further demonstrate that the associations of stemTOC with chronological age are not confounded by cell-type heterogeneity, we assembled a large collection of 18 whole-blood cohorts encompassing

14,515 samples ("Methods" section), computing associations of stem-TOC with chronological age in each one of them, before and after adjustment for immune-cell fractions using our recently validated DNAm reference matrix for 12 immune-cell types[41]. Associations of mitotic age with chronological age remained significant and even displayed a marginal increase after adjustment for CTH (Supplementary Fig. 7). Similar findings were obtained in the eGTEX normal tissue datasets for which we could infer cell-type fractions using our EpiSCORE DNA-atlas[38] ("Methods" section, Supplementary Fig. 8a, b). Thus, the correlation of stemTOC with chronological age is clearly independent of any putative age-associated variations in cell-type fractions as well as of any differences in the mitotic ages of underlying cell types.

**Fig. 2 | Validation and benchmarking of stemTOC. a** For selected normal-adjacent tissue-types, scatterplots of stemTOC's mitotic age (y-axis) vs chronological age (x-axis). Number of normal-adjacent samples is given above each plot. The Pearson Correlation Coefficient (PCC) and two-tailed *P*-value from a linear regression are given. Normal-adjacent tissue-types in TCGA are labeled by the corresponding cancer type. **b** Heatmap of PCC values between mitotic and chronological age for 7 mitotic clocks and across all normal-adjacent tissue-types from the TCGA. Two-tailed *P*-values are from the linear regression test. Number of normal-adjacent samples for tissue-types not shown in (**a**) are: CHOL(*n* = 9), KIRP(*n* = 45), THCA(*n* = 56), HNSC(*n* = 50), BLCA(*n* = 21), PAAD(*n* = 10), PRAD(*n* = 50), ESCA(*n* = 16). **c** As (**a**), but for three sorted immune-cell populations as profiled by BLUEPRINT. **d** As (**b**) but for all sorted immune-cell populations. Sorted immune-cell samples labeled by cell-type and study it derives from. BP = BLUEPRINT. **e** As (**a**) but for sorted neurons from 4 different cohorts. In each panel we give the *R*-value (same as PCC) and corresponding correlation test P-value. **f** Scatterplot of a normal tissue's relative intrinsic rate (RIR) estimated from taking the median of stemTOC's mitotic age divided by chronological age and multiplied by 10 (y-axis), against the Vogelstein–Tomasetti intrinsic annual rate of stem-cell division (IR) (x-axis), using the normal-adjacent tissues with such estimates. The Pearson and Spearman Correlation Coefficients (PCC & SCC) are given. **g** Top: Scatterplot of a normal sample's mitotic age (estimated with stemTOC) vs the estimated total number of stem-cell divisions (TNSC = IR * Age) expressed in a log2-basis, to better highlight the differences between the low and medium turnover tissues. Two-tailed *P*-value is from a multivariate linear regression including age as a covariate. Bottom: Violin plot representation of middle panel, with normal-adjacent tissue samples grouped into 5 IR-classes, as shown. Low: Thyroid + Lung, MedLow = Liver + Pancreas + Kidney, Med = Prostate + Breast + Bladder, MedHigh = Esophagus + Oral, High = Colon + Rectum + Stomach. **h** Barplots compare the PCC and significance level attained by stemTOC shown in (**g**) to the corresponding values for 6 other mitotic clocks. Mitotic clocks based on CpGs that gain/lose methylation with cell division (hypermethylated/hypomethylated) are indicated. RepliTali results are for the probes restricted to 450k beadarrays.

To further validate stemTOC, we compared the estimated mitotic ages of tissues stratified by high, medium, and low stem-cell division rates[3,16]. Using the normal-adjacent samples from the TCGA encompassing 15 tissue-types, we observed a strong correlation (PCC = 0.94) between stemTOC's median mitotic age of the tissue and the tissue-specific annual intrinsic rate of stem-cell division, as estimated by Tomasetti & Vogelstein[3,8] (Fig. 2f, "Methods" section). This association was confirmed by plotting stemTOC against the estimated total number of stem-cell divisions for each normal-adjacent sample on a log-scale, revealing clear differences in mitotic age between low, medium, and high stem-cell division rate tissues (Fig. 2g).

## Benchmarking of stemTOC in normal tissues reveals improvements

We next benchmarked stemTOC against 6 previously proposed DNAm-based mitotic counters, five of which are built from, or restrict to, 450k array probes (epiTOC2[30], epiTOC1[16], HypoClock[18,30] and EpiCMIT-hyper/hypo[33]), with the remaining one (RepliTali[24]) being built entirely from EPIC beadarrays. CpGs making up each clock displayed relatively little overlap except for the epiTOC1 and epiTOC2 clocks (Supplementary Fig. 9). We observed that counters based on hypermethylated CpGs (i.e. stemTOC, epiTOC1/2 and epiCMIT-hyper) displayed stronger correlations with age than counters based on hypomethylated ones (i.e. displaying loss of DNAm) (HypoClock, RepliTali, epiCMIT-hypo) (Fig. 2b). Similar patterns were observed in the normal-tissue EPIC DNAm data from eGTEX (Supplementary Fig. 6) as well as in the sorted immune-cell subsets (Fig. 2d). StemTOC and the hypermethylation-based counters also displayed much stronger correlations with the Vogelstein–Tomasetti stem-cell division rates, as assessed over 15 tissue-types (Fig. 2h), suggesting that these counters yield more accurate mitotic age estimates during normal/healthy aging compared to those based on hypomethylation.

To explore the effect of CTH and the additional caveat that RepliTali is EPIC-based and thus less likely to perform well on 450k datasets, we compared regression statistics with chronological age before and after adjustment for CTH, and, in the case of whole blood, restricting also only to EPIC datasets. We observed that hypermethylated counters displayed less sensitivity to the underlying CTH of the tissue, as compared to the hypomethylated ones (Supplementary Figs. 7, 8, 10). It is also noteworthy that stemTOC's mitotic age displayed stronger correlations with chronological age than those of all hypomethylated clocks, including RepliTali (Supplementary Figs. 8b, S10b). For instance, in a high-turnover-rate tissue (colon eGTEX EPIC dataset), RepliTali's association with chronological age was only significant upon adjustment for CTH, whilst stemTOC's association was stronger and independent of CTH adjustment (Supplementary Fig. 8b). Moreover, the effect of beadarray technology seemed to be

relatively minor as RepliTali's mitotic age estimates on EPIC data were very robust upon restricting to 450k probes (Supplementary Fig. 8c). Thus, overall, the results in normal tissue suggest an improvement of stemTOC and the other hypermethylated clocks over the hypomethylated based ones.

## stemTOC's mitotic age correlates with tumor cell-of-origin fraction

We reasoned that tumor samples of higher tumor purity would display a higher mitotic age if mitotic age is a key marker of cancer progression. One way to estimate tumor purity is by estimating the tumor cell-of-origin fraction in cancer samples, which could be accomplished with our EpiSCORE algorithm and DNAm atlas encompassing tissue-specific DNAm reference matrices for 13 tissue and 40 cell types ("Methods" section)[38]. Before estimating cell-type fractions in the TCGA samples however, we sought additional validation of our DNAm atlas using an independent whole-genome bisulfite-sequencing (WGBS) DNAm atlas of 102 sorted samples from Loyfer et al.[58] encompassing 15 tissue-subtypes and 17 cell types ("Methods" section). Applying EpiSCORE we were able to correctly predict the cell type in Loyfer's WGBS DNAm atlas with an overall 85% accuracy (Fig. 3a). Given this good validation, we next applied stemTOC and EpiSCORE to the TCGA samples (Supplementary Data 5–21), which revealed strong correlations between mitotic age and tumor cell-of-origin fraction, especially for those tumor-types where the cell-of-origin is reasonably well established (Fig. 3b). For instance, colon and rectal adenocarcinoma (COAD & READ) displayed very strong correlations between stemTOC's mitotic age and the epithelial cell fraction (Fig. 3b). Mitotic age in luminal and basal breast cancer displayed strongest correlations with the luminal and basal fractions, respectively. In the case of pancreatic adenocarcinoma (PAAD), mitotic age was most strongly correlated with the ductal fraction. Liver hepatocellular- (LIHC) and cholangio- (CHOL) carcinoma displayed the strongest correlations with hepatocyte and cholangiocyte fractions, as required. Mitotic age in skin cutaneous melanoma (SKCM) displayed the strongest correlations with the melanocyte fraction. Lung squamous cell (LUSC) and lung-adeno (LUAD) carcinoma displayed the strongest correlations with basal and alveolar epithelial fractions, respectively. It is worth stressing that for each tumor-type, the strongest correlation with mitotic age was attained by the presumed tumor cell-of-origin (Supplementary Fig. 11a, b). stemTOC's mitotic age also displayed strong correlations with other tumor purity indices as estimated by Aran et al.[59], although the correlations with EpiSCORE's tumor cell fraction were generally higher than for gene-expression or IHC-based tumor purity estimation methods (Supplementary Fig. 11c). These data clearly indicate that despite the CTH of TCGA samples, the mitotic age estimates from stemTOC are tracking DNAm changes in the tumor cell-of-origin of the respective cancer type. Mitotic age estimates can also

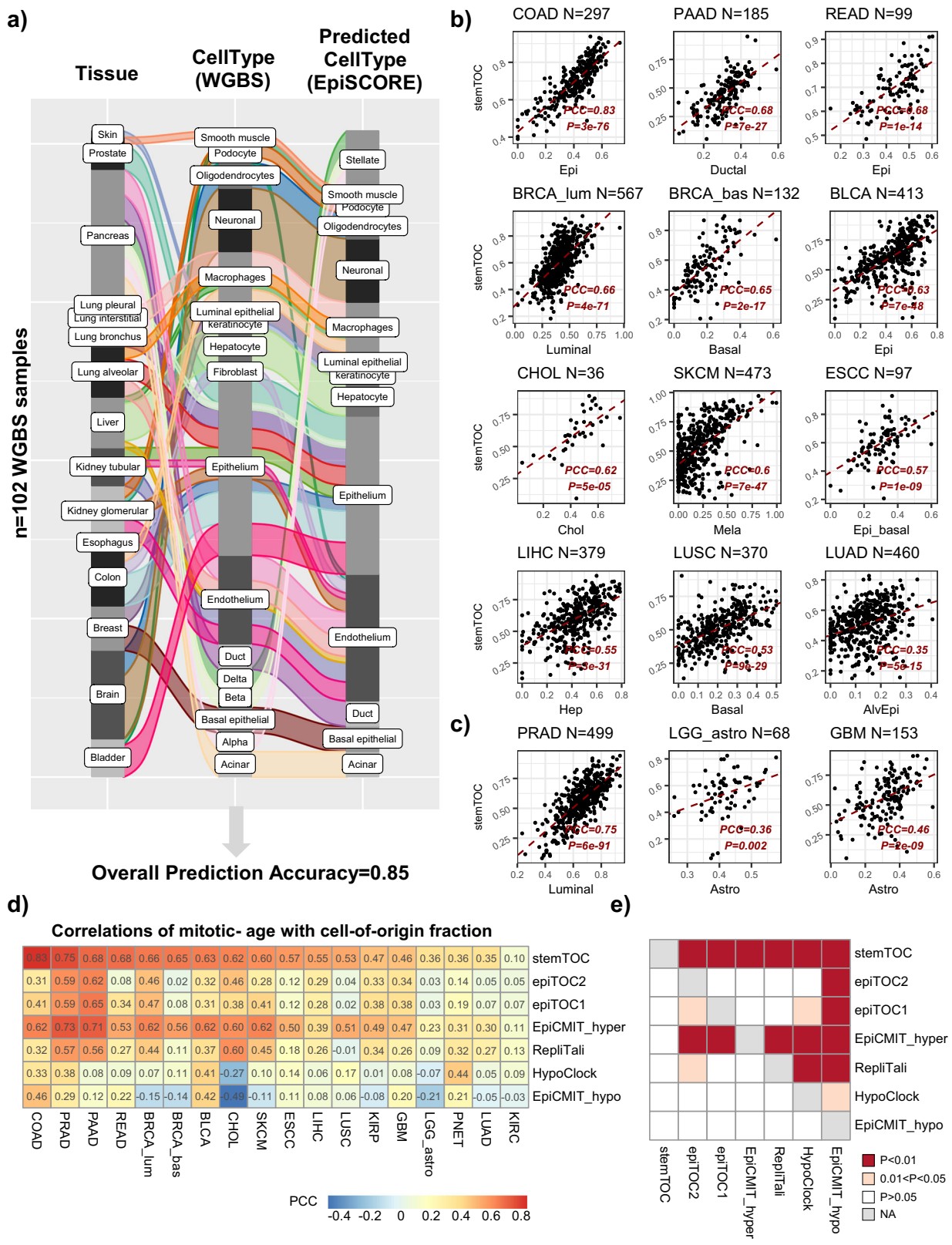

**Overall Prediction Accuracy=0.85**

inform on the potential cell-of-origin in tumor types where this is controversial or less well established. For instance, in low-grade glioma (LGG) and glioblastoma multiforme (GBM), mitotic age correlated most strongly with astrocyte fraction (Fig. 3c, Supplementary Fig. 11b), favoring an astrocyte-like progenitor as the putative cell of origin[60]. Prostate adenocarcinoma (PRAD) has long been linked to a luminal phenotype[61,62], yet studies have also indicated a potential role for basal

cells in the initiation of this cancer type[63]. Our analysis is consistent with the known luminal-cell expansion associated with PRAD (Fig. 3c, Supplementary Fig. 11b), and the prevailing view that the cell-of-origin of human PRAD is a luminal progenitor cell[64–66].

Of note, the correlation strengths with tumor cell-of-origin fraction obtained with stemTOC were significantly stronger than those obtained with the other 6 counters (Fig. 3d, e). Moreover,

**Fig. 3 | Correlation of stemTOC with tumor cell-of-origin fraction. a** Alluvial plot displaying the validation of EpiSCORE on sorted WGBS samples from Loyfer et al. DNAm atlas. The left and middle bars label the tissue-type and cell type of the WGBS sample from Loyfer et al. DNAm atlas. Right bar labels the predicted cell type using EpiSCORE. Overall prediction accuracy is given below. **b** Scatterplot of stemTOC's relative mitotic age estimate (y-axis) vs. the presumed cell-of-origin fraction derived with EpiSCORE (x-axis) for selected TCGA cancer types. The number of tumor samples is indicated above the plot. For each panel, we provide the Pearson Correlation Coefficient (PCC) and P-value from a linear regression. Of note, in each panel, we are displaying the cell type that displayed the strongest correlation with mitotic age and this coincides with the presumed cell-of-origin. **c** As (**b**), but for 3 tumor types where the tumor cell-of-origin is less well established or controversial.

**d** Heatmap of Pearson correlations for all 7 mitotic clocks and across all 18 cancer types. COAD($n = 297$), PRAD($n = 499$), PAAD($n = 185$), READ($n = 99$), BRCA_lum($n = 567$), BRCA_bas($n = 132$), BLCA($n = 413$), CHOL($n = 36$), SKCM($n = 473$), ESCC($n = 97$), LIHC($n = 379$), LUSC($n = 370$), KIRP($n = 276$), GBM($n = 153$), LGG_astro($n = 68$), PNET($n = 45$), LUAD($n = 460$), KIRC($n = 320$). **e** Heatmap displaying one-tailed paired Wilcoxon rank sum test P-values, comparing clocks to each other, in how well their mitotic age correlates with the tumor cell-of-origin fraction. Each row indicates how well the corresponding clock's mitotic age estimate performs in relation to the clock specified by the column. The paired Wilcoxon test is performed over the 18 cancer types. RepliTali results are for the probes restricted to 450k beadarrays.

none of the other 6 mitotic counters achieved 100% accuracy in predicting the putative cell-of-origin (Supplementary Fig. 11d). We also combined epiCMIT-hyper and epiCMIT-hypo into epiCMIT following Duran-Ferrer et al.[33], but in line with epiCMIT-hypo's worse performance, stemTOC and epiCMIT-hyper also outperformed epiCMIT (Supplementary Fig. 12). However, all mitotic counters were universally accelerated in TCGA cancer types, as compared to their respective age-matched normal-adjacent tissue (Supplementary Fig. 13). To further explore the relationship between mitotic counters, we computed correlations between each pair of counters across all sorted immune cell subsets, the normal tissues from GTEX and the normal-adjacent and tumor samples from TCGA, demonstrating that the hypermethylated counters are well-correlated with each other and to a lesser extent also with RepliTali (Supplementary Fig. 14–16), which is noteworthy given the aforementioned relatively little CpG overlap between them (Supplementary Fig. 9). In summary, the correlation of stemTOC's mitotic age with the tumor cell-of-origin fraction underscores its potential to quantify cancer risk.

### stemTOC predicts increased mitotic age in precancerous lesions

We next assembled 9 DNAm datasets representing different normal healthy tissues and corresponding age-matched precancerous conditions ("Methods" section). Mitotic age, as estimated with stemTOC, was significantly higher in the precancerous tissue in each of these datasets (Fig. 4a). For instance, stemTOC's mitotic age could discriminate neoplasia from benign lesions in the prostate, intestinal metaplasia from normal gastric mucosa, or Barrett's esophagus from normal esophagus. Some of the other counters did not provide the expected discrimination, and a formal comparison of all counters across the 9 datasets revealed an overall improved performance of stemTOC (Supplementary Fig. 17). Using EpiSCORE we next estimated cell-type fractions in these datasets, and broadly speaking, for most tissue-types we observed correlations of the mitotic age with tumor cell-of-origin fraction in the precancerous lesions and in cancer itself, but not in the histologically normal tissue, although in many cases the number of samples with normal histology was much lower, which limits power to detect associations in this subgroup (Supplementary Fig. 18). For instance, in colon adenoma there was a clear correlation ($R = 0.75$, $P = 3e-8$, $n = 39$) but not so in normal colon ($R = 0.44$, $P = 0.27$, $n = 8$), likely due to the much lower number of normal samples (Supplementary Fig. 18). In premalignant cholangiocarcinoma (CCA) lesions, there was a strong correlation with cholangiocyte fraction ($R = 0.45$, $P < 0.001$, $n = 60$), which was not observed in normal liver samples ($R = 0.16$, $P = 0.25$, $n = 50$), probably because the cholangiocyte fraction is much lower in these normal samples (Supplementary Fig. S18). Correlations between stemTOC's mitotic age with tumor cell-of-origin fraction in normal-adjacent tissue from the TCGA was broadly speaking also observed (e.g. COAD, BRCA, PRAD), but there were also exceptions (e.g. KIRP, LUAD) (Supplementary Fig. 19). Of note, using an upper quantile over the 371 stemTOC CpGs to define the mitotic age generally resulted in larger effect sizes as well as stronger correlations with cancer-status

and tumor cell-of-origin fraction compared to taking an average, highlighting the importance of taking the underlying stochasticity of DNAm patterns into account (Supplementary Fig. 20). Overall, these data underscore the potential of stemTOC's mitotic age to indicate cancer risk in precancerous lesions.

### Correlation of mitotic age with smoking and obesity-associated inflammation

Smoking and obesity are two main cancer-risk factors that promote inflammation and which can increase the tissue's intrinsic rate of stem-cell division[67,68]. Hence, one would expect the mitotic age of a tissue to correlate with the level of exposure to such cancer-risk factors[2,8,67,69–75]. To assess this, we analyzed an Illumina 450k DNAm dataset of 790 buccal swabs from healthy women all aged 53 at sample draw ("Methods" section)[76]. Buccal swabs contain approximately 50% squamous epithelial and 50% immune cells[77] and we reasoned that the mitotic age in these buccal swabs should correlate with both the squamous epithelial fraction as well as an individual's lifelong smoking habit. Estimating epithelial and immune-cell fractions following our validated HEpiDISH procedure[77], we observed a strong correlation of stemTOC's mitotic age with the squamous epithelial fraction, but importantly also a significant positive correlation with smoking status, which was independent of epithelial fraction (Multivariate linear regression, $P = 5e-5$, Fig. 4b).

Epithelial cells in lung tissue are the cell-of-origin of lung cancer, and so we next estimated stemTOC's mitotic age in over 200 normal lung-tissue samples from eGTEX[45], also encompassing smokers, ex-smokers, and never-smokers. Fractions for 7 cell types including epithelial cells were estimated using EpiSCORE ("Methods" section, Fig. 4c)[38]. In this cohort, donors were of different ages, and correspondingly, we observed a strong correlation between stemTOC's mitotic age with chronological age (Fig. 4c). Importantly, multivariate regression analysis including age, epithelial fraction, and smoking status, revealed a significant positive correlation of mitotic age with smoking-exposure, with the mitotic age of older individuals displaying bigger differences between smokers and non-smokers (Fig. 4c).

As another example, we focused on an EPIC DNAm dataset of liver-tissue samples from 325 obese individuals diagnosed with non-alcoholic fatty liver disease (NAFLD)[78], of which 210 displayed no fibrosis (grade-0), with the rest displaying severe fibrosis (grade 3–4) ("Methods" section). Fractions for 5 cell types including hepatocytes were estimated using EpiSCORE ("Methods" section)[38], which confirmed the known reduction of hepatocyte fraction in NAFLD[78] (Fig. 4d). We observed a strong correlation of stemTOC's mitotic age with chronological age, as well as with disease stage, both of which were significant in a multivariate regression analysis that also included hepatocyte fraction (Fig. 4d). Thus, the increased mitotic age with NAFLD stage is observed despite the reduction in hepatocyte fraction, attesting to stemTOC's sensitivity. Overall, these data support the view that stemTOC can track increased mitotic age in disease-relevant tissues exposed to major cancer-risk factors.

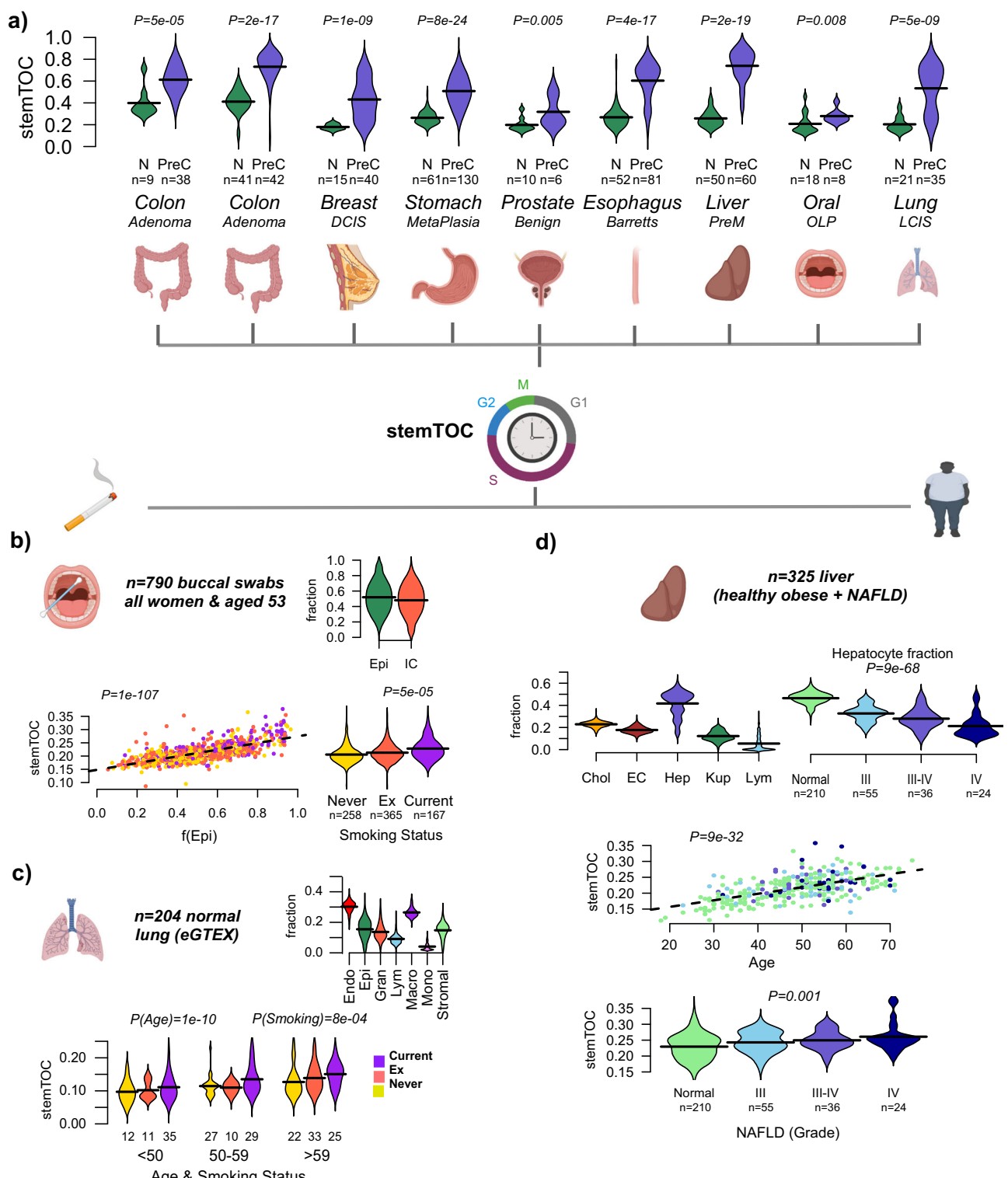

## stemTOC correlates with clock-like somatic mutational signature−1

Next, we explored the relationship of stemTOC with somatic mutational "clock-like" signatures[11,34,79]. Among the somatic mutational signatures derived from the TCGA cancer samples, two single-base substitution signatures SBS1 and SBS5 (here termed MS1 and MS5) have been shown to be associated with chronological age, with MS1 in particular being linked to deamination of methylated CpG dinucleotides resulting in $C > T$ mutations[11]. Given that somatic mutations and DNAm data in the same normal tissue specimens have not yet been

extensively characterized, we asked if the mutational loads in TCGA cancer samples correlate with the estimated total (lifetime) number of stem-cell divisions (TNSC), as derived using the Vogelstein−Tomasetti estimates and chronological age at diagnosis ("Methods" section). As remarked previously[79], although cancers display a clear acceleration of mitotic age, a broad correlation between a tumor's mutational signature load and the baseline (normal) number of stem-cell divisions is expected if the signature is of a mitotic nature. Vindicating Alexandrov et al.[11], MS1 displayed a correlation with the total number of stem-cell divisions (TNSC) across cancer types, whilst MS5 did not (Fig. 5a, b). To

**Fig. 4 | stemTOC correlates with cancer risk and cancer-risk factors. a** Violins display the distribution of stemTOC's mitotic age for normal tissue (N) and age-matched precancerous (PreC) samples for 9 different DNAm datasets. Number of samples in each category is displayed below x-axis, as well as the tissue-type and the nature of the precancerous sample. DCIS ductal carcinoma in situ, PreM pre-malignant, OLP oral lichen planus, LCIS lung carcinoma in situ. *P*-value derives from a one-tailed Wilcoxon rank sum test. **b** Top: Violin plots depicting the estimated epithelial and immune cell fraction in 790 buccal swabs from women all aged-53 at sample draw. Bottom left: Scatterplot of stemTOC (y-axis) against estimated squamous epithelial fraction (x-axis) for the same 790 buccal swabs. Samples are colored by their lifelong smoking habit (current, ex-smoker, and never-smoker). Two-tailed *P*-value is from a multivariate linear regression including epithelial fraction and smoking status. Bottom right: Violin plots display stemTOC versus smoking status. Two-tailed *P*-value is from a multivariate linear regression including smoking status and squamous epithelial fraction. **c** Top: Violins display the esti-mated cell-type fraction for 7 lung cell types (Endo endothelial, Epi epithelial, Gran granulocyte, Lym lymphocyte, Macro macrophage, Mono monocyte, Stromal) in 204 normal lung samples from eGTEX. Bottom: Violin plots display stemTOC ver-sus chronological age and smoking status in the same 204 normal lung-tissue samples. Number of samples in each smoking category and age-group is shown below each box. Violin colors label smoking status as in (**b**). Two-tailed *P*-values for age and smoking status are derived from a multivariate linear regression that included age, smoking, and lung epithelial fraction. **d** Top left: Violins of estimated cell-type fractions for 5 cell types (Hep hepatocytes, Chol cholangiocytes, EC endothelial cells, Kup Kupffer cells, Lym lymphocytes) in 325 liver samples. Top right: Violins of the hepatocyte fraction against NAFLD disease stage. Two-tailed *P*-value is from a linear regression. Middle: Scatterplot of stemTOC (y-axis) against chronological age (x-axis) for the same 325 liver samples, with colors labeling the stage of NAFLD, with color label as in the previous panel. Bottom: Violin plots display mitotic age against NAFLD stage. Two-tailed *P*-value for disease stage derives from a multivariate linear regression that included age and hepatocyte fraction. Generated with Biorender.com.

check that this correlation is independent of chronological age, we computed for each cancer type the median mutational load per Mbp and calendar year over all samples of a given cancer type, and com-pared it to the intrinsic rate of stem-cell division of the corresponding normal tissue-type, revealing a positive correlation for MS1 but not for MS5 (Fig. 5c, d). Like MS1, stemTOC displayed a clear positive corre-lation with TNSC across tumors (Fig. 5e), which was also independent of age (Fig. 5e, f). However, unlike MS1, stemTOC displayed a satura-tion effect, with highly proliferative cancer types displaying high stemTOC-values despite the relatively low turnover rate of the underlying normal tissue (e.g. lung cancers). We verified that this saturation effect is also observed if we define stemTOC in terms of an average DNAm over the 371 CpGs (as opposed to taking the upper quantile), indicating that this is not a technical artifact of our stemTOC definition (Supplementary Fig. 21). Given that in normal-adjacent tis-sues, stemTOC displays correlations with TNSC without evidence of a saturation effect (Fig. 2f, g), this suggests that stemTOC is a sensitive marker of mitotic age.

Next, we asked if stemTOC's mitotic age correlates with the mutational loads of MS1 and MS5. Correlating the median stemTOC's mitotic age of each cancer type to the corresponding median MS1 and MS5 loads, revealed a significant association with MS1, but not for MS5, further attesting to the more mitotic-like nature of MS1 (Fig. 5g, h). Of note, stemTOC correlated with MS1 within most TCGA cancer types, even after adjusting for chronological age (Supplementary Fig. 22). However, in some cancer types (e.g. lung squamous cell carcinoma) no correlation was evident (Supplementary Fig. 22). This indicates that although stemTOC and MS1 both approximate mitotic age, that they are also distinct. This is consistent with reports that MS1 may also reflect somatic mutations arising from other mutational processes[2]. Of note, a strong correlation with tumor cell-of-origin fraction is not seen if one were to use the MS1-load as a proxy for mitotic age (Supple-mentary Fig. 23), and correlations of MS1-load with CPE-based tumor purity estimates[59] were weaker than with EpiSCORE-derived tumor cell-of-origin fractions (Supplementary Fig. 24). Overall, these data indicate broad agreement between stemTOC and MS1.

## Discussion

We have here derived a pan-tissue epigenetic mitotic counter (stem-TOC) that avoids as much as possible, the confounding effects of cell-type heterogeneity and chronological age, and have used it in con-junction with a state-of-the-art DNAm atlas encompassing 13 tissue and 40 cell types, to establish a concrete direct link between the mitotic age of a sample and its tumor cell-of-origin fraction in each of 15 cancer types. We note that this result is consistent with mitotic age correlating with tumor purity under the reasonable assumption that tumor cells have a higher mitotic age than the surrounding stroma, likely owing to the tumor cell's higher proliferative potential. Most importantly

though, the potential of stemTOC to detect mitotic age increases was also demonstrated in precancerous lesions from 8 normal tissue-types, as well as in normal oral/lung tissues exposed to smoking and normal liver tissue from obese individuals, which collectively represent sce-narios where "tumor" purity indices have not been defined or vali-dated. As such, our findings are of profound biological and clinical significance.

First, they add substantial weight to the hypothesis put forward by Tomasetti and Vogelstein[3], that the mitotic age of the cell-of-origin is a major determinant of cancer risk, and that it increases with exposure to exogenous cancer-risk factors. Second, the ability to accurately measure mitotic age and tumor cell-of-origin fraction in preneoplastic lesions opens up new avenues for risk prediction as well as preventive and precision oncology. As a concrete example, the increased mitotic age in the buccal swab squamous epithelium of healthy smokers vs non-smokers could potentially be used as a non-invasive tool to monitor cancer risk in relation to oral, lung, and esophageal squamous cell carcinomas. Third, the 371 CpG loci making up stemTOC could form the basis for developing non-invasive targeted bisulfite-sequencing assays on cell-free DNA in serum[20]. Fourth, by comparing stemTOC to somatic mutational signatures MS1 and MS5 in the TCGA samples, we have confirmed that MS1 is clearly mitotic-like, whereas MS5 is not. This is consistent with MS5 (and not MS1) being the dominant somatic mutational signature in post-mitotic neurons[80], and contrasts with a previous study focusing on B cell malignancies which found that the combined epiCMIT-hyper/hypo mitotic age estimate correlated with both MS1 and MS5[33].

It is illuminating to discuss the comparison of stemTOC to MS1 in more detail. Although recent studies have demonstrated a correlation of MS1 with chronological age in normal healthy tissue-types[52,80,81], the analogous correlation of DNAm-based mitotic age with chronological age has been more widely demonstrated across many more normal tissue-types, including precancerous lesions. When correlating a molecular clock's mitotic age to chronological age in cancer samples, it is worth pointing out that for highly proliferative tumor types arising from tissues with a relatively low intrinsic rate of stem-cell division (e.g. liver, pancreas, lung), the estimated mitotic age would not be expected to correlate well with age at diagnosis since for these cancer samples the majority of cell divisions would have occurred after tumor onset. This is exactly the pattern seen for stemTOC, which displayed a more non-linear correlation and saturation effect with age across tumor samples, even if defined by an average DNAm over the stemTOC CpGs. In contrast, in the same tumors, MS1, which is effectively also an average molecular load estimate, displayed a more linear pattern. In contrast to tumors, in normal tissue stemTOC displays strong linear correlations with age without evidence of any saturation effect. Although normal samples with matched DNAm and somatic muta-tional profiles representing different tissue-types are still lacking to

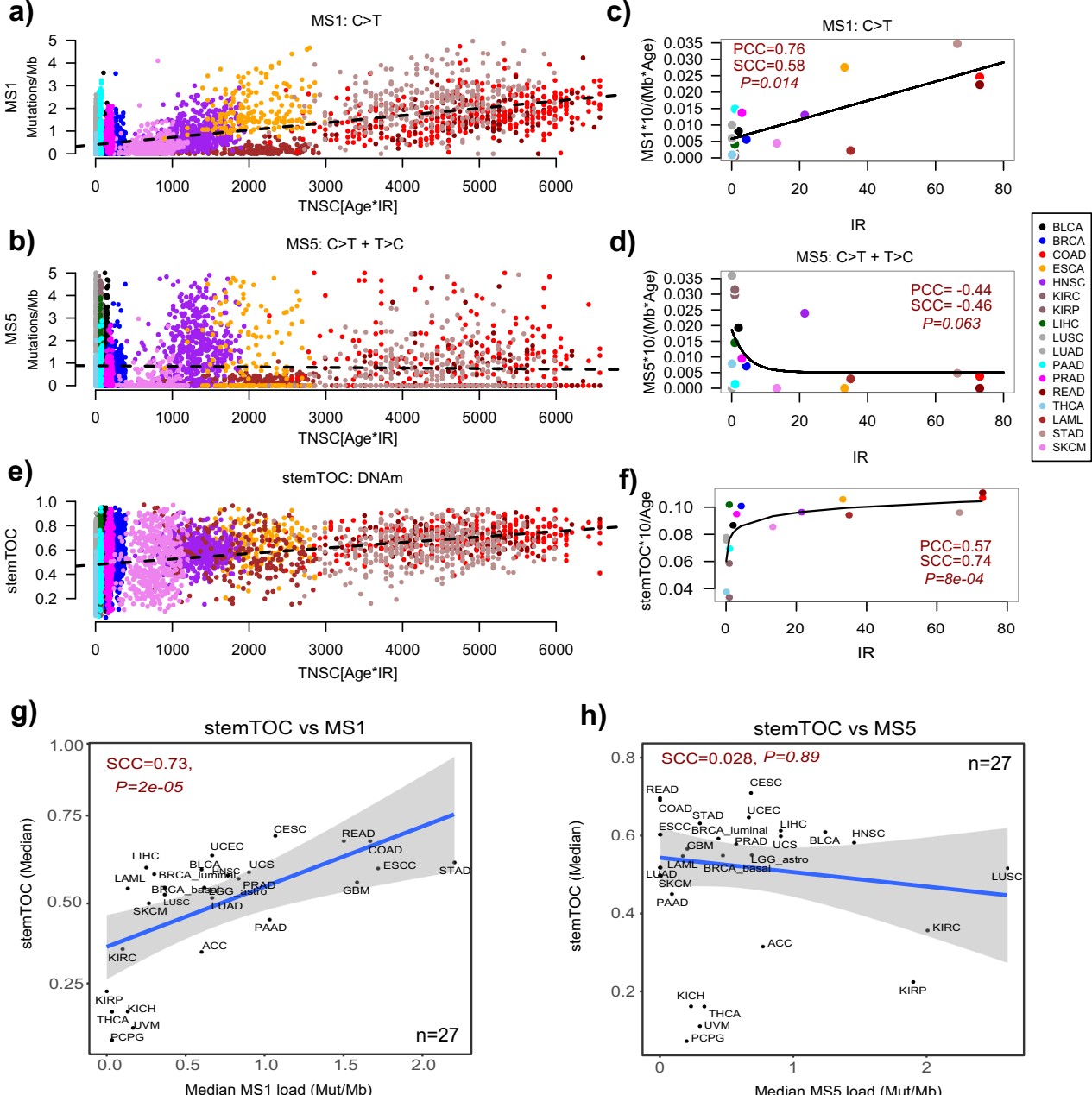

**Fig. 5 | Relation of stemTOC with somatic mutational MS1 and MS5 signatures.**
**a** Scatterplot of MS1-signature load (Mutations/Mb, y-axis) against the estimated total number of stem-cell divisions (Age*intrinsic annual rate of stem-cell division of the corresponding normal tissue (IR), x-axis) for >7000 cancer samples from 17 TCGA cancer types. Only cancer types with an IR estimate in normal tissue were used. Two-tailed *P*-values derive from a multivariate linear regression including age as a covariate. **b** As (**a**) but for somatic mutational signature MS5. **c** Median value of age-adjusted MS1-signature load (multiplied by 10 to reflect change over a decade) for each cancer type vs the annual intrinsic rate of stem-cell division of the corresponding normal tissue-type. Both Pearson (PCC) and Spearman (SCC) correlation

coefficients are given. Two-tailed *P*-value tests for significance of SCC. Fitted line is that of a linear regression model as this outperformed non-linear models. **d** As (**c**) but for MS5. Best fit was for a non-linear decreasing function. **e** As (**a**), but for stemTOC's mitotic age. **f** As (**c**), but for stemTOC, adjusted for chronological age and multiplied by 10 to reflect DNAm change over a decade. Best fit was for a non-linear increasing function displaying a saturation effect, as shown. **g** Scatterplot of the median stemTOC value of each TCGA cancer type (y-axis) vs the median mutational load (mutations per Mb) of that cancer type, for MS1. Spearman's correlation coefficient and *P*-value are given. Regression line with standard error interval is shown. **h** As (**g**) but for MS5.

allow for an objective comparison, the lack of extensive somatic mutational data in normal tissues underscores the greater intrinsic difficulty of detecting somatic mutations in such tissues[80]. Indeed, the advantage of DNAm over somatic mutations as a technically more feasible substrate to track mitotic age in normal and preneoplastic tissues, specially over the potentially shorter time scales between preneoplastic and cancer stages[36], should not be surprising given that the rate at which DNAm changes are acquired in normal cells is about

10–100 times higher than somatic mutations[82]. Thus, DNAm could be more useful than somatic mutations to track the evolution of pre-cancerous states on shorter time scales[36].

The benchmarking analysis of stemTOC against previous DNAm-based clocks also revealed an important biological insight: in general, clocks anchored on sites gaining methylation, specially stemTOC and epiCMIT-hyper, appear to provide better proxies of mitotic age in normal adult tissues compared to HypoClock and epiCMIT-hypo,

which are based on hypomethylation. However, a caveat as far as HypoClock is concerned, is that this clock's assessment on Illumina beadarray data only considers a small fraction of the millions of solo-CpGs that were originally proposed to lose DNAm with cell division[18]. RepliTali, a clock trained and optimized on a subset of solo-CpGs with representation on EPIC beadarrays performed much better than HypoClock. Correspondingly, the comparison of stemTOC (restricted to common 450k+EPIC probes) to RepliTali (trained on EPIC probes) only revealed a relatively minor improvement: whilst on most 450k datasets (a comparison that would favor stemTOC), we observed a consistent improvement of stemTOC over RepliTali, on EPIC datasets the improvements were more marginal. For instance, on the whole blood and eGTEX EPIC datasets, stemTOC's mitotic age was more strongly correlated with chronological age, and in a few datasets including high-turnover tissues like colon and blood, RepliTali's mitotic age did not pass the statistical significance threshold. Upon adjustment for CTH, RepliTali's correlations became significant, an indication that a small fraction of RepliTali's CpGs may be confounded by CTH. Overall, our data reinforces the view that DNAm gains at specific genes that are initially unmethylated in fetal tissue is a reliable way to track mitotic age, especially in the context of normal adult tissue turnover[30]. And whilst the results comparing the specific clocks favor the ones based on hypermethylation, this does not mean that the optimal subset of CpGs for measuring mitotic age are necessarily those gaining DNAm with cell division. Indeed, it is entirely plausible that a subset of solo-CpGs that are currently underrepresented on Illumina beadarrays, could lead to further improvements in mitotic-age prediction.

Any potential residual confounding by CTH and chronological is also worth discussing further. First, it is important to stress that CTH can confound mitotic age estimates in three distinct ways. One way is if the CpGs making up the counter are cell-type specific DNAm markers, i.e. if these CpGs display big differences in DNAm (typically >50% DNAm change) between cell types. In such a scenario, variations in cell-type composition between bulk-samples could affect relative mitotic age estimates. In relation to this, it is worth pointing out that by construction, stemTOC's CpGs (as well as those defining epiTOC1/2[16,30]) are not cell-type specific markers as defined in a suitably defined ground state (fetal-stage). Furthermore, we have shown that stemTOC's CpGs do not display big DNAm differences between age-matched sorted cell and tissue-types, a clear indication that these CpGs are not cell-type specific markers of adult cell or tissue-types. A second way in which CTH could bias mitotic age estimates is through the selection of CpGs that only change with cell division in a specific cell type, so that they don't generalize to other cell types and tissues. We also addressed this type of confounding, by ensuring that stemTOC CpGs correlate with cell division in cell lines representing different cell types (fibroblasts, endothelial cells and smooth muscle). In addition, because these were selected by comparing DNAm changes in-vitro with and without treatment by cell-cycle inhibitors, these stemTOC CpGs are not confounded by chronological age. Hence, by further requiring that these same CpGs also change with chronological age in-vivo whilst adjusting for 12 immune-cell fractions in blood, this likely selects for a subset of CpGs that change with cell division in-vivo. Finally, the third way in which CTH can influence mitotic age estimates is when applying the mitotic counter to a bulk-tissue sample which is made up of different cell types. Even if the CpGs making up the counter are not cell-type specific, it is clear that different cell types in a bulk-sample may have different mitotic ages, hence the mitotic age estimate reflects a weighted average over the mitotic ages of each cell type with the weights reflecting their cell-type proportions. Whilst we acknowledge that stemTOC is unable to address this type of confounding, the observed strong correlation between stemTOC's mitotic age and tumor cell-of-origin fraction in each of 15 TCGA cancer types, demonstrates that this is not a major source of confounding, probably because the mitotic age of the highly proliferative cancer or precancer cells

dominates the average estimate. A related challenge and limitation is that stemTOC's mitotic age estimate not only reflects an average over cell types, but also an average over the cells that make up the differentiation hierarchy within a lineage. As such, the mitotic-age estimate not only reflects the number of cell divisions of the underlying long-lived stem-cell pool, but more broadly also the cell divisions of short-term stem and progenitor cell expansions and their population sizes[83]. In this regard though, it is worth pointing out that it is mainly and only the DNAm changes that accumulate in the long-lived stem-cell pool, and which are inherited by progenitors and differentiated cells, that are able to accumulate in the cell population at large.

Finally, this study has also highlighted the importance of the underlying stochastic DNAm changes in aging and precancerous lesions[19,32,84]. Indeed, by approximating mitotic age with an upper quantile statistic over the pool of mitotic CpGs (as opposed to taking an ordinary average), we can better account for the inherent inter-CpG and inter-subject stochasticity, enabling the identification of DNAm outliers that very likely reflect epigenetic mosaicism and subclonal expansions[31,32]. In line with this, these DNAm outliers led to mitotic age estimates displaying stronger correlations with cancer and tumor cell-of-origin fraction. On the other hand, these DNAm outliers may also not necessarily mark the preneoplastic clones of higher mitotic age but those characterized by subtle quenching or even silencing of tissue-specific developmental genes, as such silencing may confer a selective advantage[49,85,86].

In summary, this work demonstrates how a DNAm-based mitotic age counter can detect subtle increases of mitotic age in the tumor cell-of-origin of normal and precancerous tissues, opening up new biotechnological opportunities for developing early detection and cancer-risk prediction strategies.

## Methods

All data analyzed in this manuscript has already been published in the respective publications, as described below and in the "Data availability" section. As such, this research complies with all ethical regulations.

### Illumina DNAm datasets used in the construction of stemTOC

**Fetal tissue DNAm sets.** We obtained and normalized Illumina 450k data from the Stem-Cell Matrix Compendium-2 (SCM2)[87], as described by us previously[16]. There were a total of 37 fetal tissue samples encompassing 10 tissue-types (stomach, heart, tongue, kidney, liver, brain, thymus, spleen, lung, adrenal gland). We also obtained Illumina 450k data of 15 cord-blood samples[88] and 34 fetal-tissue samples from Slieker et al.[89] encompassing amnion, muscle, adrenal, and pancreas. Both sets were normalized like the SCM2 data, i.e. by processing idat files with *minfi*[90] followed by type-2 probe bias adjustment with BMIQ[91].

**Cell-line data from Endicott et al.[24].** The Illumina EPICv1 cell-line data was downloaded from GEO under accession number GSE197512. Raw idat files were processed with *minfi* and BMIQ normalized, resulting in a normalized DNAm data matrix defined over 843,298 probes and 182 "baseline-profiling" samples. A total of 6 human cell lines were used, including AG06561 (skin fibroblast), AG11182 (veil endothelial), AG11546 (iliac vein smooth muscle), AG16146 (skin fibroblast), AG21839 (neonatal foreskin fibroblast), and AG21859 (foreskin fibroblast). One cell line (AG21837, skin keratinocyte), which displayed global non-monotonic DNAm patterns with population doublings (PDs) was removed.

**Whole-blood DNAm datasets.** We used 3 Illumina whole-blood datasets. One EPIC dataset encompassing 710 samples from Han Chinese was processed and normalized as described by us previously[92]. Briefly, idat files were processed with minfi, followed by BMIQ type-2 probe adjustment, and due to the presence of beadchip effects, data was

further normalized with ComBat[93]. Another dataset is an Illumina 450k set of 656 samples from Hannum et al.[94]. The normalization of this dataset is as described by us previously[92]. Finally, we also analyzed a 450k dataset from Johansson et al.[95]. This dataset was obtained from NCBI GEO website under accession number GSE87571. The file "GSE87571_RAW.tar" containing the IDAT files was downloaded and processed with *minfi* R-package. Probes with *P*-values <0.05 across all samples were kept. Filtered data was subsequently normalized with BMIQ, resulting in a normalized data matrix for 475,069 probes across 732 samples.

## Independent buccal swab and cord blood Illumina DNAm datasets

We analyzed two independent EPIC DNAm datasets to assess the robustness and stability of the 371 stemTOC CpGs. One dataset consists of 44 buccal swabs from infants[96] and the other of 128 cord blood samples from neonates[97]. Briefly, idat files were downloaded from GEO under accession numbers GSE229463 and GSE195595, respectively, and subsequently processed with *minfi*[90] and BMIQ[91] as described for the other datasets.

## Illumina DNAm cancer datasets

The SeSAMe[98] processed Illumina 450k beta value matrices were downloaded from Genomic Data Commons Data Portal (https://portal.gdc.cancer.gov/) with TCGAbiolinks[99]. We analyzed 32 cancer types (ACC, BLCA, BRCA, CESC, CHOL, COAD, DLBC, ESCA, GBM, HNSC, KICH, KIRC, KIRP, LAML, LGG, LIHC, LUAD, LUSC, MESO, PAAD, PCGP, PRAD, READ, SARC, SKCM, STAD, TGCT, THCA, THYM, UCEC, UCS, UVM). We did further processing of the downloaded matrices as follows: For each cancer type, probes with missing values in more than 30% samples were removed. The missing values were then imputed with impute.knn ($k = 5$)[100]. Then, the type-2 probe bias was adjusted with BMIQ[91]. Technical replicates were removed by retaining the sample with the highest CpG coverage. Clinical information on TCGA samples was downloaded from Liu et al.[101]. In addition, we analyzed an Illumina 450k DNAm dataset from Pipinikas et al.[102], consisting of 45 primary pancreatic neuroendocrine tumors (PNETs). Processing of the idat files and QC was performed with minfi[90], impute, and BMIQ as described for the TCGA datasets.

## Illumina DNAm dataset from eGTEX

The *ChAMP*[103] processed Illumina EPIC beta-valued data matrix was downloaded from GEO (GSE213478)[45]. The dataset includes 754,119 probes and 987 samples from 9 normal tissue-types: breast mammary tissue ($n = 52$), colon transverse ($n = 224$), kidney cortex ($n = 50$), lung ($n = 223$), skeletal muscle ($n = 47$), ovary ($n = 164$), prostate ($n = 123$), testis ($n = 50$), and whole blood ($n = 54$) derived from 424 GTEX subjects.

## Illumina DNAm datasets of sorted immune-cell types

We analyzed Illumina 450k DNAm data from BLUEPRINT[42], encompassing 139 monocyte, 139 CD4+ T-cell, and 139 neutrophil samples from 139 subjects. This dataset was processed as described by us previously[104]. In addition, we analyzed Illumina 450k DNAm data from GEO (GSE56046) encompassing 1202 monocyte and 214 naïve CD4+ T-cell samples from Reynolds et al.[105], and GEO (GSE56581) encompassing 98 CD8+ T-cell samples from Tserel et al.[106]. Data was normalized as described by us previously[107]. We also analyzed Illumina 450k DNAm data from EGA (EGA: EGAS00001001598) encompassing 100 B cell, 98 CD4+ T cell, and 104 monocyte samples from Paul et al.[43]. Data was normalized as described by us previously[43].

## Other Illumina whole-blood DNAm datasets

We analyzed a large collection of 18 whole-blood cohorts encompassing a total of 14,515 samples. This is a subset of the over 20,000 samples we previously analyzed[41], consisting mostly of healthy samples. Illumina DNAm data was normalized as described by us previously[41]. Briefly, the 18 whole-blood cohorts chosen here were: *LiuMS(n = 279):* The 450k dataset from Kular et al.[108] was obtained from the NCBI GEO website under the accession number GSE106648.

*Song (n = 2052):* The EPIC dataset from Song et al.[109] was obtained from the NCBI GEO website under the accession number GSE169156.

*HPT-EPIC (n = 1394) & HPT-450k (n = 418):* These datasets[110] were obtained from the NCBI GEO websites under the accession numbers GSE210255 and GSE210254.

*Barturen (n = 5740):* The EPIC dataset from Barturen et al.[111] was obtained from GEO under accession number GSE179325.

*Airwave (n = 1032):* The EPIC dataset from the Airwave study[112] was obtained from GEO under accession number GSE147740.

*VACS (n = 529):* The 450k dataset from Zhang et al.[113] was obtained from GEO under accession number GSE117860.

*Ventham (n = 380):* The 450k dataset from Ventham et al.[114] was obtained from NCBI GEO website under accession number GSE87648.

*Hannon−1 and 2 (n = 636 and 665):* The 450k datasets from Hannon et al.[115,116] were obtained from NCBI GEO websites under accession numbers GSE80417 and GSE84727.

*Zannas (n = 422):* This 450k dataset[117] was obtained from GEO under accession number GSE72680

*Flanagan/FBS (n = 184):* The 450k dataset Flanagan et al.[118] was obtained from NCBI GEO under the accession number GSE61151.

*Johansson (n = 729):* The 450k dataset from Johansson et al.[95] was obtained from GEO under accession number GSE87571.

*Lehne (n = 2707):* This 450k DNAm dataset consists of peripheral blood samples[119], and we used the already QC-processed and normalized version previously described by Voisin et al.[120].

*TZH(n = 705), Hannum (n = 656), LiuRA (n = 689), Tsaprouni (n = 464):* The TZH (EPIC)[92], Hannum (450k)[94], LiuRA (450k)[121], Tsaprouni (450k)[122] were downloaded and normalized as described by us previously[92].

## Illumina DNAm atlas from Moss et al.

We downloaded Illumina 450k & EPIC DNAm data (idat files)[44] from GEO under accession number GSE122126. We processed the 450k and EPIC data separately with *minfi*, removing low-quality controls (detection *P*-value <0.05) and further normalized the data with BMIQ, resulting in a merged dataset defined over 449,156 probes and 28 samples. These 28 samples included sorted pancreatic beta cells ($n = 4$), pancreatic ductal ($n = 3$), pancreatic acinar ($n = 3$), adipocytes ($n = 3$), hepatocytes ($n = 3$), cortical neurons ($n = 3$), leukocyte ($n = 1$), lung epithelial ($n = 3$), colon-epithelial ($n = 3$), and vascular endothelial ($n = 2$).

## Illumina DNAm datasets of sorted neurons

We analyzed a total of 4 Illumina DNAm datasets of sorted neurons (neuronal nuclei NeuN+), in all cases only using normal control samples. In all cases, raw Illumina EPIC/450k DNAm files were downloaded from GEO under accession numbers GSE112179 ($n = 28$, post-mortem frontal cortex)[123], GSE41826 ($n = 29$, post-mortem frontal cortex)[124], GSE66351 ($n = 16$, post-mortem human brains)[125] and GSE98203 ($n = 29$, post-mortem human brains)[126]. In all cases, probes not detected above the background were assigned NA. CpGs with coverage 0.99 were kept (in the 2nd and 4th sets, this threshold was relaxed slightly to 0.98). The remaining NAs were imputed with impute.knn ($k = 5$). Finally, the beta-valued data matrix was normalized with BMIQ.

## Illumina DNAm datasets from normal healthy and normal "at cancer-risk" tissue

1. *Lung preinvasive dataset*: This is an Illumina 450k DNAm dataset of lung-tissue samples that we have previously published[76]. We

used the normalized dataset from Teschendorff et al.[76] encompassing 21 normal lungs and 35 age-matched lung-carcinoma in situ (LCIS) samples, and 462,912 probes after QC. Of these 35 LCIS samples, 22 progressed to an invasive lung cancer (ILC).

2. *Breast preinvasive dataset*: This is an Illumina 450k dataset of breast tissue samples from Johnson et al.[127]. Raw idat files were downloaded from GEO under accession number GSE66313, and processed with *minfi*. Probes with sample coverage <0.95 (defined as a fraction of samples) with detected ($P < 0.05$) $P$-values were discarded. The rest of the unreliable values were assigned NA and imputed with knn ($k = 5$)[100]. After BMIQ normalization, we were left with 448,296 probes and 55 samples, encompassing 15 normal-adjacent breast tissue and 40 age-matched ductal carcinoma in situ (DCIS) samples, of which 13 were from women who later developed an invasive breast cancer (BC).

3. *Gastric metaplasia dataset*: Raw idat files were downloaded from GEO (GSE103186)[128] and processed with *minfi*. Probes with over 99% coverage were kept and missing values imputed using impute R-package using impute.knn ($k = 5$). Subsequently, data was intra-array normalized with BMIQ, resulting in a final normalized data matrix over 482,975 CpGs and 191 samples, encompassing 61 normal gastric mucosas, 22 mild intestinal metaplasias, and 108 metaplasias. Although age information was not provided, we used Horvath's clock[129] to confirm that normal and mild intestinal metaplasias were age-matched. This is justified because Horvath's clock is not a mitotic clock[30] and displays a median absolute error of ±3 years[129].

4. *Barrett's Esophagus and adenocarcinoma dataset*: This Illumina 450k dataset[49] is freely available from GEO under accession number GSE104707. Data was normalized as described by us previously[130]. The BMIQ-normalized dataset is defined over 384,999 probes and 157 samples, encompassing 52 normal squamous epithelial samples from the esophagus, 81 age-matched Barrett's Esophagus specimens, and 24 esophageal adenocarcinomas.

5. *Colon adenoma dataset*[131]: Illumina 450k raw idat files were downloaded from ArrayExpress E-MTAB-6450 and processed with minfi. Only probes with 100% coverage were kept. Subsequent data was intra-array normalized with BMIQ, resulting in a normalized data matrix over 483,422 CpGs and 47 samples, encompassing 8 normal colon specimens and 39 age-matched colon adenomas. Although age information was not made publicly available, we imputed them using Horvath's clock, confirming that normals and adenomas are age-matched.

6. *Cholangiocarcinoma (CCA) dataset*[132]: Raw idat files were downloaded from GEO (GSE156299). This is an EPIC dataset and was processed with minfi. Probes with >99% coverage (fraction of samples with $P < 0.05$) were kept. NAs were imputed with impute.knn ($k = 5$). Subsequent data was intra-array normalized with BMIQ, resulting in a normalized data matrix over 854,026 probes and 137 samples, encompassing 50 normal bile duct specimens, 60 premalignant, and 27 cholangiocarcinomas. Normal bile duct, premalignant, and CCA specimens were derived from the same patient. Thus, normal and premalignant samples are mostly age-matched.

7. *Prostate cancer progression dataset*: Illumina 450k raw idat files were downloaded from GEO (GSE116338)[133] and processed with minfi. CpGs with coverage >0.95 were kept. NAs were imputed with impute.knn ($k = 5$). Subsequent data matrix was intra-array normalized with BMIQ. Samples included benign lesions ($n = 10$), neoplasia ($n = 6$), primary tumors ($n = 14$) and metastatic prostate cancer ($n = 6$). In this dataset, benign lesions were significantly older compared to neoplasia and primary tumors, thus this dataset provides a particularly good test that mitotic clocks are measuring mitotic age as opposed to chronological age.

8. *Oral squamous cell carcinoma (OSCC) progression dataset*: The processed beta-valued Illumina 450k DNAm data matrix was downloaded from GEO (GSE123781)[134]. Probes not detected above background had been assigned NA. CpGs with coverage >0.95 were kept. The remaining NAs were imputed with impute.knn ($k = 5$). Then the beta matrix was BMIQ normalized. Samples included 18 healthy controls, 8 lichen planus (putative premalignant condition), and 15 OSCCs. This is the only dataset where premalignant samples older than the healthy controls.

9. *Colon adenoma dataset 2*. Processed beta-valued Illumina 450k DNAm data matrix was downloaded from GEO (GSE48684)[135], encompassing 17 normal-healthy samples, 42 age-matched colon adenoma samples, and 64 colon cancer samples. Probes not detected above background were assigned NA. CpGs with coverage >0.95 were kept. The remaining NAs were imputed with impute.knn ($k = 5$). Then the beta matrix was BMIQ normalized. Although age information was not made publicly available, ages were imputed with Horvath's clock confirming that normals and adenomas are age-matched.

10. *Normal breast Erlangen dataset*: This Illumina 450k dataset is freely available from GEO under accession number GSE69914. Data was normalized as described by us previously[32]. The BMIQ-normalized dataset is defined over 485,512 probes and 397 samples, encompassing 50 normal-breast samples from healthy women, 42 age-matched normal-adjacent samples, and 305 invasive breast cancers. We note that this dataset was excluded from the formal comparison of stemTOC to all other clocks, because this dataset was used to select the upper-quantile threshold in the definition of stemTOC (see later).

### Illumina DNAm datasets of normal tissues exposed to cancer-risk factors

1. *Buccal swabs+smoking (n = 790)*: This Illumina 450k DNAm dataset was generated and analyzed previously by us[76]. We used the same normalized DNAm dataset as in this previous publication. Briefly, the samples derive from women all aged 53 years at sample draw and belong to the MRC1946 birth cohort. This cohort has well-annotated epidemiological information, including smoking-status information. Among the 790 women, 258 were never-smokers, 365 ex-smokers, and 167 current smokers at sample draw.

2. *Lung-tissue+smoking (n = 204)*: This Illumina EPIC DNAm dataset of normal lung-tissue samples derives from eGTEX[45] and was processed as the other tissue datasets from eGTEX. Age information was only provided in age-groups. Smoking status distribution was 89 current smokers, 54 ex-smokers, and 61 never-smokers.

3. *Liver-tissue+obesity (n = 325)*: This Illumina EPIC DNAm dataset of liver tissue is derived from GEO: GSE180474, and encompasses liver tissue samples from obese individuals (minimum BMI = 32.6), all diagnosed with non-alcoholic fatty liver disease (NAFLD)[78]. Of the 325 samples, 210 had no evidence of fibrosis (grade-0), 55 had grade-3 fibrosis, 36 intermediate grade 3-4 fibrosis, and 24 grade-4 fibrosis. We downloaded the processed beta and P-values and only kept probes with 100% coverage across all samples. Data was further adjusted for type-2 probe bias using BMIQ.

### Construction of stemTOC: identification of mitotic CpGs

The construction of stemTOC initially involves a careful selection of CpGs that track mitotic age. Initially, we follow the procedure as described for the epiTOC+epiTOC2 mitotic clocks, identifying CpGs mapping to within 200 bp of transcription start sites and that are constitutively unmethylated across 86 fetal tissue samples encompassing 15 tissue-types. Here, by constitutive unmethylation, we mean

a DNAm beta value <0.2 in each of 86 fetal tissue samples encompassing 13 tissue-types. Of note, this means that all these CpGs occupy the same methylation state in the fetal ground state, i.e. these CpGs are not cell-type or tissue-specific in this ground state. Next, we use the cell-line data from Endicott et al.[24] to further identify the subset of CpGs that display significant hypermethylation with population doublings, but which do not display such hypermethylation when the cell lines are treated with a cell-cycle inhibitor (mitomycin-C) or when the cell-culture is deprived of growth-promoting serum. In detail, for each of 6 cell lines (AG06561, AG11182, AG11546, AG16146, AG21837, AD21859) representing fibroblasts (3), smooth muscle (1), keratinocyte (1) and endothelial cell types, we ran linear regressions of DNAm against population doublings (PD) (a total of 182 samples), identifying for each cell-line CpGs where DNAm increases with PDs (q-value (FDR) < 0.05). Out of a total of 843,98 CpGs, 14,255 CpGs displayed significant hypermethylation with PDs in each of the 6 cell lines. Of these 14,225 CpGs, we next removed those still displaying hypermethylation (unadjusted $P < 0.05$) with days in culture in cell lines treated with mitomycin-C or in cell lines deprived of serum. This resulted in a much-reduced set of 629 "vitro-mitotic" CpG candidates. In the next step, we asked how many of these "vitro-mitotic" CpGs display age-associated hypermethylation in-vivo, using 3 separate large whole-blood cohorts. The rationale for using whole blood is that this is a high-turnover-rate tissue, and so chronological age should be correlated with mitotic age. Moreover, large whole-blood cohorts guarantee adequate power to detect age-associated DNAm changes. And thirdly, by intersecting CpGs undergoing hypermethylation with PDs in-vitro with those undergoing hypermethylation with age in-vivo, we are more likely to be identifying CpGs that track mitotic age in-vivo. We used the 3 large whole-blood datasets as described in the previous section, each containing approximately 700 samples, whilst adjusting for all potential confounders including cell fractions for 12 immune-cell subtypes[40]. Specifically, we used a DNAm reference matrix defined over 12 immune cell types, which we have recently validated across over 23,000 whole-blood samples from 22 cohorts[41], to estimate cell-type proportions using our EpiDISH procedure[136]. These proportions were then included as covariates when identifying age-DMCs in each cohort separately. To arrive at a final set of age-DMCs, we used the directional Stouffer method over the 3 large cohorts to compute an overall Stouffer z-statistic and P-value, selecting CpGs with $z > 0$ and $P < 0.05$. Of the 629 "vitro-mitotic" CpGs, 371 were significant in the Stouffer meta-analysis of whole-blood cohorts. This set of 371 CpGs defined our "vivo-mitotic" CpGs making up stemTOC.

### Estimating relative mitotic age with stemTOC

Given an arbitrary sample with a DNAm profile defined over these 371 CpGs, we next define the mitotic age of the sample as the 95% upper quantile of DNAm values over the 371 vivo-mitotic CpGs. The justification for taking an upper quantile value, as opposed to taking an average is as follows: extensive DNAm data from previous studies indicate that DNAm changes that mark cancer risk (and hence also mitotic age) are characterized by an underlying inter-CpG and inter-subject stochasticity[19,31,32,84]. Specifically, relevant DNAm changes in normal tissues at cancer risk constitute mild outliers in the DNAm distribution representing subclonal expansions, that occur only infrequently across independent subjects, with the specific CpG outliers displaying little overlap between subjects. Thus, for a given pool of mitotic CpG candidates (i.e. the pool of 371 vivo-mitotic CpGs derived earlier), only a small subset of these may be tracking mitotic age in any given subject at-risk of tumor development. Thus, taking an average DNAm over the 371 CpGs is not optimized to capturing the effect of subclonal expansions that track the mitotic age. To understand this, we ran a simple simulation model, with parameter choices inspired by real DNAm data[19,31,32,84], for a reduced set of 20 CpGs and 40 samples representing 4 disease stages (normal, normal at-risk,

preneoplastic, cancer) with 10 samples in each stage. In the initial stage, all CpGs are unmethylated with DNAm beta-values drawn from a Beta distribution $Beta(a = 10, b = 90)$, where $Beta(a,b)$ is defined by the probability distribution

$$p(X = x \mid a,b) \sim x^{a-1}(1-x)^{b-1} \qquad (1)$$

which has a mean $E[X] = a/(a+b)$ and variance $Var[X] = ab/((a+b+1)(a+b)^2)$. For each "normal at-risk" sample, we randomly selected 1–3 CpGs (precise number was drawn from a uniform distribution), simulating DNAm gains for these CpGs by drawing DNAm values from $Beta(a = 3, b = 7)$. For each preneoplastic sample, we randomly selected 5–10 CpGs with DNAm values drawn from $Beta(a = 5, b = 5)$. Finally, for invasive cancer, we randomly selected 11–17 CpGs with DNAm gains drawn from $Beta(a = 8, b = 2)$. Under this model, discrimination of normal-at-risk samples from normal healthy is difficult if one were to average DNAm over 20 CpGs. However, taking a 95% upper quantile (UQ) of the DNAm distribution over the 20 CpGs, one can discriminate the normal at-risk samples. In this instance, a 95% UQ over 20 CpGs corresponds to taking the maximum value over these 20 CpGs. Another way to understand this is by first identifying CpGs that display DNAm outliers in the normal at-risk group compared to normal. This can be done using differential variance statistics[19,137] to identify hypervariable CpGs, or alternatively, by finding CpGs for which a suitable upper quantile over the normal at-risk samples is much greater than the corresponding UQ value over the normal samples. To determine what UQ-threshold may be suitable, we analyzed our Illumina 450k DNAm dataset encompassing 50 normal-breast tissues from healthy women and 42 age-matched normal-adjacent samples ("at-risk" samples) from women with breast cancer[32]. For each of the 371 CpGs, we computed the mean and UQ over the 50 normal-breast samples, and separately over the 42 normal-adjacent "at-risk" samples. We considered a range of UQ thresholds from 0.75 to 0.99. We then compared the difference (i.e. effect size) in the obtained values between the normal-adjacent and normal-healthy, averaging over the 371 CpGs. The effect size increases with higher UQ values. We selected a 95% UQ because at this threshold we maximized effect size without compromising variability (choosing higher UQs leads to increased random variation). Although in this analysis, the UQ is taking across samples, it is reasonable, given the underlying stochasticity of the patterns, to apply the same UQ-threshold across CpGs. Thus, for any independent sample, we define the relative mitotic age of stemTOC as the 95% upper quantile of the DNAm distribution defined over the 371 CpGs. We note however that results in this manuscript are not strongly dependent on the particular UQ value, i.e. results are generally very robust to UQ values in the range 75–95%.

### Estimating cell-type fractions in solid tissues with EpiSCORE and HEpiDISH

In this work we estimate the proportions of all main cell types within tissues from the TCGA using our validated EpiSCORE algorithm[37] and its associated DNAm atlas of tissue-specific DNAm reference matrices[38]. This atlas comprises DNAm reference matrices for lung (7–9 cell types: alveolar epithelial, basal, other epithelial, endothelial, granulocyte, lymphocyte, macrophage, monocyte and stromal), pancreas (6 cell types: acinar, ductal, endocrine, endothelial, immune, stellate), kidney (4 cell types: epithelial, endothelial, fibroblasts, immune), prostate (6 cell types: basal, luminal, endothelial, fibroblast, leukocytes, smooth muscle), breast (7 cell types: luminal, basal, fat, endothelial, fibroblast, macrophage, lymphocyte), olfactory epithelium (9 cell types: mature neurons, immature neurons, basal, fibroblast, gland, macrophages, pericytes, plasma, T-cells), liver (5 cell types: hepatocytes, cholangiocytes, endothelial, Kupffer, lymphocytes), skin (7 cell types: differentiated and undifferentiated keratinocytes, melanocytes, endothelial, fibroblast, macrophages, T-cells),

brain (6 cell types: astrocytes, neurons, microglia, oligos, OPCs and endothelial), bladder (4 cell types: endothelial, epithelial, fibroblast, immune), colon (5 cell types: endothelial, epithelial, lymphocytes, myeloid and stromal) and esophagus (8 cell types: endothelial, basal, stratified, suprabasal and upper epithelium, fibroblasts, glandular, immune). Hence, when estimating cell-type fractions in the TCGA tissue samples, we were restricted to those tissues with an available DNAm reference matrix. EpiSCORE was run on the BMIQ-normalized DNAm data from the TCGA with default parameters and 500 iterations. EpiSCORE was also used to estimate cell-type fractions in solid tissues from non-TCGA datasets. For instance, we used EpiSCORE to estimate 7 lung cell-type fractions (endothelial, epithelial, neutrophil, lymphocyte, macrophage, monocyte, and stromal) in normal lung-tissue samples from eGTEX[45], and 5 liver cell-type fractions (cholangiocyte, hepatocyte, Kupffer, endothelial and lymphocyte) in liver tissue[78]. For the buccal swab dataset, because buccal swabs only contain squamous epithelial and immune-cells, we used the validated HEpiDISH algorithm and associated DNAm reference matrix to estimate these fractions in this tissue[77].

### Validation of EpiSCORE fractions using a WGBS DNAm atlas

Whilst EpiSCORE has already undergone substantial validation[38], here we decided to further validate it against the recent human WGBS DNAm atlas from Loyfer et al.[58], which comprises 207 sorted samples from various tissue-types. Briefly, we downloaded the beta-valued data and bigwig files (hg38) from the publication website. We required CpGs to be covered by at least 10 reads. We then averaged the DNAm values of CpGs mapping to within 200 bp of the transcription start sites of coding genes, resulting in a DNAm data matrix defined over 25,206 gene promoters and 207 sorted samples. The 207 sorted samples represent 37 cell types from 42 distinct tissues (anatomical regions). We validate EpiSCORE in this WGBS-atlas by asking if it could predict the corresponding cell type. This was only done for tissues for which there is an EpiSCORE DNAm reference matrix. For instance, for breast tissue, the WGBS DNAm atlas profiled 3 luminal and 4 basal epithelial samples. Hence, for breast tissue, we applied EpiSCORE with our breast DNAm reference matrix, derived via imputation from a breast-specific scRNA-Seq atlas, to these 7 sorted samples to assess if it can predict the basal/luminal subtypes using a maximum "probability" criterion (using the estimated fraction as a probability). This analysis was done for WGBS-sorted cells from 15 anatomical sites (brain, skin, colon, breast, bladder, liver, esophagus, pancreas, prostate, kidney tubular, kidney glomerular, lung alveolar, lung pleural, lung bronchus, and lung interstitial) using our EpiSCORE DNAm reference matrices for brain, skin, colon, breast, bladder, liver, esophagus, pancreas, prostate, kidney and lung. An overall accuracy was estimated as the number of sorted WGBS samples where the corresponding cell type was correctly predicted.

### Benchmarking against other DNAm-based mitotic clocks

We benchmarked stemTOC against 6 other epigenetic mitotic clocks: epiTOC[16], epiTOC2[30], HypoClock[18], RepliTali[24], epiCMIT-hyper and epiCMIT-hypo[33]. Briefly, epiTOC gives a mitotic score called pcgtAge, which is an average DNAm over 385 epiTOC-CpGs. In the case of epiTOC2, we used the estimated total cumulative number of stem-cell divisions (tnsc). In the case of HypoClock, the score is given by the average DNAm over 678 solo-WCGWs with representation on Illumina 450k arrays. Of note, for Hypoclock, smaller values mean a larger deviation from the methylated ground state. In the case of RepliTali, which was trained on EPIC data, the score is calculated with 87 RepliTali CpGs and their linear regression coefficients as provided by Endicott et al.[24]. Of the 87 RepliTali CpGs, only 30 are present on the Illumina 450k array. As far as epiCMIT[33] is concerned, this clock is based on two separate lists of 184 hypermethylated and 1164 hypomethylated CpGs. Because the biological mechanism by which CpGs gain or lose DNAm during cell division is distinct, the strategy recommended by Ferrer-Durante to compute an average DNAm over the two lists to then select the one displaying the biggest deviation from the ground state as a measure of mitotic age, is in our opinion not justified. Their strategy could conceivably result in the hypermethylated CpGs being used for one sample, and the hypomethylated CpGs being used for another. Instead, in this work we separately report the average DNAm of the hypermethylated and hypomethylated components. Thus, for the hypermethylated CpGs, the average DNAm over these sites defines the "epiCMIT-hyper" clock's mitotic age, whereas for the hypomethylated CpGs we take 1-average(hypomethylated CpGs) as a measure of mitotic age, thus defining the "epiCMIT-hypo" clock.

We compare all mitotic clocks using the following evaluation strategies. In the analysis correlating mitotic ages to the fractions of the tumor cell-of-origin in the various TCGA cancer types, we compare the inferred Pearson Correlation Coefficients (PCCs) between each pair of clocks across the TCGA cancer types using a one-tailed Wilcoxon rank sum test. Likewise, in the analysis where we correlate mitotic age to chronological age in the normal-adjacent samples from the TCGA, in the normal samples from eGTEX, and the sorted immune-cell subsets, we compare the inferred PCCs between each pair of clocks across the independent datasets using a one-tailed Wilcoxon rank sum test. Finally, when assessing the clocks for predicting cancer risk, for each of the 9 datasets with normal-healthy and age-matched "normal at-risk" samples and for each clock, we first computed an AUC-metric, quantifying the clock's discriminatory accuracy. For each pair of mitotic clocks, we then compare their AUC values across the 9 independent datasets using a one-tailed Wilcoxon rank sum test.

### Comparison to stem-cell division rates and somatic mutational signatures

We obtained estimated tissue-specific intrinsic stem-cell division rates of 13 tissue-types (bladder, breast, colon, esophagus, oral, kidney, liver, lung, pancreas, prostate, rectum, thyroid, and stomach) from Vogelstein & Tomasetti[3,8], supplemented with estimates for skin and blood[30], corresponding to 17 TCGA cancer types (BLCA, BRCA, COAD, ESCA, HNSC, KIRP & KIRC, LIHC, LSCC & LUAD, PAAD, PRAD, READ, THCA, LAML, STAD, and SKCM). Thus, for each normal-adjacent sample of the TCGA, we can estimate the total number of stem-cell divisions (TNSC) by multiplying the intrinsic rate (IR) of the tissue with the chronological age of the sample. Somatic mutational clock signature-1 (MS1) and signature-5 (MS5) were derived from Alexandrov et al.[11]. These loads represent the number of mutations per Mbp. Of note, because these mutational loads and stemTOC's mitotic age both correlate with chronological age, in specific analyses we divide these values by the chronological age of the sample to arrive at age-adjusted values.

### Reporting summary

Further information on research design is available in the Nature Portfolio Reporting Summary linked to this article.

## Data availability

The main Illumina DNA methylation 450k/EPIC datasets used here are freely available from the following public repositories: Endicott (182 cell-line samples, GSE197512)[24]. Hannum (656 whole blood, GSE40279, [https://www.ncbi.nlm.nih.gov/geo/query/acc.cgi?acc=GSE40279])[94]; MESA (214 purified CD4+ T-cells and 1202 Monocyte samples, GSE56046 GSE56581)[105]; Tserel (98 CD8+ T-cells, GSE59065)[106]; BLUE-PRINT (139 matched CD4+ T-cells, Monocytes and Neutrophils, EGAS00001001456)[42]; Paul (100 B cells, 98 T cells, and 104 monocytes, EGA: EGAS00001001598)[43]; Liu (335 whole blood, GSE42861)[121]; Pai (n = 28 sorted neurons, GSE112179)[123], Guintivano (n = 29 sorted neurons, GSE41826)[124], Gasparoni (n = 16 sorted neurons, GSE66351)[125] and Kozlenkov (n = 29 sorted neurons, GSE98203)[126]; Gastric tissue (191

normal and metaplasia, GSE103186)[128]; Colon tissue (47 normal and adenoma, E-MTAB-6450)[131] Colon tissue2 (123 normal, adenoma, and cancer samples, GSE48684)[135]; Breast Erlangen (397 normal, GSE69914)[32]; Breast2 (121 normal, GSE101961)[138]; Breast Johnson (55 normal and ductal carcinoma in situ samples, GSE66313)[127]; Liver tissue (137 normal, premalignant and cholangiocarcinoma samples, GSE156299)[132]; Prostate tissue (36 normal, neoplasia, primary and metastatic samples, GSE116338)[133]; Oral tissue (41 normal, lichen planus and oral squamous cell carcinoma samples, GSE123781)[134]; Esophagus (50 normal, 81 Barrett's Esophagus and 24 adenocarcinomas, GSE104707)[49]; Liver-NAFLD ($n = 325$, GSE180474)[78]; SCM2 (37 fetal tissue samples, GSE31848)[87]; Cord Blood (15 samples, GSE72867)[88]; Slieker (34 fetal tissue samples, GSE56515)[89]; eGTEX (987 samples from 9 normal tissue-types, GSE213478)[45]; Buccal Swabs from Infants (44 samples, GSE229463)[96]; Cord Blood from Neonates (128 samples, GSE195595)[97]; TCGA data was downloaded from https://gdc.cancer.gov; The DNAm dataset in buccal cells from the NSHD MRC1946[76] is available by submitting data requests to mrclha.swiftinfo@ucl.ac.uk; see full policy at http://www.nshd.mrc.ac.uk/data.aspx. Managed access is in place for this 69-year-old study to ensure that the use of the data is within the bounds of consent given previously by participants and to safeguard any potential threat to anonymity since the participants are all born in the same week. The DNAm atlas encompassing DNAm reference matrices for 13 tissue-types encompassing over 40 cell types is freely available from the EpiSCORE R-package https://github.com/aet21/EpiSCORE. Source data are provided with this paper. The remaining data are available within the Article, Supplementary Information, or Source Data file. Source data are provided with this paper.

## Code availability

An R-package "EpiMitClocks" with functions to estimate the mitotic age for each of the clocks in this work is freely available from https://github.com/aet21/EpiMitClocks The package comes with a tutorial vignette. The R-package EpiSCORE for estimating cell-type fractions in solid tissue-types is available from https://github.com/aet21/EpiSCORE The package comes with a tutorial vignette. We have also made an OceanCode Capsule available from https://codeocean.com/capsule/3811164/tree.

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

## Acknowledgements

This work was supported by NSFC (National Science Foundation of China) grants, grant numbers 32170652 and 32370699. We would also like to thank the TCGA and everyone who supports open-access data.

## Author contributions

A.E.T. conceived and designed the study. A.E.T. wrote the manuscript. T.Z., H.T., and A.E.T. performed the statistical and bioinformatic analyses. H.T. and Z.D. helped with bioinformatic analyses. S.B. provided useful feedback.

## Competing interests

The authors declare no competing interests.
