## [Peer Review File · Nature Communications]

An improved epigenetic counter to track mitotic age in normal and precancerous tissuesReviewers' Comments:

Reviewer #1:

Remarks to the Author:

General comments:

Zhu and colleagues develop a new DNA methylation-based mitotic clock called stemTOC to estimate the mitotic age of tissues using data from fetal tissues, in vitro data, blood datasets with available donor age as well as normal and tumor breast tissue. They apply stemTOC to TCGA tumor types as well as normal and pre-cancerous tissues. They also correlate the stemTOC with the predicted cell-of-origin fraction based on DNA methylation deconvolution of tumor samples as well as with other mitotic clocks and with TNSC. Next, they apply stemTOC to other data including sorted neurons, immune cells, buccal swaps and lung tissue samples considering smoking status, and liver from obese vs NAFLD. Finally, they compare in matched samples stemTOC with mutational signatures SBS1 and SBS5.

I read the manuscript with enthusiasm and do find the comparison of multiple mitotic clocks across different conditions of general interest. Nonetheless, I have concerns in some analytical steps and subsequent biological interpretations of stemTOC, which may affect the validity and impact of the conclusions. Conceptually, the authors assume that the total mitotic age of samples is the same as the accumulated stem cell divisions, which is likely not 100% accurate, as these 2 concepts are uncoupled as a function of tissue turnover rates. They develop stemTOC with CpGs correlating with donor's age, which will inevitably lead to capture changes associated with aging and not exclusively with cell divisions. Most importantly, the subsequent (end expected) correlation of stemTOC with age should not be used to justify its accuracy as mitotic clock. Next, the cell-of-origin fraction detected by DNA methylation most likely reflects the tumor purity of cancer samples, and therefore all the subsequent biological interpretations derived in the current manuscript may be wrong. During the validation of stemTOC, the authors correlate it with whole tissue samples during aging, which are composed by different cellular proportions with distinct mitotic ages, most likely leading to confounded associations. They correlate stemTOC with other mitotic clocks as well as SBS1 and SBS5 mutational signatures, and conclude that stemTOC is better than these other mitotic clocks/molecular marks, but the data is not conclusive at all but instead show a similar performance across cancer types, with some instances being more accurate hypo vs hyper clocks, and vice versa. Finally, there are some claims of novelty that are already present in literature, and therefore should be discussed in the present manuscript. Please, find all the specific comments below:

Specific comments:

Major:

1. During the whole manuscript, the authors assume that the total mitotic age of a tissue is equivalent to the accumulated number of stem cell divisions of a particular tissue. However, this assumption is probably not 100% correct, since the total mitotic age of a tissue is composed by the i) the accumulated number of stem cell divisions of the tissue and ii) the cell divisions of their more differentiated progeny. Although these two measurements may be correlated at some point, they are not necessary the same, and will greatly differ in tissues with higher turnover rates. The total mitotic age/accumulated cell divisions of a tissue will be always superior to the number of their stem cell divisions. This is particularly important in tissue types with higher turnover such as colon, blood, etc. This should be clarified and most importantly not assumed to be the same.

2. In the abstract and introduction, the authors state that relationship between somatic mutational and DNA methylation clocks is unexplored and uncharted territory (i.e. "the relation between clock-like

DNAm and mutational signatures remains uncharted territory”), but this is not entirely correct. In 2018, a paper by Zhou et al (PMID: 29610480) showed an anticorrelation with PMD-hypomethylation levels (ie hypomethylation mitotic clock) and somatic mutation burden/density accros cancer types. Afterwards, a study by Duran-Ferrer 2020 (PMID: 34079956) demonstrated how their epiCMIT mitotic clock correlated with SBS1 and SBS5 mitotic-like mutational signatures. Therefore, these previous findings should be considered and discussed in the current manuscript.

3. In the abstract, the authors state: “promising molecular substrate for identifying the tumor cell of origin”, but this is not a novel finding in the current work. Actually, this has been already proposed in a previous work form the same authors (PMID: 35277705). Therefore, this should not be stated as novel finding here.

4. The authors use data from 86 samples encompassing 13 fetal tissue-types. In line 86, they then claim that “Under this model, cell-type specific DNAm differences that may arise at these CpGs in aged tissues is due to differences in the cell-type specific mitotic rates.”.

4.1. This assumption is not demonstrated in this manuscript but based on previous literature, so could you please discuss it in the context of previous reports?

4.2. In addition, I suggest to soften it, as other DNA methylation changes that may accumulate in those CpGs may be related to other less prevalent biological processes. In fact, the authors seem to recognize in the next sentence at line 87 that these CpGs could gain DNA methylation during aging.

5. The authors state at line 79 that they want stemTOC CpGs to be unaffected by chronological age. Accordingly, they use the 6 cell lines to select CpGs which i) gain methylation upon cell passaging (ie doubling population) but ii) do not gain methylation when cells do not cycle, ie avoiding methylation changes upon passage of time as a surrogate for chronological age. Subsequently, the authors use available large cohorts of blood data with age annotation. However, in this case, and contradicting the rationale of previous analyses and their initial goal at line 79, the authors do retain CpGs which are correlated with patient age.

5.1. Please, can you clarify this apparent contradiction?

5.2. This last step will inevitably lead to CpGs that are not exclusively related to mitosis but to chronological age (please also see next point).

6. Related to the previous point, when using the blood datasets, the authors justify to correlate methylation values with donor ages with the claim that “in line with the fact that mitotic age increases with chronological age”. With all due respect, I do not agree with this interpretation. While older tissues are more likely to have been divided more, the mitotic age is not always linked to chronological age and greatly depend on the cell-intrinsic (e.g., cell type, genetic alterations, etc) and extrinsic (e.g., microenvironment stimulus) factors. For instance, post-mitotic cells or long-term HSC will show low mitotic age and high chronological age, while faster dividing cells such as lymphocytes upon antigen recognition will accumulate greater number of cell divisions with few days. Likewise, pediatric tumors will show high mitotic ages. In line with this, the claim that “blood has a relatively high turnover rate and it is only for such tissues that we can expect a correlation between chronological and mitotic age” is not 100% accurate, as lymphocytes which have encountered their cognate antigen (ie memory B/T lymphocytes) will have greater mitotic age compared to naïve lymphocytes regardless of their chronological age. For instance, the epiCMIT mitotic clock for lowly- (naïve) and highly-divided memory B cells does not seem to be correlated with the chronological age, while the chronological Horvath clock does (Duran-Ferrer 2020). The authors could check the value of

stemTOC vs other mitotic clocks in the data of Fig.2 of sorted immune cells considering naïve and memory cells separately in different age ranges (ie young vs elderly).

7. Before benchmarking the stemTOC with other mitotic clocks, it is necessary to understand the relationship among all the mitotic clocks across normal and cancer tissues. I really miss in the analyses a section correlating all the mitotic clocks across cancer types.

7.1. Is there any overlap with CpGs previously used to develop previous mitotic clocks?

7.2. If stemTOC reflects mitotic cell divisions, it should be at some degree correlated with previous mitotic clocks.

7.3. In addition, although the separation of epiCMIT in its underlying hyper- and hypo clocks does make sense to be compared with others hyper and hypo clocks, I would suggest also to incorporate the epiCMIT value as reported originally by the authors.

7.4. Also, what is the correlation of stemTOC with Horvath clock in normal tissues? As mitotic and chronological epigenetic clocks are expected to trace distinct biological processes (ie accumulated cell divisions vs chronological age), these clocks should not show a significant correlation in normal cells?

8. In Fig 2, the authors correlate their stemTOC with chronological age of different cancer and tissue types, and they interpret that this is indicative of stemTOC being an accurate mitotic clock. However, this is not surprising, as they previously selected CpGs which correlated with age in blood samples. Therefore, this is a self-fulfilling prophecy, and should not be used to demonstrate that stemTOC reflects cell divisions. In line with this, the associations in Fig. 2f, 2g and 2h are not surprising and instead expected, as the authors selected for CpGs associated with age, and intrinsic stem cell divisions are known to be accumulated in aging.

9. The differences observed in high vs low turnover tissues (Fig 2b) could be confounded by different tumor purities (ie, sample specimens from highly proliferative tumors will contain probabilistically a higher tumor content, whereas lowly proliferative will contain a mixture of tumor and stromal cells). As this is TCGA data, there should be matched mutational information for the majority of these cases, which could be used (eg, with ABSOLUTE-like methods) to plot the tumor content and interpret the current associations in the context of tumor purity.

10. The authors correlate different mitotic clocks with chronological age in different immune cell samples. However, (and related to point 6) it is known that there is a change in proportions of immune subpopulations during aging, with younger individuals having a higher percentage of B/T naïve cells which have not recognized any antigen and have not suffered any significant clonal expansion (ie lower mitotic age) and elderly donors with higher percentage of B/T memory cells which have clonally expanded upon antigen recognition (ie higher mitotic age). Therefore, if this is not considered, there will be inevitably spurious association of mitotic clocks with chronological age. These analyses should be then performed in homogenous subpopulations of cells (ie naïve vs memory) and not with a bulk sample with all immune cell types together.

11. The authors plot stemTOC vs chronological age in sorted neuros, and claim that there is no correlation (Person=0.21, P-value=0.06). However, there is clear batch effect (ie the cohorts clearly clustered separately) that seems to confound the total correlation. How does the correlations look like within each cohort? Following the same rationale as the previous points, I do expect a correlation with age of the samples, which would suggest that the stemTOC is related strongly linked to methylation changes associated with age.

12. The claim that hyper- mitotic clocks are more accurate than hypo-clocks based solely on their correlation with chronological age in tissues is from my point of view an overinterpretation of the data. As previously mentioned, for tissues whereby there is a high turnover such as blood, chronological age and mitotic age are not expected to correlate. For instance, naïve T/B cells will always show a lower mitotic age compared to memory T/B cells regardless of their chronological age, as the latter have suffered a massive clonal expansion upon antigen recognition. It is therefore expected that this proliferation will lead to higher DNA methylation changes than the comparatively fewer cell divisions of hematopoietic stem cells accumulated during aging. In addition, the data from fig S4 analyze samples which are composed by a mixture of different cell types, which in turn could also change during aging, and therefore will most likely lead to confounding associations (as each cell type has its own DNA methylation profile). Other conflictive data that strongly argues against this claim comes from the study by Endicott et al 2022, which has also been used by the authors to derive the stemTOC. In that study, the authors used pure cell types under controlled settings and traced the number of population doublings, and concluded that RepliTali and epiCMIT mitotic clocks were more accurate in their controlled settings. Finally, human data from Duran-Ferrer et al 2020 show how distinct B cell tumors patients show higher tendencies to gain or loss DNA methylation, suggesting that hyper- or hypomethylation mitotic clocks may be more appropriate in certain situations. This data led to the authors to collapse the epiCMIT hyper- and hypo clocks into its highest value. Therefore, the authors should provide much more (solid) data to make this claim, and discuss it in the context of the existing literature.

13. I have some serious concerns about the biological interpretation of all the data presented in the subheading termed "stemTOC tracks mitotic age of the tumor cell-of-origin", line 177. I present them in the following paragraphs:

13.1. First, I truly do not understand and found counterintuitive the following claim: "we reasoned that tumors with a higher proportion of the cell-of-origin would display a higher mitotic age if mitotic age is a key marker of tumor progression". By definition, a tumor with a higher proportion of the normal cell-of origin cell type, ie normal healthy cells, will have a lower mitotic age compared with tumors samples with a lower proportion of the normal cell-of-origin cells (ie higher proportion of tumor cells), as cancer cells will inevitably have higher number of accumulated cell divisions and therefore a higher mitotic age.

13.2. Second, and related to the previous point, cancer samples do maintain a DNA methylation signature of their cell-of-origin (eg, PMID: 29625048 & PMID: 37582362, among others). Overall, in a tumor sample, the infiltration of the normal cell cell-of-origin is expected to be comparatively minor or even negligible with respect the proportion of tumor cells. In other words, the inferred proportion of the putative cell-of-origin predicted by DNA methylation in tumor samples will likely represent the percentage of tumor cells in the sample, ie tumor purity. This had been shown to be the case in blood tumors (Duran-Ferrer 2020). Therefore, the observed correlation between the cell-of-origin signature and mitotic clocks most likely reflects tumor purity. In solid tumors, this technical artifact may be expected, as tissue samples that have not been purified from tumors with higher proliferation will probabilistically contain a higher proportion of tumor cells. This is critical to be assessed to properly interpret the data. To address this, the authors could use matched genomic data from this TCGA samples and use methods like ABSOLUTE to infer the tumor content through genomic data and correlate it with the identified cell-of-origin proportions.

13.3. The correlations in figS8 of stemTOC and cell-of-origin in cancer and precancerous lesions but not in normal supports the previous point 13.2, ie that the cell-of-origin prediction reflects tumor purity. The data in FigS8 shows that in normal conditions, the proportions of the cell-of-origin normal

subtype that will give rise to the tumor such as luminal cells in breast cancer or cholangiocytes in cholangiocarcinoma, is always lower, consistent with a normal cell composition of a healthy tissue. In premalignant lesions, the malignant cell derived from eg luminal or cholangiocytes starts to proliferate and occupies a higher proportion of cells in the sample (ie higher cell-of-origin prediction), which becomes even higher upon overt cancer. Again, it is imperative to correlate the cell-of-origin prediction with tumor purity prediction from matched genetic data in both pre- and malignant conditions to clarify this issue and properly interpret the data.

13.4. Finally, and following the same line of thought of previous comments, the fact that stemTOC is the mitotic clock with the highest correlation with the cell-of-origin across tumor types (fig. S5C) would be indicative of it being the one mostly affected by distinct tumor cell content. Therefore, this would make stemTOC a less useful mitotic clock in samples with lower tumor content.

14. The authors conclude that “formal comparison of all clocks across the 9 datasets revealed an overall improved performance of stemTOC (SI fig.S7)”. However, the data does not seem to clearly support this conclusion, but instead shows a similar performance of mitotic clocks across cancer types, with hyper- and hypo behaving differently in certain tumor types.

14.1. How the boxplots in Fig S7a look like for all the mitotic clocks? Please, could you represent all the boxplots for all the mitotic clocks?

14.2. From Fig S7B, there are really tiny differences in AUC across mitotic clocks and cancer types, which likely do not represent notable biological effects. In addition, in half (4/9, breast, lung, prostate and esopagous) of the cancers analyzed stemTOC shows a worse AUC value compared to other mitotic clocks (either hypo or hyper).

15. At line 329, there is the following conclusion: Given that in normal-adjacent tissues, stemTOC displays correlations with TNSC without evidence of a saturation effect (Fig.2e), this suggests that stemTOC is a more sensitive tracker of mitotic age than somatic mutations”.

15.1. First, is it possible that the reference to figure panel fig 2e is wrong? There is no TNSC info in the plot.

15.2. Second, could the authors plot the number of SBS1 mutations for adjacent tissues? I expect to see also an increase for SBS1 numbers. This is noted as better performance of stemTOC compared to SBS1 but the authors so not show data on SBS1.

16. At line 332, the authors state: “Of note, a strong correlation with tumor cell of origin fraction is not seen if one were to use the MS1-load as a proxy for mitotic age (SI fig.S12), and correlations of MS1-load with CPE-based tumor purity estimates were weaker than with EpiSCORE-derived tumor cell-of-origin fractions (SI fig.S13)”.

16.1. There is evidence that from hundreds to thousand mutations associated to SBS1 are not only accumulated in cancer genomes, but also in normal healthy cells during aging. Therefore, both analyses showed at fig S12 and S13 should consider the age of the patient at sampling.

16.2. Next, the authors conclude that “Overall, these data indicate broad agreement between DNAm and somatic mutational, but with DNAm being a more sensitive marker of mitotic age”. Sorry but I cannot see the evidence from the previous data, could you please explain this a little bit more?

Minor:

1. An initial table/fig to show whole study design would improve clarity
2. In Fig1, in the selection of CpGs using cell culture, it seems that the green line is the color that represents the selected CpGs (upper panel). However, in the bottom panel, could it be that the green color represents the excluded CpGs...? To clarify this, I would emphasize with a small legend what the different color code means (ie selected vs excluded CpGs), and would also avoid to use colors from the previous panel (ie, pink as in muscle group, to avoid possible misleading interpretations).
3. Please, could you indicate a measure of effect size in Fig 4B? Also, indicate always the p-value, although not being significant.
4. It would be of interest to include the value of other mitotic clocks in all the supplementary tables together with the stemTOC score.

Reviewer #2:

Remarks to the Author:

As with fishing expeditions, gear, bait, and method of capture are essential. Fishing for 'mitotic CpGs' that reflect the overall number stem cell divisions in various tissues (normal, pre-cancer, cancer) is an ambitious goal, out in the open sea of the DNA methylome. The study is well justified and the authors go about it with great effort analyzing a large number of relevant data sets. The rationale given is clear. Since mitotic age has been shown to be generally an important risk factor for cancer (as demonstrated e.g. by Tomasetti and Vogelstein), a quantitative assessment of mitotic tissue age, as proposed by the authors, offers a new paradigm/biomarker for evaluating the neoplastic potential of a tissue in an individual because replication errors (in DNA sequence or in DNA methylation imprints) occur during DNA replication and contribute to cancer.

Complementing purely statistical approaches that have no biological underpinnings (e.g. Hannum's and Horvath's elastic nets), the authors offer a biological approach to identify a new clock based on DNAm changes, starting with essentially hypomethylated CpG sites in fetal tissues. Theoretical explanations of why stochastic gain in DNA methylation during DNA replication may occur, have been given by others (Genereux et al, PNAS 2005; Sontag et al, JTB 2006). This process is inherently stochastic. What exactly the authors mean by 'quasi-stochastic' (line 118) is not clear as the molecular machinery that maintains methylated and/or unmethylated CpGs is inherently error-prone upon DNA replication, and the resulting drift is naturally stochastic along a given cell lineage involving 100s if not 1000s of cell divisions. It is hard to make sense of the sentence between lines 118-120. For any given subject, using the bead arrays, DNAm is measured in bulk across many cells and the observed drift (change in methylation) at any given CpG ought to reflect the mitotic age for all CpGs in the selected clock-CpG set, not just in random subsets. It might help to clarify what the authors mean by 'quasi-stochastic' and their reference to random subsets.

Similar, for clarity, it may help readers to better understand what the authors mean when they refer to "hypomethylation" vs "hypermethylation", since many readers may confuse the state of a group of CpGs (e.g. CpG island as being lowly (hypo) methylated vs highly (hyper) methylated) with the process of gradual loss vs gain of DNA methylation in clonal cell populations.

An interesting finding is the finding that the stemTOC correlates well with the tumor stem cell-of-origin fraction. Maybe this is obvious, but this reader is somewhat puzzled by why this should be so. Naively one might expect that in any tumor, the fraction of the cell-of-origin is approximately constant and not

related to size or time of the clonal expansion (sojourn time). What exactly is expected to change in the cell-type composition of the tumor so that the fraction of the cell-of-origin increases with mitotic age? Would the authors care to elaborate?

Finally, upon publication, a list of the 371 stemTOC CpGs and their annotations should be provided in the SI.

Some minor points:

Line 147: It would be helpful to also reference the clocks used fig.S4 in the Supplemental Material.

Line 249: "Of note ..." Notwithstanding a larger effect size, the rationale given that the "upper quantile" (the authors mean the 95% percentile) somehow minimizes the underlying stochasticity of DNAm patterns leading to better mitotic age estimates, is suspect. It is unclear whether or not using the interquartile range (i.e. removing extremes) would not yield better correlations.

Line 320: Fig 2e) shows neurons. ??

Line 331: "Of note ...". It appears that in this comparison with MS1, stemTOC was adjusted for tumor cell-of-origin while MS1 load is not. Is a fair comparison even possible?

Heatmaps in fig 1b: How are the 629 mitCpGs (in columns) ordered? Also, there is no description of the adjacent curves. What do they mean?

Reviewer #3:

Remarks to the Author:

Zhu et al. presented an epigenetic counter of mitotic age and applied it to cancer and pre-cancerous tissues. The new clock is qualitatively similar to the previous mitotic clocks from the same authors. The result interpretation on the cell of origin tracking seems misleading. The presented data only suggests that stemTOC associates tumor purity, which can be achieved by any tumor-specific measures. Other comments are provided below:

Is stemTOC a real clock? Despite the claim, stemTOC readout does not give mitotic age. stemTOC gives a numeric from 0 to 1. It perhaps correlates with mitotic age (as many other chronological clocks would also do). But the actual projection to the mitotic age is missing as all evidence supporting the "mitotic" nature is provided indirectly. This is in contrast with the chronological clocks (e.g, the Horvath clock) which does predict time in year.

Overall, the model assumed constant methylation change over mitosis, among cell lineages and developmental stage. No evidence is provided to support this key assumption. In fact, constant methylation change over mitosis is unlikely and would be affected by the turnover rate, different biochemical environments (DNMTs/TETs) etc.

The authors claimed that the ticking of the clock is only due to division of stem cells (in contrast to a combined effect including expansions?). But no direct evidence is given to support this claim (hence the name stemTOC). Only a heavily confounded analysis (see comment below) of life-time stem cell division rate was used to show correlation with tumor stemTOC.

The 317 vivo-mitCpGs was unmethylated in 86 fetal samples of different tissues but defined for

methylation gain based solely on blood data. Whether these CG methylation tracks mitotic age in other cell types remain unvalidated. There is also a significant difference between fetal and postnatal tissues for some lineages.

In Figure 3b, the cell of origin correspondence is not relevant here since one can replace the X-axis label of all the panels by simply tumor purity (as estimated by EpiSCORE). To really show stemTOC tracking tumor cell of origin (if it does), one should show how one can distinguish tumors of different cells of origin purely based on stemTOC readouts. This analysis is also significantly confounded by the mitotic age of other cell types in the TCGA samples.

The authors claim that stemTOC addressed the issue of CTH. But from the presented data, stemTOC does not nominate mitotic age of each cell components. Instead, for most of the presented analysis, the authors merely estimate the cell composition (the finding of which is known) and mitotic age of the bulk tissue separately. This is the case for the smoking and NAFLD studies in Fig 4.

Fig 5e: This analysis is founded on rough (and simplistic) estimates of annual stem cell division rates from prior studies. It is known that stem cells do not divide at a constant rate throughout the life span. Within each cell type/cancer type, very little correlation is visible, suggesting mostly a cell type-specific effect. Further, any age-associated metric would manifest an association as age is explicitly formulated in the X-axis. Minor: Why were only a subset of TCGA cancers presented? PCC is also missing.

Detailed Response to Reviewer Points:

Reviewer #1 (expertise in DNA methylation and cancer risk prediction):

General comments:

Comment-1: I read the manuscript with enthusiasm and do find the comparison of multiple mitotic clocks across different conditions of general interest.

Response: We would like to thank the Reviewer for taking time to evaluate our MS, and for acknowledging his/her enthusiasm for this topic.

Comment-2: Nonetheless, I have concerns in some analytical steps and subsequent biological interpretations of stemTOC, which may affect the validity and impact of the conclusions. Conceptually, the authors assume that the total mitotic age of samples is the same as the accumulated stem cell divisions, which is likely not 100% accurate, as these 2 concepts are uncoupled as a function of tissue turnover rates.

Response: We sincerely thank the reviewer for raising this point. We apologize for being imprecise in our terminology which may have caused confusion, as indeed we had used the term “turnover-rate” to also refer to the intrinsic rate of stem-cell division of a tissue. We agree with the reviewer that it usually refers to the cell-divisions of expanding progenitor populations, and hence, we fully agree that stemTOC is only measuring a proxy to the cumulative number of stem-cell divisions, as indeed temporary progenitor expansions due to tissue turnover will also inevitably contribute to the stemTOC estimate. On the other hand, progenitors and differentiated cells alike will inherit most of the DNAm changes that accumulate in the underlying stem-cell pool. From a practical perspective, specially in relation to cancer risk prediction, it would be better if a mitotic clock could estimate the total mitotic age of the sample, as opposed to just the number of stem-cell divisions, since the total mitotic age would also capture DNAm changes in precancerous subclonal expansions that are potentially informative of cancer-risk. In response to the reviewer’s excellent point we have now clarified that stemTOC is a proxy for total mitotic age, meaning that includes the contributions not only from the cell-divisions that the stem-cell pool has undergone but also the cell-divisions of temporary amplifying progenitor populations. In line with this, we now explicitly define total mitotic age in the Introduction, and use the term “total mitotic age” more often in the Introduction as well as the beginning of the Results section to make this point clear. Later on, we interchangeably use the terms mitotic age and total mitotic age to mean the same quantity. In revising our MS, we are now also more careful when using the term “tissue-turnover” and “turnover-rate”, and only use it in connection to events that are uncoupled to the intrinsic rate of stem-cell division.

Comment-3: They develop stemTOC with CpGs correlating with donor's age, which will inevitably lead to capture changes associated with aging and not exclusively with cell divisions. Most importantly, the subsequent (end expected) correlation of stemTOC with age should not be used to justify its accuracy as mitotic clock.

Response: We appreciate the reviewer's point. We worry that the reviewer may not have understood the exact procedure by which stemTOC was built. The CpG selection procedure in stemTOC imposes three separate requirements and all three have to be met: (R1) CpGs must be unmethylated in many different fetal tissue types (as well as cord-blood), thus defining a ground-state of ultra-low methylation across many different cell-types, (R2) CpGs must display increased DNAm with population doublings (PDs) in 6 separate cell-lines (representing fibroblasts, endothelial and smooth muscle) and not display such increases when cell-lines are treated with cell-cycle inhibitors or when using low serum levels, (R3) CpGs must display increased DNAm with age in real whole blood cohorts. Thus, it should be clear that we are **not** selecting CpGs purely by correlation with donor's age (R3), because R2 requires the same CpGs to display increased DNAm with PDs in 6 separate cell-lines. Moreover, the cell-cycle inhibitors and low-serum experiments are used to dissect DNAm changes that occur with cell-division from those that occur with passage of time (age). Requirement R2 is critical to resolve the reviewer's concern and indeed, we note that this cell-line data was published by Endicott et al Nat Commun 2022, precisely to deconvolve the effects of aging and cell-division. In response to the Reviewer's point, we have now improved Fig.1a-b to make it clearer that we are using a feature selection procedure that removes as much as it is possible the confounding effect of chronological age. This new version of Fig.1a-b is depicted below for the Reviewer's convenience:

Construction of the stochastic epigenetic mitotic timer of cancer (stemTOC)

New Figure-1a-b: Construction of stemTOC and estimation of mitotic age. a) We first identify CpGs (n=30,257) mapping to within 200bp upstream of the TSS of genes that are unmethylated (defined by DNAm beta-value < 0.2) across 86 fetal-tissue samples from 13 different fetal-tissues (including post-natal cord-blood tissue). These are then filtered further by the requirements that they display hypermethylation as a function of population doublings (PDs) in 6 cell-lines representing fibroblasts, endothelial and smooth muscle cell-types. To avoid confounding by passage of time/chronological age, we also demand that they don't display such hypermethylation

when cell-lines are deprived from growth-promoting serum or when treated with mitomycin (MMC, a cell-cycle inhibitor), resulting in 629 “vitro-mitCpGs” satisfying these conditions. **b)** To exclude cell-culture specific effects, we then ask if DNAm at each of these 629 CpGs displays significant hypermethylation with chronological age, as assessed in 3 separate whole blood cohorts, after adjusting for variations in 12 immune-cell type fractions. *Because blood has a high rate of stem-cell divisions, correlations with chronological age will capture the subset of vitro-mitCpGs that also change with cell-division in-vivo. In the heatmaps, rows label samples, which have been ordered according to increasing age, so that older samples appear towards the bottom of the heatmap. Columns label the CpGs which have been ordered according to hierarchical clustering. This meta-analysis over the 3 separate whole blood cohorts results in 371 “vivo-mitCpGs”.*

The reviewer then raises the subsequent concern that we can't validate a mitotic-counter against chronological age. In response to this, let us first note that it is extremely difficult (if not currently impossible) to validate a mitotic counter in *in-vivo* human tissue data. Thus, using chronological age as a surrogate for mitotic age is one of the few options available. Whilst we agree that for a general tissue we can't use chronological age as a surrogate for mitotic age, it is nevertheless true that in tissues where the stem-cell division rate is relatively high (e.g. colon, rectum, blood,...) that total mitotic age will very likely correlate with chronological age. In contrast, in postmitotic tissues (e.g sorted neurons), or in tissues with a relatively low stem-cell division rate that undergo significant dynamic turnover due to e.g. hormonal factors (e.g. breast, ovary, endometrium), we would not expect there to be a strong correlation. Thus, the purpose of Fig.2a-e and related SI figs.S5-S6 is to demonstrate how the correlation strength with chronological age varies as a function of a tissue's stem-cell-division rate and tissue-turnover due to other factors e.g. hormonal factors. Unfortunately, for most tissue-types, quantitative precise estimates of stem-cell division and turnover rates are still unknown (see e.g. Sender R et al Nat Med 2021 for the most recent update) and for some tissues we also don't have sufficient numbers of normal samples to confidently estimate correlation strengths, so that a more detailed quantitative analysis of this is not possible. Nevertheless, our current Fig.2a-e and SI figs.S5-S6 make a strong qualitative statement in support of stemTOC measuring a total mitotic age.

Comment-4: Next, the cell-of-origin fraction detected by DNA methylation most likely reflects the tumor purity of cancer samples, and therefore all the subsequent biological interpretations derived in the current manuscript may be wrong.

Response: We thank the reviewer for raising this point. The cell-type fractions are being estimated using a very recent DNAm-atlas that we built using the EpiSCORE procedure (see Zhu T et al Nat Methods 2022). As such, we agree with the reviewer that the estimated fraction of the tumor cell of origin in a tumor sample will give us an estimate of tumor purity. However, they are not equivalent and in many ways the EpiSCORE-based estimate improves upon more traditional tumor purity estimates: for instance, with EpiSCORE we can easily discriminate tumor subtypes from each other, say liver hepatocellular carcinoma (originating from hepatocytes) from liver cholangiocarcinoma (originating from cholangiocytes) (see Zhu T et al Nat Methods 2022). A more traditional tumor purity measure such as

ABSOLUTE (CNV-based) would not be able to discriminate these tumor subtypes. Thus, in this regard, our EpiSCORE cell-type fractions supersede tumor purity estimates obtained with other methods, so whilst we agree that the x-axis in Fig.3b-c could have been labelled with “tumor purity”, it is important to note that that they are not formally equivalent.

More importantly though, we do not understand why the reviewer then implies that “all subsequent biological interpretations may be wrong”? Quite the opposite in fact, and indeed we fear that the reviewer may have misunderstood the meaning and significance of the results displayed in Fig.3 and Fig.4. In fact, what we demonstrate in Fig.3 is a very clear correlation between the mitotic age of a tumor sample and the estimated tumor cell of origin fraction, or, given the clonality of a tumor cell population, we can call this the “tumor-cell fraction”, or alternatively, if the reviewer prefers, we can call it “tumor purity”. The bottom line here though is that, biologically, these results makes a lot of sense: the tumor cells are proliferating at a much higher rate than the endothelial cells, fibroblasts and to some extent also immune-cells, and will thus have generally speaking a higher mitotic age than the surrounding stroma and other normal cells. Thus, the correlation we observe in each of 15 TCGA cancer-types is entirely expected on biological grounds.

The biological and clinical significance of having demonstrated this result in human in-vivo data is that we could in principle use such a DNAm-based mitotic counter to track subtle increases in the tumor cell of origin fraction in tissues that are not yet neoplastic, as shown in Fig.4b. For instance, we show how stemTOC increases with the squamous epithelial fraction in normal buccal swabs, and how the mitotic age proxy is also increased in smokers vs non-smokers. This is an entirely unique and novel observation, which offers new strategies to monitor mitotic-age non-invasively in a tissue (squamous epithelium) that is representative of the cell-of-origin of smoking-related cancer types such as oral, lung and esophageal squamous cell carcinomas.

In response to the reviewer’s point, we have now clarified in the Results section that the tumor cell of origin fraction is in effect an estimate of tumor purity, and indeed we have added a new panel (new SI fig.S10C) to new SI fig.S10 to demonstrate the results for different tumor purity estimation methods. We display this panel in our more detailed response to a similar reviewer’s point further below. In Discussion we now also emphasize more the biological and potential clinical significance of the findings presented in Fig.4, as described above.

Comment-5: During the validation of stemTOC, the authors correlate it with whole tissue samples during aging, which are composed by different cellular proportions with distinct mitotic ages, most likely leading to confounded associations.

Response: We thank the reviewer for raising this excellent point. In response to this, let us begin by alerting the reviewer to the fact that in Fig.2c-d we correlated mitotic age estimates to chronological age in purified i.e. sorted immune-cell subsets. For the reviewer’s convenience, we redisplay the relevant figure Fig.2c-d below:

Figure 2c-d legend: **c)** Scatterplots of stemTOC's mitotic age (y-axis) vs chronological age (x-axis) in the three sorted immune-cell populations (MO=monocytes, NEU=neutrophils, naiveCD4T= naïve CD4+ T-cells) from BLUEPRINT (BP). The number of samples N is given above plot. The Pearson Correlation Coefficient (PCC) and P-value from a linear regression is given. **d)** Heatmap of PCC values between mitotic and chronological age for 7 mitotic clocks across all sorted immune-cell populations. Sorted immune-cell populations are labeled by cell-type and study it derives from

As the reviewer can see, stemTOC's mitotic age correlates with chronological age in every sorted immune-cell dataset, including the naïve CD4+ T-cell subset from the BLUEPRINT consortium (Chen & Soranzo Cell 2016), clearly suggesting that CTH is not driving these associations. Indeed, as already remarked in our previous manuscripts (specially in Teschendorff et al Genome Med 2020), the CpGs that make up clocks like epiTOC1, epiTOC2 and now also stemTOC, are not cell-type specific markers, and therefore they are not subject to confounding by age-associated variations in immune-cell fractions. As shown in our previous Genome Med 2020 paper, this is however not true for the HypoClock CpGs, many of which are immune-cell specific. To further explore this, we have now evaluated all 7 mitotic clocks in 18 large whole blood cohorts, encompassing close to 15,000 samples, correlating mitotic age to chronological age before and after adjusting for variations in 12 immune-cell fractions using our recently published 12 immune-cell type DNAm reference matrix to estimate these 12 immune-cell fractions (Quo L et al Genome Med 2023). If the reviewer's concern is justified, adjusting for the 12 immune-cell fractions should result in many associations between mitotic-age and chronological age becoming non-significant as a result of age-associated variations in immune-cell fractions and differences in mitotic age between immune cell-types. However, the results as shown below are unequivocal in demonstrating that for all hypermethylated clocks (ie epiTOC1/2, stemTOC, epiCMIT-hyper) and RepliTali, the strong associations with chronological age remain highly significant even after adjustment for cell-type fractions (CTF):

On the other hand, for HypoClock and epiCMT-hypo we observe how adjustment for CTH leads to a reduction in significance levels. This is consistent with our previous observation (Teschendorff Genome Med 2020) that HypoClock CpGs are immune-cell specific and hence confounded by age-associated variations in immune-cell fractions (e.g. the age-associated shift from naïve to mature T-cell subsets being one clear example but there are others -see Quo L et al Genome Med 2023). Hence, adjusting for these fractions would lead to a reduction in the associations with age, as seen above.

Of note, the advantage of using whole blood to carry out the previous analysis is that for this tissue we can accurately estimate fractions for 12 immune-cell types. For other tissues, we can approximately estimate underlying cell-type fractions using our EpiSCORE DNAm-atlas. Hence, to further demonstrate that putative age-associated changes in cell-type fractions are not driving the correlations of stemTOC's mitotic age with chronological age, we have applied all 7 mitotic clocks to 5 of the eGTEx normal-tissue datasets for which we can estimate underlying cell-type fractions. Once again, results are conclusive in demonstrating that as far as the hypermethylated clocks are concerned, the associations

(significant or non-significant) with chronological age are unaltered upon adjustment for CTH, as shown below:

In response to the reviewer's point, we have now included the two above figures as new Supplementary Figures, SI fig.S7 and SI fig.S8.

Finally, we acknowledge that the mitotic age estimate in a bulk sample is an average over the mitotic ages of the underlying cell-types. Assuming that underlying cell-type fractions were changing with age and that the underlying cell-type specific differences in mitotic age were driving the associations of total mitotic age with chronological age, then clearly the adjustment for cell-type fractions would lead to big changes in the significance of the associations, which is not what we observe based on the above figures, as well as the results obtained in sorted immune cell populations. Thus, all our data and results unequivocally converge in demonstrating that the association of hypermethylated clocks with chronological age is not confounded by cell-type heterogeneity.

Comment-6: They correlate stemTOC with other mitotic clocks as well as SBS1 and SBS5 mutational signatures, and conclude that stemTOC is better than these other mitotic clocks/molecular marks, but the data is not conclusive at all but instead show a similar performance across cancer types, with some instances being more accurate hypo vs hyper clocks, and vice versa.

Response: We thank the reviewer for raising this point. As far as the comparison of stemTOC to the other epigenetic mitotic clocks is concerned, it is clear from Fig.2h, Fig.3e, SI figs.S5c-S6c and SI fig.S10d that clocks based on hypermethylated CpGs outperform those based on hypomethylated CpGs, and for several panels we have backed up our claim with formal statistics. For the reviewer's convenience we have merged together the most relevant panels, and display them together below:

For the 3 heatmap figures above displaying paired Wilcoxon rank sum test P-values, comparing each of the 7 mitotic clocks to each other, the convention is that a significant P-value means that the counter labelling the row displays stronger correlations than the counter labelled by the column. Thus, overall, the evidence that clocks based on hypomethylated CpGs (specially HypoClock and epiCMT-hypo) perform worse than hypermethylated ones is strong in the context of (1) correlations with independent stem-cell division rates in normal tissue (Fig.2h), (2) correlations with tumor cell of origin fraction (Fig.3e), correlations with chronological age in normal tissues with high or moderate stem-cell division rates (SI Fig.S5c & SI Fig.S6c), as well as in predicting tumor cell of origin (new SuppFig.10d). The heatmaps shown in Fig.3d, SuppFig.5b, SuppFig.6b further attest that the improved performance of hypermethylated clocks is robust across studies, so we are puzzled by the reviewer's comment. Perhaps the reviewer was referring to cancer vs normal comparisons, where hypermethylated and hypomethylated clocks perform similarly, as expected, because loss of methylation due to incomplete DNAm maintenance is more operative in the context of highly proliferating cells (e.g. cancer cells, see Teschendorff et al Genome Med 2020). As far as the comparison of stemTOC to the other hypermethylated mitotic clocks is concerned, we agree that the performance is only marginally better. Whilst this may seem disappointing, the benchmarking or comparison to existing hypermethylated clocks is not an essential or critical part of this MS. The global comparison of hypermethylated vs hypomethylated clocks does however push boundaries in the field, clearly demonstrating for the first time, that hypermethylated clocks perform better in the context of cancer risk prediction.

As far as the comparison to somatic mutational signatures is concerned, the data we present in Fig.5c & 5f (which we note is for cancer-tissue), suggests that MS1 load is a linear function of a tissue's stem-cell division rate, whereas stemTOC displays a non-linear trend with evidence of a saturation effect.

Bearing in mind that this plot is for cancer-tissues, which represent highly proliferative cell populations, the saturation effect suggests that stemTOC is a more “sensitive” marker of mitotic age than MS1 load. Indeed, in normal tissue, stemTOC displays a linear trend with no evidence of a saturation effect. Overall, we agree with the reviewer that a more rigorous comparison between DNAm and somatic mutations would require fairly large collections of normal multi-tissue samples with matched somatic mutation and DNAm profiles encompassing a wide age-range, yet such data is not presently available. In relation to this, it is worth noting that detecting DNAm changes in normal tissue is easier than somatic mutations (see e.g. Abascal F et al Nature 2021), which already points to clear advantages of using DNAm. Finally, the increased sensitivity of DNAm over somatic mutations to measure mitotic age should not be surprising at all, given that the rate of epimutations is known to be about 10-100 times higher than that of somatic mutations (see e.g. Horsthemke et al <https://pubmed.ncbi.nlm.nih.gov/16909906/>). In response to the reviewer’s point, we have now clarified these points in Discussion.

Comment-7: Finally, there are some claims of novelty that are already present in literature, and therefore should be discussed in the present manuscript.

Response: We thank the reviewer for drawing our attention to this and sincerely apologize for the omissions. Indeed, the studies by Zhou et al and Duran-Ferrer et al did explore correlations with somatic mutations, although the study by Zhou et al did not explicitly consider the mutational signatures from Alexandrov et al, and the study by Duran-Ferrer only explored this in the context of B-cell malignancies. In response to the reviewer’s point, we have now revised our statements in the Introduction, citing these two papers, and also briefly explaining in what way our results extend their previous analyses.

Major points:

Comment 1. During the whole manuscript, the authors assume that the total mitotic age of a tissue is equivalent to the accumulated number of stem cell divisions of a particular tissue. However, this assumption is probably not 100% correct, since the total mitotic age of a tissue is composed by the i) the accumulated number of stem cell divisions of the tissue and ii) the cell divisions of their more differentiated progeny. Although these two measurements may be correlated at some point, they are not necessary the same, and will greatly differ in tissues with higher turnover rates. The total mitotic age/accumulated cell divisions of a tissue will be always superior to the number of their stem cell divisions. This is particularly important in tissue types with higher turnover such as colon, blood, etc. This should be clarified and most importantly not assumed to be the same.

Response: This is effectively identical to General Comment-2 above. As such, we kindly refer the reviewer to our response given previously.

Comment 2. In the abstract and introduction, the authors state that relationship between somatic mutational and DNA methylation clocks is unexplored and uncharted territory (i.e. “the relation between clock-like DNAm and mutational signatures remains uncharted territory”), but this is not entirely correct. In 2018, a paper by Zhou et al (PMID: 29610480) showed an anticorrelation with PMD-hypomethylation levels (ie hypomethylation mitotic clock) and somatic mutation burden/density across cancer types. Afterwards, a study by Duran-Ferrer 2020 (PMID: 34079956) demonstrated how their epiCMIT mitotic clock correlated with SBS1 and SBS5 mitotic-like mutational signatures. Therefore, these previous findings should be considered and discussed in the current manuscript.

Response: We thank the reviewer for drawing our attention to this, and sincerely apologize for the omissions. Indeed, these two studies did explore correlations with somatic mutations, although the study by Zhou et al did not explicitly consider the mutational signatures from Alexandrov et al, and the study by Duran-Ferrer only explored this in the context of B-cell malignancies. In response to the reviewer’s point, we have now revised our statements in the Introduction, citing these two papers, and also briefly explaining in what way our results extend their previous analyses.

Comment 3. In the abstract, the authors state: “promising molecular substrate for identifying the tumor cell of origin”, but this is not a novel finding in the current work. Actually, this has been already proposed in a previous work from the same authors (PMID: 35277705). Therefore, this should not be stated as novel finding here.

Response: We thank the reviewer for drawing our attention to this. We agree that we were not being precise in the Abstract (probably due to the 150-word limit). As a result, the last sentence in the Abstract had lost its intended meaning. In response to this, we have now rewritten the last sentence of the abstract to the following: “Our data suggests that stemTOC is a more sensitive proxy of mitotic age than somatic mutational signatures, pointing towards DNAm as a more promising molecular substrate for detecting mitotic-age increases in the tumor cell of origin of precancerous lesions and for developing cancer-risk prediction strategies.” This revision now makes it clear that we meant “mitotic age of the tumor cell of origin”, not “identifying tumor cell of origin”.

Comment 4.1: The authors use data from 86 samples encompassing 13 fetal tissue-types. In line 86, they then claim that “Under this model, cell-type specific DNAm differences that may arise at these CpGs in aged tissues is due to differences in the cell-type specific mitotic rates.” This assumption is not demonstrated in this manuscript but based on previous literature, so could you please discuss it in the context of previous reports?

Response: We thank the reviewer for raising this important point. The reviewer is right that we are basing this assertion on previous manuscripts, notably the paper by Nejman D & Cedar H Cancer Res

2014, as well as our previous paper in Genome Med 2020. In a nutshell, the two key points are as follows: (1) by construction, our CpGs are not cell-type specific in the fetal ground-state and retain ultra-low DNAm in the neonatal state: we have now added a new SI fig.S1 to demonstrate the latter. (2) consequently, any differences in DNAm that these CpGs may display between adult cell-types, could be due to either these CpGs being involved in differentiation and thus becoming cell-type specific markers, or due to differences in the lifetime mitotic rates of these cell-types. From having analyzed large numbers of DNAm datasets, or for instance by inspecting the recent WGBS DNAm-atlas from Loyfer et al Nature 2023, it should be clear that CpGs involved in differentiation and which define cell-type specific markers would display rather big differences in DNAm between cell-types, typically at least >50% (or >0.5 in a beta-value scale) if we are comparing purified/sorted cell-types. In contrast, if the differences in DNAm that arise between adult cell-types is more modest, then these are more likely to be the result of other processes, of which differences in mitotic-age are a clear candidate, specially if the underlying CpGs have been chosen to change with cell-division in culture and if they also change with chronological age in a tissue with a high underlying stem-cell division rate (so that mitotic and chronological age will be correlated). Thus, one way to demonstrate that stemTOC CpGs are depleted for cell-type specific markers in adult tissues is to compare their average DNAm levels between adult age-matched sorted cell-types or tissue types. As a benchmark we compare the stemTOC CpGs to those of the original HypoClock, since previously we already showed that the latter contain a fraction of cell-type specific markers. The comparison is performed on the a) sorted age-matched immune-cells from BLUEPRINT (Chen & Soranzo Cell 2016), b) a separate study of age-matched sorted immune cell-types (Paul D et al Nat Commun 2016) and c) the large age-matched multi-tissue DNAm dataset of eGTEx (Oliva M et al Nat Genet 2022), and the results are depicted below:

The figure above depicts the fraction of HypoClock (purple) and stemTOC (orange) CpGs that display absolute differences in DNAm between immune cell/tissue-types greater than specific thresholds. Focusing on the immune-cell subsets (panels a+b), we can see that the fraction of stemTOC CpGs that display >0.5 DNAm differences is effectively zero, and that this fraction is clearly much higher for HypoClock. Similar patterns are observed across the multi-tissue eGTEx datasets (panel-c) and by focusing on less stringent thresholds (e.g. > 0.25 , because these are mixtures). Of note, the only tissue displaying some notable differences for stemTOC is colon, the most proliferative tissue. We note that the age-matched nature of all these tissues and cell-types is really important as this adjusts for the confounding effect of chronological age. So, clearly what this figure above therefore demonstrates is that our stemTOC CpGs are not cell-type specific, and that the small differences seen between colon and other age-matched tissues is due to the much higher stem-cell division rate of colon. Cell-type specific markers would be displaying >0.5 DNAm changes between cell-types and >0.25 changes between tissue-types (the mixed nature of tissues means that absolute differences in DNAm would be diluted down).

To further demonstrate this, we repeated the above analysis for 8 sorted non-immune cell-types (hepatocytes, adipocytes, colon-epithelial, lung epithelial, neurons, pancreas endocrine,

pancreas_exocrine, endothelial) from the Moss et al DNAm atlas (Illumina 450k, Moss J et al Nat Commun 2018), and once again results are unequivocal in demonstrating that stemTOC CpGs are not cell-type specific, and that only when comparing colon (the most proliferative tissue) to other much less proliferative tissues, that we see some smaller differences, as expected:

Thus, we conclude from all of this that our stemTOC CpGs are not cell-type specific markers of adult tissues or cell-types. In response to the reviewer’s point, we have now added the above two figures as new SI figs.S2-S3.

Comment 4.2. In addition, I suggest to soften it, as other DNA methylation changes that may accumulate in those CpGs may be related to other less prevalent biological processes. In fact, the authors seem to recognize in the next sentence at line 87 that these CpGs could gain DNA methylation during aging.

Response: To clarify, on old lines ~87 we were describing our stemTOC CpG feature selection procedure. So, in effect we are trying to find CpGs that gain DNAm with population doublings in 6 different cell-lines, but that they do not display gains when these cell-lines are treated with a cell-cycle inhibitor or if deprived from growth-promoting serum: hence these CpGs would be changing primarily due to in-vitro cell-divisions. Later, when we also demand that these CpGs gain DNAm in blood with

chronological age, after adjusting for variations of 12 immune-cell type fractions, we are effectively enriching for CpGs that also gain DNAm with cell-divisions *in-vivo*, because blood stem-cells divide at a fairly high rate, which means that chronological age is a reasonable surrogate for mitotic age, specially given that the CpGs have already been selected to gain DNAm with cell-divisions *in-vitro*. Thus, we think that this should now resolve the reviewer's concern, specially given the additional evidence that we have now provided in the response to the previous point (new SI fig.S2-S3).

Comment 5.1 The authors state at line 79 that they want stemTOC CpGs to be unaffected by chronological age. Accordingly, they use the 6 cell lines to select CpGs which i) gain methylation upon cell passaging (ie doubling population) but ii) do not gain methylation when cells do not cycle, ie avoiding methylation changes upon passage of time as a surrogate for chronological age. Subsequently, the authors use available large cohorts of blood data with age annotation. However, in this case, and contradicting the rationale of previous analyses and their initial goal at line 79, the authors do retain CpGs which are correlated with patient age. Please, can you clarify this apparent contradiction?

Response: We opine that there is no contradiction. To clarify, we are filtering an initially large pool of TSS200 CpGs (n=30,257) that are unmethylated across each one of 86 fetal tissue + neonatal samples, by demanding that they should also gain DNAm with population doublings in each of 6 different cell-lines and that they do not gain DNAm when the cell-lines are treated with a cell-cycle inhibitor (which effectively enriches for CpGs that gain DNAm with cell-division *in-vitro* and NOT with passage of time/chronological age), all of this resulting in a much smaller subset of 629 CpGs. In other words, of the 30,257 CpGs, only 629 remain. Because cell-lines represent an *in-vitro* system, and we have grounds to suspect that DNAm changes associated with cell-division *in-vitro* may be different to those associated with cell-division *in-vivo*, we further require that the 629 CpGs should gain DNAm with chronological age in a tissue like blood which has a high stem-cell division rate (considering both LT and ST-HSCs), which thus reasonably allows us to use chronological age as a correlative surrogate for mitotic age. Of the 629 CpGs, a total of 371 survive. Thus, the 371 stemTOC CpGs (i) gain DNAm with cell-divisions *in-vitro*, (ii) do not gain DNAm with chronological age *in-vitro*, AND (iii) gain DNAm with chronological age *in-vivo* in a tissue where mitotic and chronological age should be correlated. Overall, this procedure should hence enrich for CpGs that gain DNAm with cell-divisions *in-vivo*.

Perhaps it would help further if we make it clear in which way our stemTOC construction differs from RepliTali (Endicott et al Nat Commun 2022), which is the paper presenting the cell-line data. The RepliTali clock CpGs are effectively selected using only the cell-line data, without a further filtering step based on *in-vivo* normal tissue data. In contrast, stemTOC uses an additional stringent requirement that the CpGs changing with cell-division *in-vitro*, should also change with chronological age *in-vivo*. So, in effect, there is no contradiction at all. We are in effect using the *in-vitro* and *in-vivo* data together to demand that our CpGs change simultaneously with cell-division *in-vitro* and with chronological age *in-vivo*. We are taking the intersection, not the union.

We have now improved the presentation of Fig.1a-b as well as improving the legend to Fig.1a-b (as shown in one of our previous responses) in the hope that this clarifies the stemTOC feature selection step.

Comment 5.2. This last step will inevitably lead to CpGs that are not exclusively related to mitosis but to chronological age (please also see next point).

Response: If we were taking the union of CpGs selected from the in-vivo and in-vitro datasets, we would agree with the reviewer, but we are **not** taking the union. Once again, the 371 stemTOC CpGs are a subset of the 629 CpGs that gain DNAm with cell-division in-vitro, hence it is more likely that the gains they display in the in-vivo data are due to increased mitotic age (which correlates with chronological age) and not because of passage of time (chronological age). Indeed, we further note that mitotic clocks like epiTOC1/epiTOC2 and stemTOC are not good predictors of chronological age...in stark contrast with the CpGs that make up clocks like Horvath clock, which are excellent predictors of chronological age.

Comment 6. Related to the previous point, when using the blood datasets, the authors justify to correlate methylation values with donor ages with the claim that “in line with the fact that mitotic age increases with chronological age”. With all due respect, I do not agree with this interpretation. While older tissues are more likely to have been divided more, the mitotic age is not always linked to chronological age and greatly depend on the cell-intrinsic (e.g., cell type, genetic alterations, etc) and extrinsic (e.g., microenvironment stimulus) factors. For instance, post-mitotic cells or long-term HSC will show low mitotic age and high chronological age, while faster dividing cells such as lymphocytes upon antigen recognition will accumulate greater number of cell divisions with few days. Likewise, pediatric tumors will show high mitotic ages. In line with this, the claim that “blood has a relatively high turnover rate and it is only for such tissues that we can expect a correlation between chronological and mitotic age” is not 100% accurate, as lymphocytes which have encountered their cognate antigen (ie memory B/T lymphocytes) will have greater mitotic age compared to naïve lymphocytes regardless of their chronological age. For instance, the epiCMIT mitotic clock for lowly- (naïve) and highly-divided memory B cells does not seem to be correlated with the chronological age, while the chronological Horvath clock does (Duran-Ferrer 2020). The authors could check the value of stemTOC vs other mitotic clocks in the data of Fig.2 of sorted immune cells considering naïve and memory cells separately in different age ranges (ie young vs elderly).

Response: We appreciate the reviewer's criticism and agree that stemTOC (and for that matter all other clocks too) are measuring a “total” or “average” mitotic age that takes into account not only the DNAm changes that accrue in the underlying stem-cell pool, but also the DNAm changes associated with temporary and highly dynamic progenitor cell expansions. Nevertheless, for tissues where the

underlying stem-cells are dividing fairly frequently, we would expect a broad correlation with chronological age, whilst for postmitotic tissues or tissues with a relatively low rate of stem-cell division that are also subject to tissue-turnover due to hormonal factors, such correlations, if any, would be much weaker. As remarked in our earlier responses, this is exactly what Fig.2a-e + SI figs.S5-S6 demonstrate. Moreover, as shown in Fig.2f-g, stemTOC's estimates do correlate, broadly speaking, with the intrinsic stem-cell division rate of normal tissues, suggesting that it does approximate stem-cell divisions. That the correlation in Fig.2f-g is not better is likely due to the effect of tissue turnover and expanding progenitor populations, as pointed out by the reviewer. Addressing the combined effect of stem-cell division and progenitor expansions is extremely challenging, specially given that the most recent study reporting tissue-turnover rates does not confidently quantify this for many solid tissues (see Sender R et al Nat Medicine 2021 The distribution of cellular turnover in the human body). We also acknowledge that stemTOC in blood is measuring an average mitotic age over immune cell types, and that for instance naïve and mature subsets would have somewhat different mitotic ages. We agree that this could be problematic since the naïve to mature Tcell fraction ratio decreases with age. Fortunately however, the 139 T-cell samples from BLUEPRINT (BP) are naïve CD4T-cells (see Chen & Soranzo Cell 2016) and as shown in Fig.2c there is a good correlation between stemTOC and chronological age in this naïve CD4T-cell set. We apologize that in the old version of Fig.2 we had not made it clear that the T-cells from BP are naïve CD4T-cells, but we have now revised this figure accordingly. Moreover, Fig.2c-d demonstrates clear correlations of stemTOC's mitotic age with chronological age in all sorted immune-cell datasets.

As far as the comment about epiCMIT in naïve and memory B-cells is concerned, we have checked Fig.1a from Duran-Ferrer et al, and it would appear that this analysis only included 5-10 naïve B-cells (NBCs) and 10 memory B-cells (MBCs). In our sincere opinion, these numbers are rather small, so that a negative finding could easily be down to lack of power. In contrast, the datasets we analyzed in Fig.2c-d are among the largest available sorted immune-cell datasets available, e.g.

Monocytes_Reynolds (n>1200), CD4T-cells_Reynolds (n>200), BLUEPRINT (n=139 for monocytes, 139 neutrophils and 139 naïve CD4T-cells). We have now added these sample numbers to Fig.2d, as they were previously missing, to make the point clear that these are large datasets.

To further resolve the reviewer's concern, we have now evaluated all 7 mitotic clocks in 18 large whole blood cohorts, encompassing close to 15,000 samples, correlating mitotic age to chronological age before and after adjusting for variations in 12 immune-cell fractions using our recently published 12 immune-cell type DNAm reference matrix to estimate these 12 immune-cell fractions (Quo L et al Genome Med 2023). If the reviewer's concern is justified, adjusting for the 12 immune-cell fractions should result in many associations between mitotic-age and chronological age becoming weaker or non-significant as a result of age-associated variations in immune-cell fractions and differences in mitotic age between immune cell-types. However, the results as shown below are unequivocal in demonstrating that for all hypermethylated clocks (ie epiTOC1/2, stemTOC, epiCMIT-hyper) and

RepliTali, the strong associations with chronological age remain highly significant even after adjustment for cell-type fractions (CTF):

On the other hand, for HypoClock and epiCMIT-hypo we observe how adjustment for CTF leads to a reduction in significance levels. This is consistent with our previous observation (Teschendorff Genome Med 2020) that HypoClock CpGs are immune-cell specific and hence confounded by age-associated variations in immune-cell fractions (e.g. the age-associated shift from naïve to mature T-cell subsets being one clear example, but there are others -see Luo Q et al Genome Med 2023). Hence, adjusting for these fractions would lead to a reduction in the associations with age, as seen above for HypoClock and epiCMIT-hypo.

Of note, the advantage of using whole blood to carry out the previous analysis is that for this tissue we can accurately estimate fractions for 12 immune-cell types, as demonstrated by us recently (Luo Qi et al Genome Med 2023). For other tissues, we can approximately estimate underlying cell-type fractions using our EpiSCORE DNAm-atlas. Hence, to further demonstrate that putative age-associated changes in cell-type fractions are not driving the correlations of stemTOC's mitotic age with chronological age,

we have applied all 7 mitotic clocks to 5 of the eGTEx normal-tissue datasets for which we can estimate underlying cell-type fractions. Once again, results are conclusive in demonstrating that as far as the hypermethylated clocks are concerned, the associations (significant or non-significant) with chronological age are unaltered upon adjustment for CTH, as shown below:

In response to the reviewer's point, we have now included the two above figures as new Supplementary Figures, SI fig.S7 and SI fig.S8.

Hence, overall, we think that we have resolved the reviewer's concern since we have demonstrated a clear correlation between stemTOC's mitotic age and chronological age in (i) many sorted immune-cell datasets, including a large naïve CD4T-cell subset, (ii) in 18 large whole blood cohorts (after adjusting for 12 immune-cell subtypes which includes naïve + memory subtypes, see Luo Q et al Genome Med 2023) and (iii) in 5 large normal-tissue datasets (after adjusting for underlying cell-type fractions).

Comment 7. Before benchmarking the stemTOC with other mitotic clocks, it is necessary to understand the relationship among all the mitotic clocks across normal and cancer tissues. I really miss in the analyses a section correlating all the mitotic clocks across cancer types.

Response: The reviewer presumably means a comparison between cancer and its corresponding normal-adjacent tissue, since Fig.3 is dedicated to a benchmarking of all mitotic counters across all cancer-types in relation to tumor cell of origin fraction (or tumor purity). The actual comparison of mitotic counters (epiTOC2 vs HypoClock) between normal and cancer tissue was already discussed by us in Teschendorff et al Genome Med 2020), the result being that both types of clock perform equally well with a marginally better performance for the hypermethylated clock. The comparison between normal

and cancer tissue for all 7 mitotic counters is displayed in SI fig.S11, which for convenience we display again below:

SI fig.S11: Mitotic counters discriminate cancer from normal-adjacent tissue. a) Boxplots of stemTOC's mitotic age (y-axis) using only tumor normal-adjacent pairs for TCGA cancer types with sufficient numbers of normal samples. P-value derives from a one-tailed paired Wilcoxon rank sum test. b) Corresponding AUC-values for all mitotic clocks.

Overall, in this normal vs cancer context, all 7 mitotic clocks perform similarly (with a few exceptions, AUC-values are always higher than 0.7).

Common 7.1. Is there any overlap with CpGs previously used to develop previous mitotic clocks?

Response: In response to the reviewer's question, below we display an upset plot for all 7 mitotic clocks considered here:

To help the reviewer understand this type of plot, it shows that the overlap of stemTOC with epiTOC is $21+11=32$ CpGs. The overlap of stemTOC with epiTOC2 is 11 CpGs. stemTOC only has 1 CpG in common with epiCMIT-hyper, and unsurprisingly displays zero overlap with RepliTali, HypoClock and epiCMIT-hypo. We note that this is unsurprising because stemTOC is based on TSS200 probes that are unmethylated in fetal-stage and which gain DNAm with cell-division, in contrast to HypoClock and RepliTali which are largely based on solo-CpGs mapping to PMDs, and which therefore lose DNAm with cell-division. Of note, RepliTali and HypoClock only have 1 CpG in common, but epiCMIT_hypo and HypoClock have 24 in common, with RepliTali and epiCMIT_hypo having zero CpGs in common. These patterns of overlap are consistent with the performance of the various clock as observed in this MS. For instance, HypoClock and epiCMIT-hypo were generally speaking the weakest predictors. In response to the reviewer's point we have added the above figure as a new SI fig.S9.

Comment 7.2. If stemTOC reflects mitotic cell divisions, it should be at some degree correlated with previous mitotic clocks.

Response: The reviewer is absolutely right and indeed stemTOC does correlate with previous mitotic clocks. Below we provide the heatmaps of Pearson and Spearman correlations between each pair of the 7 epigenetic mitotic clocks, in each of the sorted immune-cell datasets, in each of the normal-tissue datasets from GTEX, in each of the normal-adjacent as well as cancer-types of the TCGA. The results

demonstrate that in most cases stemTOC correlates fairly well with the 3 other hypermethylated clocks, including epiTOC1, epiTOC2 and epiCMIT-hyper, despite the fact that the CpG overlaps are not substantial. In many datasets, there also clear correlations with the hypomethylated clocks, but in others, there is no correlation between hypermethylated and hypomethylated clocks, except for RepliTali.

a)

Sorted Immune Cells

b)

Normal Tissues (GTEx)

NormalAdj (TCGA)

Cancer (TCGA)

Overall, this data once again underscores potential problems with HypoClock and epiCMIH-hypo, and indeed, as shown by us previously in the case of HypoClock (Teschendorff Genome Med 2020), a considerable fraction of the CpGs in HypoClock are cell-type specific markers, which can lead to confounding by cell-type heterogeneity.

In response to the reviewer's point we have added the above figures as new Supplementary SI figs.S12, S13 and S14.

Comment 7.3. In addition, although the separation of epiCMIH in its underlying hyper- and hypo clocks does make sense to be compared with others hyper and hypo clocks, I would suggest also to incorporate the epiCMIH value as reported originally by the authors.

Response: Whilst we very much like the Duran-Ferrer et al paper, we have a technical concern regarding the construction of the epiCMIH-value, which is the reason why we decided to separate it out into the epiCMIH-hyper and epiCMIH-hypo scores. To clarify this, for any given sample, the epiCMIH procedure compares the two separate scores and then selects the one showing the biggest deviation from the normal ground state. In our opinion, this could incur selection bias and some might argue that it violates one of the central dogmas of machine learning science. According to machine-learning dogma we should apply the same "predictor" (in this case this means the same set of CpGs and the same rule to arrive at a score) to every sample. However, the epiCMIH-procedure may change the rule (i.e. uses one of two mutually exclusive sets of CpGs) for certain samples. Consider as a counter-example a predictor like Horvath's clock. Do we change the set of CpGs to predict the chronological age of different samples? Clearly, the answer is no. Thus, for this reason, we feel uncomfortable presenting results for a "clock" (epiCMIH) that is actually a composite, and hence not a bona-fide machine-learning predictor. However, epiCMIH-hyper and epiCMIH-hypo are perfectly acceptable estimators, and what our analyses demonstrate is that epiCMIH-hyper outperforms epiCMIH-hypo quite substantially: this is an important insight that needs emphasizing, hence we prefer to keep the presentation as it is.

Comment 7.4. Also, what is the correlation of stemTOC with Horvath clock in normal tissues? As mitotic and chronological epigenetic clocks are expected to trace distinct biological processes (ie accumulated cell divisions vs chronological age), these clocks should not show a significant correlation in normal cells?

Response: We thank the reviewer for raising this good point. We already made a very detailed comparison of epiTOC1 to Horvath's clock in our original Yang et al Genome Biology 2016 publication. Horvath's clock, by construction, can never be a mitotic clock, and indeed Horvath's clock fails to discriminate normal tissue at cancer-risk from healthy normal tissue (see Yang et al Genome Bio 2016). Horvath's clock also fails to display a correlation with the Vogelstein/Tomasetti stem-cell division rate estimates. As the reviewer may know, the acceleration of Horvath's clock in cancer is also *not* universal (Steve Horvath published an erratum after his seminal 2013 paper making this point), and

indeed our Yang et al 2016 paper further verified that Horvath's clock is often even decelerated in cancer. Because it is already clear that Horvath's clock is not a mitotic clock we don't see any point in including a discussion of Horvath's clock in this MS, as this was already covered in our previous publication.

Comment 8. In Fig 2, the authors correlate their stemTOC with chronological age of different cancer and tissue types, and they interpret that this is indicative of stemTOC being an accurate mitotic clock. However, this is not surprising, as they previously selected CpGs which correlated with age in blood samples. Therefore, this a self-fulfilling prophecy, and should not be used to demonstrate that stemTOC reflects cell divisions. In line with this, the associations in Fig. 2f, 2g and 2h are not surprising and instead expected, as the authors selected for CpGs associated with age, and intrinsic stem cell divisions are known to be accumulated in aging.

Response: We thank the reviewer for raising this point. However, we must disagree with the reviewer's assessment. First of all, Fig.2a-b does **not** display data of cancers. The data shown in Fig.2a-b is for the ****normal-adjacent**** tissue of the corresponding cancer-types. We further note that this normal-adjacent tissue is not blood. When we produced Fig.2a-b, we were concerned that using the TCGA labels may confuse a reviewer, but we need to retain the TCGA label to indicate which tumor type the normal-tissue is adjacent to. Moreover, in the figure legend we clearly stated that this is normal-adjacent tissue and we used the term "Normal Adjacent Tissues" above the heatmap in Fig.2b, so it should have been clear that we were dealing with normal-adjacent tissue. Nevertheless, in response to the reviewer's point, we have now clarified in Fig.2a itself that we are only displaying normal-adjacent tissue, by changing the labels to e.g "NADJ-COAD".

In Fig.2c-d we display results for sorted immune-cell populations, as already explained in great detail in our previous responses. As mentioned earlier, demonstrating the correlation with chronological age in completely independent sorted immune-cell subsets, which were **not** used in the stemTOC CpG feature selection, is extremely important, as it shows that these associations are not confounded by CTH. The term "fulfilling prophecy" only applies to procedures that validate a predictor on exactly the same data it was trained on, which is clearly **not** the case here.

Finally, let us also clarify that Fig.2f plots the stemTOC mitotic age estimate against the annual intrinsic rate of stem-cell division (IR) of the tissue, which measures the number of stem-cell division ****per year****. In other words, IR does not increase with age: it is independent of age. Indeed, the whole reason why Fig.2f was included, is precisely to demonstrate that the association of mitotic age with stem-cell division rates is ****independent**** of chronological age. Moreover, in Fig.2g where the x-axis now labels TNSC [IR*Age], the P-value we quote has been adjusted for chronological age, as clearly stated in the corresponding Fig.2g legend.

Comment 9. The differences observed in high vs low turnover tissues (Fig 2b) could be confounded by different tumor purities (ie, sample specimens from highly proliferative tumors will contain

probabilistically a higher tumor content, whereas lowly proliferative will contain a mixture of tumor and stromal cells). As this is TCGA data, there should be matched mutational information for the majority of these cases, which could be used (eg, with ABSOLUTE-like methods) to plot the tumor content and interpret the current associations in the context of tumor purity.

Response: Once again, Fig.2b depicts results for the normal-tissue that is adjacent to cancer. That should have been clear because above the heatmap in Fig.2b we wrote “Normal Adjacent Tissues”. As such we can dismiss this reviewer’s concern, because clearly there is no tumor purity to consider.

Comment 10. The authors correlate different mitotic clocks with chronological age in different immune cell samples. However, (and related to point 6) it is known that there is a change in proportions of immune subpopulations during aging, with younger individuals having a higher percentage of B/T naïve cells which have not recognized any antigen and have not suffered any significant clonal expansion (ie lower mitotic age) and elderly donors with higher percentage of B/T memory cells which have clonally expanded upon antigen recognition (ie higher mitotic age). Therefore, if this is not considered, there will be inevitably spurious association of mitotic clocks with chronological age. These analyses should be then performed in homogenous subpopulations of cells (ie naïve vs memory) and not with a bulk sample with all immune cell types together.

Response: This is an excellent point and we thank the reviewer for raising it. We agree with what the reviewer is saying and indeed we recently published a MS (see Luo Q et al Genome Med 2023) quantifying this shift from naïve to mature subsets in over 25,000 whole blood samples. So, we agree that it would be important to assess the mitotic clocks in a naïve or memory subset, and fortunately the dataset labelled Tcell_BP in Fig.2b consists only of naïve CD4T-cells (see the BLUEPRINT paper Chen & Soranzo Cell 2016). As the reviewer can observe from Fig.2b, stemTOC’s mitotic age increases with chronological age in this naïve CD4T subset. This would suggest that there is no confounding by shifts in these immune cell subsets.

For other immune cell subtypes, available datasets of naïve or memory subsets only encompass on the order of 10 samples, which is simply not enough to do any meaningful analysis. A positive result could easily be a false positive, a negative result is most likely due to lack of power. We need many more samples to confidently call true positives or true negatives when assessing associations with age.

Hence, to further address the reviewer’s point, we have now evaluated all 7 mitotic clocks in 18 large whole blood cohorts, encompassing close to 15,000 samples, correlating mitotic age to chronological age before and after adjusting for variations in 12 immune-cell fractions, using our recently published 12 immune-cell type DNAm reference matrix to estimate these 12 immune-cell fractions (Luo Q et al Genome Med 2023). Of note these 12 immune-cell types include naïve and memory CD4T-cells, naïve and memory CD8T-cells and naïve and memory B-cells, and the estimated fractions have been well validated as shown in Luo Q et al Genome Med 2023. Hence, if the reviewer’s concern is justified, adjusting for the 12 immune-cell fractions should result in many associations between mitotic-age and

chronological age becoming weaker or non-significant as a result of age-associated variations in immune-cell fractions and differences in mitotic age between immune cell-types. However, the results as shown below are unequivocal in demonstrating that for all hypermethylated clocks (ie epiTOC1/2, stemTOC, epiCMIT-hyper) and RepliTali, the strong associations with chronological age remain highly significant even after adjustment for cell-type fractions (CTF).

On the other hand, for HypoClock and epiCMIT-hypo we observe how adjustment for CTF leads to a reduction in significance levels. This is consistent with our previous observation (Teschendorff Genome Med 2020) that HypoClock CpGs are immune-cell specific and hence confounded by age-associated variations in immune-cell fractions (e.g. the shift from naïve to mature T-cell subsets being one clear example but there are others -see Luo Q et al Genome Med 2023).

Comment 11. The authors plot stemTOC vs chronological age in sorted neuros, and claim that there is no correlation (Person=0.21, P-value=0.06). However, there is clear batch effect (ie the cohorts clearly

clustered separately) that seems to confound the total correlation. How does the correlations look like within each cohort? Following the same rationale as the previous points, I do expect a correlation with age of the samples, which would suggest that the stemTOC is related strongly linked to methylation changes associated with age.

Response: We thank the reviewer for raising this valid point. Whilst old Fig.2e did not represent a clustering, the reviewer is absolutely right that there appeared to be a batch effect, affecting at least one of the cohorts. It is precisely because of this, that the P-value we had quoted in old Fig.2e had been adjusted for cohort. In other words, we ran the linear model $\text{lm}(\text{stemTOC} \sim \text{age} + \text{cohort})$, which in effect runs the linear regression within each cohort individually to arrive at a global P-value of association between stemTOC and chronological age. We think that the reviewer was misled by us drawing a regression line through all of the data, but the actual P-value and PCC that we quoted in the figure were unrelated to this regression line. Reason why we had plotted all datasets together is that each single cohort is quite small and they also tend to differ in terms of the age-distribution. So running the multivariate linear model above was a way to improve power, but is still subject to confounding by non-linear batch effects. Hence, we agree that it would be better to present the data for each cohort separately, i.e. separately quoting the PCC and P-values for each cohort, as requested by the reviewer. Below is the new plot, which has been incorporated into Fig.2 as the new panel Fig.2e:

Based on this plot, there is no clear correlation between stemTOC and chronological age. There is only one cohort where there is a marginal association ($P=0.04$).

Thus, if we now contrast these results with those of the normal-adjacent rectal tissue shown in Fig.2a below for which we only have $n=7$ samples (ie LESS samples than sorted neurons) and where there is a very clear correlation between stemTOC and

chronological age, this is in our view convincing evidence that stemTOC is indeed capturing a process related to cell-division:

Incidentally, the $n=7$ NADJ-READ samples display a very tight age-range of ~ 20 years as most are in the range 50 to 70 years, whilst the sorted neuron datasets contain more samples and the age-range is often 50 years if not bigger. This further supports the view that the absence of a correlation in the sorted neuron datasets is not because of a lack of power or because of a tight age-range, and that therefore stemTOC is truly not correlated with age in sorted neurons. Having said this, we can't however exclude a very small correlation, which would not be inconsistent with the fact that neural stem cells can give rise to new neurons even in adult brain, see e.g. Denoth-Lippuner A et al Nat Rev Neuroscience 2021.

Comment 12.1: The claim that hyper- mitotic clocks are more accurate than hypo-clocks based solely on their correlation with chronological age in tissues is from my point of view an overinterpretation of the data. As previously mentioned, for tissues whereby there is a high turnover such as blood, chronological age and mitotic age are not expected to correlate. For instance, naïve T/B cells will always show a lower mitotic age compared to memory T/B cells regardless of their chronological age, as the latter have suffered a massive clonal expansion upon antigen recognition. It is therefore expected that this proliferation will lead to higher DNA methylation changes than the comparatively fewer cell divisions of hematopoietic stem cells accumulated during aging. In addition, the data from fig S4 analyze samples which are composed by a mixture of different cell types, which in turn could also change during aging, and therefore will most likely lead to confounding associations (as each cell type has its own DNA methylation profile).

Response: We thank the reviewer for raising these points. First of all, we provided very clear proof in our previous Teschendorff et al Genome Med 2020 paper that for instance the solo-CpGs of the HypoClock are strongly confounded by cell-type heterogeneity (CTH), and this is precisely why results with HypoClock are so much more variable and inconsistent. It is also the reason why it does so badly in the sorted immune-cell populations (Fig.2d): in sorted cells, the CTH is much lower, so that age-associated changes in CTH can't drive the artefactual association with age. Indeed, Peter Laird's lab, who produced the HypoClock, have seemingly acknowledged this now, which is why they have produced a new clock called RepliTali (Endicott et al Nat Commun 2022) using cell-line data to select

CpGs, but which is largely still based on sites losing DNAm with cell-division/age. The data we display in this MS confirms that RepliTali performs much better than HypoClock, precisely because RepliTali is much less confounded by CTH. Importantly, we have used formal statistics to compare all clocks, not only in the context of associations with chronological age, but also in terms of associations with tumor cell of origin fraction (tumor purity), in terms of associations with preneoplastic lesions, and in relation to literature based stem-cell division rates, and the results shown in Fig.2h, Fig.3e, SI fig.S5c, S6c, S10d all converge and are unequivocal in demonstrating that clocks based on hypermethylated sites do better in the context of normal tissue homeostasis and in early precancerous lesions. Moreover, in the context of comparing normal to cancer tissue, hyper and hypoclocks perform similarly, because the mechanism of DNAm loss is more relevant when cells are under high replicative stress.

Comment 12.2: Other conflictive data that strongly argues against this claim comes from the study by Endicott et al 2022, which has also been used by the authors to derive the stemTOC. In that study, the authors used pure cell types under controlled settings and traced the number of population doublings, and concluded that RepliTali and epiCMIT mitotic clocks were more accurate in their controlled settings.

Response: Concerning the Endicott et al paper, there is absolutely no conflict whatsoever. First of all, figure Fig.4b,c,d of Endicott et al demonstrate that the hypermethylated clock epiTOC2 also does very well. Indeed, we have analyzed Endicott et al's cell-line data very extensively and below is a side-by-side comparison of epiTOC2 to epiCMIT in the 6 cell-lines analyzed here:

As the reviewer will see, the correlations of epiTOC2 with population doublings (PD) are excellent, whilst epiCMIT fails to correlate in one of the cell-lines, and displays lower R^2 values in 4 of the 6 cell-lines. Second, Endicott et al did not perform an extensive and comprehensive comparison of hyper and hypo-clocks in the context of correlations with chronological age, nor with tumor cell of origin, nor with precancerous states, and it is precisely in these contexts where we see the hypermethylated clocks do much better.

Comment 12.3: Finally, human data from Duran-Ferrer et al 2020 show how distinct B cell tumors patients show higher tendencies to gain or loss DNA methylation, suggesting that hyper- or hypomethylation mitotic clocks may be more appropriate in certain situations. This data led to the authors to collapse the epiCMIT hyper- and hypo clocks into its highest value. Therefore, the authors should provide much more (solid) data to make this claim, and discuss it in the context of the existing literature.

Response: We appreciate the reviewer's point, but as remarked earlier, we are concerned that the epiCMIT-procedure of selecting the score displaying the greatest deviation may inadvertently incur selection bias. We sincerely opine that if in one tumor sample, the set of hyperM CpGs performs "better" whilst in another tumor sample, it is the set of hypoM CpGs that performs "better", that an equally valid interpretation is that the neither set of CpGs is a robust marker of mitotic age. In other words, we think that it is necessary to separate out epiCMIT into epiCMIT-hyper and epiCMIT-hypo, because otherwise we don't have a predictor that conforms to the rules of machine-learning science. Indeed, it is a well-known golden-rule of machine learning science that one should always apply the same predictor (in our context, this means the same set of CpGs and the same rule combining these CpGs) to every sample. That in some samples some CpGs may be more important than others is fine, as long as all CpGs are included and as long as the rule for combining the CpGs is fixed. However, the epiCMIT procedure changes the rule for each sample, i.e it uses potentially different CpGs for different samples. Moreover, what the data in our MS clearly demonstrates is that epiCMIT-hyper is a much better predictor of mitotic age and cancer-risk than epiCMIT-hypo, which is another strong argument for not combining these separate scores.

Comment 13.1 I have some serious concerns about the biological interpretation of all the data presented in the subheading termed "stemTOC tracks mitotic age of the tumor cell-of-origin", line 177. I present them in the following paragraphs: First, I truly do not understand and found counterintuitive the following claim: "we reasoned that tumors with a higher proportion of the cell-of-origin would display a higher mitotic age if mitotic age is a key marker of tumor progression". By definition, a tumor with a higher proportion of the normal cell-of origin cell type, ie normal healthy cells, will have a lower mitotic age compared with tumors samples with a lower proportion of the normal cell-of-origin cells (ie higher

proportion of tumor cells), as cancer cells will inevitably have higher number of accumulated cell divisions and therefore a higher mitotic age.

Response: We thank the reviewer for raising this point. That section has the heading “stemTOC tracks mitotic age in the ***tumor*** cell of origin”. Hence, in the subsequent sentence we implicitly meant the ***tumor*** cell of origin, not a normal cell, so we apologize for the confusion caused. In any case, we have now rephrased the beginning of this paragraph, to signal that we are talking about a tumor cell. Hence, this should resolve the reviewer’s point.

Comment 13.2. Second, and related to the previous point, cancer samples do maintain a DNA methylation signature of their cell-of-origin (eg, PMID: 29625048 & PMID: 37582362, among others). Overall, in a tumor sample, the infiltration of the normal cell cell-of-origin is expected to be comparatively minor or even negligible with respect the proportion of tumor cells. In other words, the inferred proportion of the putative cell-of-origin predicted by DNA methylation in tumor samples will likely represent the percentage of tumor cells in the sample, ie tumor purity. This had been shown to be the case in blood tumors (Duran-Ferrer 2020). Therefore, the observed correlation between the cell-of-origin signature and mitotic clocks most likely reflects tumor purity. In solid tumors, this technical artifact may be expected, as tissue samples that have not been purified from tumors with higher proliferation will probabilistically contain a higher proportion of tumor cells. This is critical to be assessed to properly interpret the data. To address this, the authors could use matched genomic data from this TCGA samples and use methods like ABSOLUTE to infer the tumor content through genomic data and correlate it with the identified cell-of-origin proportions.

Response: We completely agree with the reviewer, and indeed, what we are measuring with the EpiSCORE tumor cell of origin fraction, is in effect, an estimate of tumor purity. Indeed, the reviewer’s request was already addressed in our EpiSCORE DNAm-atlas Nat Methods 2022 paper, where we showed that EpiSCORE’s tumor cell of origin fraction correlates reasonably well with independent tumor purity estimates obtained by other methods, including CPE, ABSOLUTE, ESTIMATE, IHC and LUMP. On a separate but related note, it is worth emphasizing here that our EpiSCORE fractions would be able to discriminate say liver hepatocellular carcinoma from liver cholangiocarcinoma, since we can estimate hepatocyte and cholangiocyte fractions. Clearly, this is not possible with traditional tumor purity estimators such as e.g. ABSOLUTE.

Comment 13.3. The correlations in figS8 of stemTOC and cell-of-origin in cancer and precancerous lesions but not in normal supports the previous point 13.2, ie that the cell-of-origin prediction reflects tumor purity. The data in FigS8 shows that in normal conditions, the proportions of the cell-of-origin normal subtype that will give rise to the tumor such as luminal cells in breast cancer or cholangiocytes in cholangiocarcinoma, is always lower, consistent with a normal cell composition of a healthy tissue. In premalignant lesions, the malignant cell derived from eg luminal or cholangiocytes starts to proliferate

and occupies a higher proportion of cells in the sample (ie higher cell-of-origin prediction), which becomes even higher upon overt cancer. Again, it is imperative to correlate the cell-of-origin prediction with tumor purity prediction from matched genetic data in both pre- and malignant conditions to clarify this issue and properly interpret the data.

Response: We completely agree with the reviewer's interpretation, and indeed in Fig.3 as well as in new SI figs.S16 (old figS8 is now fig.S16), we could label the x-axis with "tumor purity", for reasons given in the response to the previous point above. Unfortunately, the samples being displayed in SI fig.S16 do not have matched CNV or genetic data for us to obtain "tumor purity" estimates, specially because most of these samples are precancerous lesions, where tumor purity estimation using traditional methods would be much harder. Since we have already clearly demonstrated that our EpiSCORE tumor cell of origin fractions correlates with tumor purity across all main TCGA cancer-types (see Zhu T et al Nat Methods 2022) we don't see why we need to repeat that analysis here. In response to the reviewer's point, we have now clarified throughout the main text that EpiSCORE is measuring tumor purity ***and that we can do so also in precancerous lesions, which is exactly why our MS should be of interest***.

Comment 13.4. Finally, and following the same line of thought of previous comments, the fact that stemTOC is the mitotic clock with the highest correlation with the cell-of-origin across tumor types (fig. S5C) would be indicative of it being the one mostly affected by distinct tumor cell content. Therefore, this would make stemTOC a less useful mitotic clock in samples with lower tumor content.

Response: We appreciate the reviewer's point. However, we have to disagree with the reviewer's interpretation, as we take the precise opposite view. A stronger correlation with the tumor cell of origin fraction means that this particular mitotic clock would be more sensitive to capture the subtle increases in mitotic age in the tumor cell of origin of a precancerous lesion, which may in fact constitute a cancer-risk marker. To further clarify this point, suppose you want to build a predictor that can discriminate normal healthy tissue from normal "at cancer-risk" tissue (say a precancerous lesion). If you have a more sensitive clock that can detect the marginal subtle increase of mitotic age in that lesion, resulting from a marginal increase in the tumor cell of origin population, then surely that is better! So, this is exactly why stemTOC delivers an improvement, as shown in new SI fig.S15 (old SI fig.S7) (see next point).

Comment 14. The authors conclude that "formal comparison of all clocks across the 9 datasets revealed an overall improved performance of stemTOC (SI fig.S7)". However, the data does not seem to clearly support this conclusion, but instead shows a similar performance of mitotic clocks across cancer types, with hyper- and hypo behaving differently in certain tumor types.

Response: For convenience, we redisplay old SI fig.S7 (new SI fig.S15) below:

SI fig.S15: Benchmarking of stemTOC in precancerous conditions . a) Mitotic-age estimates for stemTOC, RepliTali and epiCMIT-hyper in three different DNAm datasets encompassing normal healthy tissue and precancerous lesions, including prostate cancer progression (Benign, Neoplasia, Primary and Metastasis), progression of intestinal metaplasia (N=normal, MildIM=mild intestinal metaplasia, IM=intestinal metaplasia) and esophageal adenocarcinoma (N=normal healthy squamous, BE=Barrett's Esophagus, EAC=esophageal adenocarcinoma). Number of samples in each group is indicated. We provide one-tailed P-values from a Wilcoxon rank sum test comparing successive stages. b) Barplot displaying the AUC from the Wilcoxon test comparing normal-healthy to normal at-risk groups across a total 9 different DNAm datasets, and for 7 different mitotic clocks. c) Heatmap displaying one-tailed paired Wilcoxon rank sum test P-values, comparing clocks to each other, in how well their mitotic age distinguished normal-healthy from normal at-cancer-risk tissue. Each row indicates how well the corresponding clock's mitotic age estimate performs in relation to the clock specified by the column. The paired Wilcoxon test is performed over the 9 datasets.

In the above figure, it is important to pay attention to panel-c), because there we use formal statistics to compare the AUC values obtained by all clocks in each study. What this panel shows is that stemTOC outperforms 4 other clocks (epiTOC2, epiCMIT-hyper, HypoClock and epiCMIT-hypo), and that in no study does a clock outperform stemTOC. Thus, whilst we agree that stemTOC does not outperform RepliTali, it nevertheless performs favourably. This is why we used the adjective “overall”. Moreover, we end that section by stating “Overall, these data underscore the potential of stemTOC’s mitotic age to indicate cancer risk in precancerous lesions.”, thus avoiding any statement that directly compares clocks.

Comment 14.1. How the boxplots in Fig S7a look like for all the mitotic clocks? Please, could you represent all the boxplots for all the mitotic clocks?

Response: There are obvious space limitations, hence why old SI fig.S7a just shows a selected number of clocks and studies. Indeed old fig.S7b-c (new fig.S15b-c) summarizes the results, and it is this summary that readers will be most interested in. However, if the editor thinks it is necessary we are happy to add all these figures (lots of figures!) to Supp.Information.

Comment 14.2. From Fig S7B, there are really tiny differences in AUC across mitotic clocks and cancer types, which likely do not represent notable biological effects. In addition, in half (4/9, breast, lung, prostate and esophagous) of the cancers analyzed stemTOC shows a worse AUC value compared to other mitotic clocks (either hypo or hyper).

Response: We appreciate the reviewer's point. First of all, let us point out that we can't compare one clock (stemTOC) to another "composite clock" that is made up of six other clocks. That would not be a fair comparison. It is precisely because the differences in AUC values are *sometimes* small, that we decided to compare the clocks formally using a paired Wilcoxon-test, and the result in old Fig.S7c (new SI fig.S15c) shows that stemTOC outperforms 4 other clocks (epiTOC2, epiCMIT-hyper, HypoClock and epiCMIT-hypo), and that in no study does a clock outperform stemTOC. Thus, whilst we agree that stemTOC does not outperform RepliTali, overall, in comparison to some of the other clocks, there is a statistically significant improvement.

Comment 15.1 At line 329, there is the following conclusion: Given that in normal-adjacent tissues, stemTOC displays correlations with TNSC without evidence of a saturation effect (Fig.2e), this suggests that stemTOC is a more sensitive tracker of mitotic age than somatic mutations". First, is it possible that the reference to figure panel fig 2e is wrong? There is no TNSC info in the plot.

Response: The reviewer is absolutely right. It is a typo. We meant Fig.2f-g, and we have now corrected this.

Comment 15.2. Second, could the authors plot the number of SBS1 mutations for adjacent tissues? I expect to see also an increase for SBS1 numbers. This is noted as better performance of stemTOC compared to SBS1 but the authors so not show data on SBS1.

Response: We would love to, but that data does not seem to be publicly available. The mutational signature loads have to our knowledge only been published for the TCGA cancers, not for the normal-adjacent tissues. For a number of normal tissue-types, SBS1 loads have been characterized by Stratton & Campbell groups, and published as separate stories, but for these normal-tissue samples there is no corresponding DNAm data, so a direct comparison in normal-tissues is not yet possible.

Comment 16.1 At line 332, the authors state: "Of note, a strong correlation with tumor cell of origin fraction is not seen if one were to use the MS1-load as a proxy for mitotic age (SI fig.S12), and correlations of MS1-load with CPE-based tumor purity estimates were weaker than with EpiSCORE-

derived tumor cell-of-origin fractions (SI fig.S13)". There is evidence that from hundreds to thousand mutations associated to SBS1 are not only accumulated in cancer genomes, but also in normal healthy cells during aging. Therefore, both analyses showed at fig S12 and S13 should consider the age of the patient at sampling.

Response: We appreciate the reviewer's point but don't fully understand what the reviewer is driving at here. The underlying issue with these analyses is that there is a lack of normal tissue data with matched somatic mutations AND DNAm calls (see previous point). That SBS1 load increases with age, be it in normal or cancer-tissue is indeed well-known, but the data obtained in normal tissues are not from TCGA patients, whereas old SI Fig.S12 and old SI Fig.S13 deal with TCGA cancer samples. The purpose of these two figures was simply to see if MS1/SBS1 correlates with tumor purity, as measured by EpiSCORE's tumor cell of origin fraction (old SI.Fig.S12), or by CPE (a measure of tumor purity from Aran D et al Nat Comm 2015) (old SI Fig.S13). We don't see how age of the patient is relevant to the tumor purity estimate.

Comment 16.2. Next, the authors conclude that "Overall, these data indicate broad agreement between DNAm and somatic mutational, but with DNAm being a more sensitive marker of mitotic age". Sorry but I cannot see the evidence from the previous data, could you please explain this a little bit more?

Response: We are happy to clarify this. First of all, let us make it absolutely clear that in order to show this convincingly one would need to have matched somatic mutational and DNAm data for many different normal tissue-types, each with a wide enough age-range. As mentioned earlier, we are not aware that such a dataset exists. Hence, the evidence we present is preliminary, and yet it is strong enough that we consider it worthy of dissemination. The key observations that provide the supporting evidence for our statement are shown in Fig.2f-g and Fig.5e-f. Focusing first on Fig.2f-g, which displays data of normal-adjacent samples, we can see that the correlation between stemTOC and IR is fairly linear. Clearly, there is no evidence of a saturation effect here. On the other hand, Fig.5e-f, which displays the stemTOC values for cancers, clearly demonstrates a saturation-effect: some cancer-types like BRCA and LIHC, already display extremely high stemTOC values, as large as e.g. colon cancer (COAD). This "saturation effect" reflects the highly proliferative nature of the tumor samples, so that in effect the mitotic age of the normal cell of origin is no longer relevant. Indeed, the intrinsic rate of stem-cell division of a tissue like breast or liver is not as high as that of colon, and yet the corresponding tumor types all show comparable stemTOC values. In contrast to stemTOC, MS1/SBS1 displays a linear pattern for the actual cancers, as shown in Fig5a-c. The obvious interpretation therefore is that we see a saturation effect for stemTOC in the case of cancer samples, because the very high proliferation rate of cancer cells drives big increases in DNAm that eventually "saturate" at the maximum values of >0.9, as is clear from Fig.5e. This saturation effect is not seen for MS1/SBS1 because the rate at which somatic mutations are acquired is smaller than the rate at which DNAm changes are acquired. If we now return to the normal-tissue data, it follows by extrapolation that

stemTOC should also be a more sensitive marker of mitotic age than SBS1/MS1. Although, as mentioned before, we lack the matched DNAm and somatic mutation cells in the normal-tissue samples to prove this, the increased sensitivity of DNAm should not be surprising: it is actually already well-known that the rate of epimutations in normal tissue is about 10-100 times higher than that of somatic mutations (see e.g. Horsthemke et al <https://pubmed.ncbi.nlm.nih.gov/16909906/>).

Indeed, a major reason why detecting somatic mutations in normal tissue has proved to be so much harder than detecting DNAm changes, is that the number of cells carrying a given somatic mutation alteration is much smaller (see e.g. Abascal F et al Nature 2021). In response to the reviewer's point, we have now clarified this important point further in the Discussion section of the MS.

Minor points:

Comment 1. An initial table/fig to show whole study design would improve clarity

Response: We appreciate the comment, but don't think that this would necessarily be helpful. Instead, we have made improvements to Fig1a-b to further clarify the CpG selection procedure. In effect, Fig.1 describes the construction of stemTOC, and we don't feel it is necessary to have another initial figure to describe the validation and applications of stemTOC.

Comment 2. In Fig1, in the selection of CpGs using cell culture, it seems that the green line is the color that represents the selected CpGs (upper panel). However, in the bottom panel, could it be that the green color represents the excluded CpGs...? To clarify this, I would emphasize with a small legend what the different color code means (ie selected vs excluded CpGs), and would also avoid to use colors from the previous panel (ie, pink as in muscle group, to avoid possible misleading interpretations).

Response: The reviewer has raised a valid question. In the bottom cell culture panel the green line is the same as in the top panel indicating CpGs that display DNAm-gain with PDs/days in culture whereas the blue line (we have changed the color from magenta to blue) indicates the subset of CpGs that no longer display gains when the cell-line is treated with a cell-cycle inhibitor or if it is grown in low serum conditions. The meaning of the colors was provided as legends already. Hence, the vitro-mitCpGs are the subset of those that gain DNAm with PDs in 6 different cell-lines which do not display gains when growth is suppressed. We have now modified Fig.1a to make this clearer.

Comment 3. Please, could you indicate a measure of effect size in Fig 4B? Also, indicate always the p-value, although not being significant.

Response: We don't understand what the purpose of this is, when the effect size can be inferred from the actual figure. The reviewer also did not specify which panel in 4B he wants the effect size for. Concerning the comment about P-values, let us clarify that in any panel where a P-value would be of interest, we do specify it, regardless of whether it is significant or not. So, for instance, in Fig.4B there is no P-value in the panel comparing the epithelial and immune-cell fractions to each other, because

whether there is a significant difference or not is largely irrelevant to the message that Fig.4B is trying to convey. The purpose of Fig4B is to show that the mitotic age of buccal swabs increases with the epithelial fraction and to show that the difference in mitotic age between smokers and non-smokers increases with age. For this reason, we add P-values to the respective panels to support our claim and to convey the important and relevant message. The panel comparing immune-cell and epithelial fractions in buccal swabs is only shown to give the reader an idea of what the mean and variance of these fractions are in buccal swabs. Whether the immune cell fraction is significantly lower or higher than the epithelial fraction is not of interest, so adding a P-value to this panel would be an unnecessary distraction.

Comment 4. It would be of interest to include the value of other mitotic clocks in all the supplementary tables together with the stemTOC score.

Response: We agree and we have now done so, adding these values to new SuppTablesS2-S4.

Reviewer #2

General Comments:

Comment: As with fishing expeditions, gear, bait, and method of capture are essential. Fishing for 'mitotic CpGs' that reflect the overall number stem cell divisions in various tissues (normal, pre-cancer, cancer) is an ambitious goal, out in the open sea of the DNA methylome. The study is well justified and the authors go about it with great effort analyzing a large number of relevant data sets. The rationale given is clear. Since mitotic age has been shown to be generally an important risk factor for cancer (as demonstrated e.g. by Tomasetti and Vogelstein), a quantitative assessment of mitotic tissue age, as proposed by the authors, offers a new paradigm/biomarker for evaluating the neoplastic potential of a tissue in an individual because replication errors (in DNA sequence or in DNA methylation imprints) occur during DNA replication and contribute to cancer.

Response: We sincerely thank the reviewer for taking time to evaluate our MS. We agree that trying to identify CpGs that track mitotic in human tissues is a very ambitious goal, and we are delighted that the reviewer acknowledges the justification, rationale and comprehensiveness of this work.

Comment: Complementing purely statistical approaches that have no biological underpinnings (e.g. Hannum's and Horvath's elastic nets), the authors offer a biological approach to identify a new clock based on DNAm changes, starting with essentially hypomethylated CpG sites in fetal tissues. Theoretical explanations of why stochastic gain in DNA methylation during DNA replication may occur, have been given by others (Genereux et al, PNAS 2005; Sontag et al, JTB 2006). This process is inherently stochastic. What exactly the authors mean by 'quasi-stochastic' (line 118) is not clear as the molecular machinery that maintains methylated and/or unmethylated CpGs is inherently error-prone upon DNA replication, and the resulting drift is naturally stochastic along a given cell lineage involving 100s if not 1000s of cell divisions. It is hard to make sense of the sentence between lines 118-120. For any given subject, using the bead arrays, DNAm is measured in bulk across many cells and the observed drift (change in methylation) at any given CpG ought to reflect the mitotic age for all CpGs in the selected clock-CpG set, not just in random subsets. It might help to clarify what the authors mean by 'quasi-stochastic' and their reference to random subsets.

Response: We thank the reviewer for raising this point. We agree that the notion of "quasi-stochasticity" can be confusing and that we did not describe its meaning. We had used the term "quasi-stochasticity" to refer to the empirical observation that DNAm changes in aging and precancerous lesions (as well as cancer) are happening preferentially in certain parts of the genome (e.g. PRC2-marked sites) but that otherwise they appear quite random. However, to avoid unnecessary confusion, we have decided to replace "quasi-stochasticity" with "stochasticity" throughout the whole MS. In the context of this MS, "stochasticity" is also and primarily being used to refer to the random subset of CpGs within the larger

pool of mitotic-CpGs that track mitotic age within each subject. To clarify this further, we would agree with the reviewer that in theory all 371 mitotic-clock CpGs are tracking mitotic age, but in reality the tissues we are analysing are dynamic mosaics of subclones. In a precancer lesion, it is plausible for a specific subclone with a marginally higher mitotic-rate to mildly outcompete other clones, so that say 30% of the sample is now made up of this specific subclone. In terms of the DNAm patterns of the 371 mitotic-clock CpGs, it would be unrealistic to assume that all 371 CpGs are “synchronously tracking” this particular subclone. Instead, it is more realistic to assume that owing to the stochasticity of the DNAm changes (as outlined for instance by Genereux and Sontag’s papers), that the 371 mitotic-clock CpGs display mild differences in DNAm levels, and hence that the sudden emergence of the precancerous subclone is only marked by a subset of these CpGs, as indicated in Fig.1c. Key point here is that we have observed these 30% DNAm outliers in several studies comparing normal to normal “at cancer-risk” tissue (e.g. our EVORA & iEVORA papers) and that these CpG outliers vary between independent subjects, and so there is empirical evidence in favour of considering a metric for mitotic age that captures the mitotic-age of the dominant subclones. This is why we take an upper 95% quantile of the DNAm distribution over the 371 mitotic-CpGs, because this is more likely to capture these subclones. As shown by our previous EVORA/iEVORA studies (Teschendorff et al Genome Med 2012, Nat Commun 2016), capturing these DNAm-outliers marks subjects at higher risk of cancer. Of note, we had mentioned this connection to subclonal expansion in Discussion where we stated that “Indeed, by approximating mitotic age with an upper quantile statistic over the pool of mitotic CpGs (as opposed to taking an ordinary average), we can better account for the inherent inter-CpG and inter-subject stochasticity, enabling the identification of DNAm outliers that very likely reflect epigenetic mosaicism and subclonal expansions.”

In response, to the reviewer’s point, we have now clarified these points more in the Results section. Finally, we thank the reviewer for reminding us of the two papers by Genereux and Sontag, which we now also cite in the Introduction of the MS.

Comment: Similar, for clarity, it may help readers to better understand what the authors mean when they refer to “hypomethylation” vs “hypermethylation”, since many readers may confuse the state of a group of CpGs (e.g. CpG island as being lowly (hypo) methylated vs highly (hyper) methylated) with the process of gradual loss vs gain of DNA methylation in clonal cell populations.

Response: The reviewer is absolutely right that there are different conventions in terminology being used in the community. We have long used the terms “hypo” and “hyper” to refer to the process of losing and gaining DNAm, because it is a convenient shorthand for saying “losing DNAm” or “gaining DNAm”. Moreover, if we want to denote a region of the genome where DNAm levels are low (say <0.2), we can refer to this as “unmethylated”, and likewise regions that have high DNAm-levels (say >0.75) we can simply refer to them as “methylated”. In response to the reviewer’s point we have now clarified the meaning of the hypo and hypermethylation terms when first used in the text.

Comment: An interesting finding is the finding that the stemTOC correlates well with the tumor stem cell-of-origin fraction. Maybe this is obvious, but this reader is somewhat puzzled by why this should be so. Naively one might expect that in any tumor, the fraction of the cell-of-origin is approximately constant and not related to size or time of the clonal expansion (sojourn time). What exactly is expected to change in the cell-type composition of the tumor so that the fraction of the cell-of-origin increases with mitotic age? Would the authors care to elaborate?

Response: We thank the reviewer for raising this question, as it suggests that there is a potential misunderstanding in how we are defining “tumor cell-of-origin”. Effectively, our definition refers to the tumor cell fraction that is derived from the cell-of-origin. So, for instance, if we are talking about melanomas, then by tumor cell of origin fraction in a given melanoma sample, we mean the fraction of “tumor cells that derive from the tumor cell of origin, which is a transformed melanocyte”. Hence, the positive correlation we observe between stemTOC’s mitotic age and EpiSCORE’s tumor cell of origin fraction makes biological sense, because the higher the fraction of proliferative tumor cells in a tumor sample, the less non-proliferating stromal cells you have, hence the average mitotic age of the tumor sample should go up with the number of tumor cells. In effect, EpiSCORE’s tumor cell of origin fraction is an estimate of tumor purity, as shown by us previously (see Zhu T et al Nat Methods 2022), although it is also important to realize that it is not equivalent, as the tumor cell of origin fraction is more informative (e.g. we can use to discriminate liver hepatocellular carcinomas from liver cholangiocarcinomas, whilst traditional tumor purity indices may not). Hence, whilst the result depicted in Fig.3 is biologically intuitive, demonstrating this as we have done in this work, at cell-type resolution, is important, as it shows that the mitotic age estimates from stemTOC could be used to detect subtle increases in pretumor cells in a preneoplastic context, and indeed we dedicated a whole Fig.4 to demonstrating this in the context of premalignant conditions, as well as normal tissues exposed to major cancer risk factors.

In response to the reviewer’s point, and to avoid further confusion, we have rephrased our terminology in the Results section, and have also clarified in Discussion why the results depicted in Fig.3 are biologically intuitive, yet nevertheless also very significant as it forms the basis for detecting subtle increases in mitotic age in normal cells exposed to cancer risk factors, as shown in Fig.4.

Comment: Finally, upon publication, a list of the 371 stemTOC CpGs and their annotations should be provided in the SI.

Response: The 371 stemTOC CpGs are part of the EpiMitClocks R-package that we have made freely available from our github repository: <https://github.com/aet21/EpiMitClocks> Nevertheless, we agree that we should also provide them in a Supp.Table, and so we have now included them in a new SI table S1.

Minor points:

Line 147: It would be helpful to also reference the clocks used fig.S4 in the Supplemental Material.

Response: We appreciate the reviewer's point but at the stage of the MS when we first referred to old SI fig.S4 (new SI fig.S6), we are exclusively dealing with stemTOC and it would disrupt the flow if we suddenly refocus the discussion on benchmarking, as indeed, we have a separate subsection dedicated to the benchmark where we reference the old SI fig.S4 (new SI fig.S6) again.

Line 249: "Of note ..." Notwithstanding a larger effect size, the rationale given that the "upper quantile" (the authors mean the 95% percentile) somehow minimizes the underlying stochasticity of DNAm patterns leading to better mitotic age estimates, is suspect. It is unclear whether or not using the interquartile range (i.e. removing extremes) would not yield better correlations.

Response: We thank the reviewer for raising this point. We think that the reviewer may have misunderstood why we take an upper quantile and why we take a quantile as high as 95%. Taking an upper quantile is certainly **not** to minimize the underlying stochasticity. On the contrary, taking an upper quantile over the CpGs is intended to *capture* the underlying stochasticity that characterizes DNAm changes in precancerous lesions. In fact, based on earlier work, see Teschendorff et al Genome Med 2012, Nat Commun 2016 and Nat Rev Genetics 2018, there is strong evidence that DNAm outliers capture subclonal expansions that correlate with cancer-risk. These outliers represent the biggest DNAm deviations from the ground state, which is why taking a 95% upper quantile over as many as 371 CpGs is sensible as this achieves an optimal trade-off between capturing these DNAm outliers (the higher the quantile the better) and the unwanted sampling variability (which increases for larger quantiles). Taking say a smaller quantile such as 75% would not be consistent with what we have learned from previous papers studying DNAm patterns in precancerous lesions. Indeed, in the context of this study, we also in fact explored a lower quantile such as 75%, but this gave results that were more similar to taking an average, which is why it was not displayed in SI fig.S18. As shown in SI fig.S18 (displayed again below for the reviewer's convenience), taking the 95% upper quantile leads to a marginal improvement in terms of AUC, but a more significant improvement in terms of the effect size:

SI fig.S18: Accounting for stochasticity improves associations of mitotic age. a) Barplots depict the effect size (effS) of the average mitotic age difference between cancer and normal-adjacent tissue for TCGA cancer-types with at least 9 normal samples, and with the mitotic age per sample computed using a 95% upper quantile (UQ, stemTOC) or the mean (Mean) over the 371 mitotic-CpGs. Lower barplots display the corresponding AUCs discriminating cancer from normal-adjacent tissue. For both sets of barplots, we provide the one-tailed P-value from a paired Wilcoxon test comparing UQ to mean. **b)** Barplot depicts the Pearson Correlation Coefficient (PCC) between mitotic age and tumor cell of origin fraction across TCGA cancer-types, with the mitotic age per sample computed using a 95% upper quantile (UQ, stemTOC) or the mean (Mean) over the 371 mitotic-CpGs. We provide the one-tailed P-value from a paired Wilcoxon test comparing the PCCs from using UQ (stemTOC) to those from using the mean. **c)** Upper barplots depict the effect size (effS) of the average mitotic age difference between normal-healthy and normal “at-cancer-

risk” tissue for various cancer-types, and with the mitotic age per sample computed using a 95% upper quantile (UQ, stemTOC) or the mean (Mean) over the 371 mitotic-CpGs. Barplots below display the corresponding AUCs discriminating normal-healthy from normal at-cancer-risk tissue. For both sets of barplots, we provide the one-tailed P-value from a paired Wilcoxon test comparing UQ to mean.

Line 320: Fig 2e) shows neurons. ??

Response: We thank the reviewer for pointing this typo to us. We have now corrected this, as the correct reference should be Fig.2f-g.

Line 331: “Of note ...”. It appears that in this comparison with MS1, stemTOC was adjusted for tumor cell-of-origin while MS1 load is not. Is a fair comparison even possible?

Response: We thank the reviewer for asking this. First of all, let us clarify stemTOC was *not* adjusted for tumor cell-of-origin. In fact, stemTOC correlates with the tumor cell of origin fraction, in line with the expectation that mitotic age would be higher in tumors where the corresponding tumor cell fraction is also higher (since cancer cells are highly proliferative). Old SI fig.S12 (new SI fig.S20) was merely intended to explore if the MS1 mutation load also correlates with tumor cell of origin fraction. Because the somatic mutations are only present in the tumor cells, no adjustment is really needed and indeed you would not want to adjust for the tumor-cell of origin fraction. That the MS1 load does not correlate well with the tumor cell of origin fraction in many cancer-types probably indicates that the mutation load itself, being a digital measurement, is not sensitive to the tumor purity (as long as the tumor purity is sufficiently large). Alternatively, it may indicate that MS1 is a less sensitive marker of mitotic age. To summarize, the comparison of the correlation of stemTOC and MS1-load to tumor cell of origin fraction is fair, because neither measure needs to be adjusted (and should NOT be adjusted) for tumor cell of origin fraction.

Heatmaps in fig 1b: How are the 629 mitCpGs (in columns) ordered? Also, there is no description of the adjacent curves. What do they mean?

Response: We thank the reviewer for asking this and apologize for the confusion. To clarify, the CpGs are depicted in the columns, whereas the rows are labelling the samples. What we are ordering here are the samples, not CpGs! The adjacent curves represent the ordered ages of the samples, i.e. samples are being ordered according to their age, so that the older samples appear at the bottom. The CpGs labelling the columns have only been clustered using hierarchical clustering, so the ordering of the CpGs/columns reflects this clustering. In response to the reviewer’s point we have now clarified this in the Figure legend.

Reviewer #3

General Comments:

Comment: Zhu et al. presented an epigenetic counter of mitotic age and applied it to cancer and pre-cancerous tissues. The new clock is qualitatively similar to the previous mitotic clocks from the same authors.

Response: We would like to thank the reviewer for taking time to evaluate our MS and for all the feedback provided.

Comment: The result interpretation on the cell of origin tracking seems misleading. The presented data only suggests that stemTOC associates tumor purity, which can be achieved by any tumor-specific measures.

Response: We fully understand the reviewer's point but don't share the reviewer's negative interpretation that it is "misleading". We agree with the reviewer that the estimated tumor cell of origin fraction will correlate with tumor purity. Indeed, this was one of the key results of our previous EpiSCORE DNAm-atlas Zhu T et al Nat Methods 2022 paper, where we showed, across all major TCGA cancer-types, that the tumor cell of origin fraction, as estimated using EpiSCORE, correlated reasonably well with tumor purity estimates obtained with other methods (e.g. CPE, ABSOLUTE, ESTIMATE, LUMP, IHC, see Aran D et al Nat Commun 2015 for definition of these tumor purity measures). In this regard though, it is important to point out to the reviewer that using the EpiSCORE DNAm-atlas to directly estimate tumor cell of origin fraction has clear advantages over using a tumor purity measure like ESTIMATE (gene-expression based) or ABSOLUTE (CNV-based): for instance, tumor purity obtained using ESTIMATE or ABSOLUTE would not be able to distinguish liver hepatocellular from liver cholangiocarcinomas, but EpiSCORE would be able to easily distinguish them based on estimated hepatocyte and cholangiocyte fractions, as indeed shown in our Zhu T et al Nat Methods 2022 paper. Now, the real significance of Fig.3 is not in using EpiSCORE's tumor cell of origin fraction as an improved estimate of tumor purity, but in demonstrating that the mitotic age of a tumor correlates so well with the tumor cell of origin fraction or, as the reviewer likes to put it, tumor purity. Although the result shown in Fig.3 is biologically intuitive, no one has actually previously demonstrated this correlation between mitotic age and tumor purity. Indeed, the reviewer must agree that this is in fact an excellent way to test mitotic counters. Hence we can view Fig.3 as an important *in-vivo* validation of stemTOC in the tumor context. Furthermore, the downstream significance of Fig.3's result becomes much more apparent in the context of normal tissues exposed to cancer risk factors, as shown in Fig.4b, where "tumor purity" is not easily defined or measured. Instead, with our EpiSCORE DNAm-atlas we can easily estimate the fractions of each individual cell-type and detect subtle increases in the mitotic age of that cell-type, as shown in the case of buccal swab epithelial cells from smokers.

In response to the reviewer's excellent point, we have now changed the title of the corresponding subsection, avoiding the term "tracking" and simply stating that stemTOC correlates with the tumor cell of origin fraction. We also explain that in effect we are measuring tumor purity, albeit in a more meaningful way, as explained above. To more explicitly demonstrate this, we have now also correlated stemTOC's mitotic age to tumor purity estimates obtained using different methods including CPE, ABSOLUTE, ESTIMATE, LUMP and IHC (see Systematic pan-cancer analysis of tumour purity. Aran D, Sirota M, Butte AJ. Nat Commun. 2015 Dec 4;6:8971, for a description of these tumor purity estimation methods), and the result is shown below for the reviewer's convenience:

From this figure, we can see that stemTOC's mitotic age Pearson correlation (PCC) with tumor purity is significantly improved by using EpiSCORE's tumor cell of origin fraction compared to immunohistochemistry (IHC, paired Wilcox test $P < 0.01$) and a gene-expression based method (ESTIMATE, paired Wilcox test $P < 0.01$). The above figure has been incorporated into a new SI fig.S10c.

Comment: Is stemTOC a real clock? Despite the claim, stemTOC readout does not give mitotic age. stemTOC gives a numeric from 0 to 1. It perhaps correlates with mitotic age (as many other chronological clocks would also do). But the actual projection to the mitotic age is missing as all evidence supporting the "mitotic" nature is provided indirectly. This is in contrast with the chronological clocks (e.g, the Horvath clock) which does predict time in year.

Response: We agree and in the title of the MS we used the term "counter", not clock. Admittedly though, throughout the MS, we used the two terms interchangeably, because the term "epigenetic clock" is now being used very extensively in the community, even in the context of epigenetic clocks that are not aimed at predicting chronological age but a form biological age (e.g. the GrimAge-clock). In other words, the community is using the term "clock" to also include relatively weak predictors of biological age. Nevertheless, we agree with the reviewer that other terms like "counter" are more appropriate, and so in response to the reviewer's point, in the revised version we now use the term "counter" as much as possible.

With regards to “the actual projection to mitotic age missing”, the reviewer should bear in mind that in Fig.1a-b we outline the strategy by which the stemTOC CpGs are selected, and this includes re-analysis of the Endicott et al Nat Commun 2022 cell-line data, where cell-lines were treated with cell-cycle inhibitors, precisely to deconvolve the effects of chronological and mitotic age. Thus, stemTOC is similar to RepliTali in avoiding as much as it is possible, the confounding effect of chronological age, although stemTOC also differs markedly from RepliTali in the set of CpGs from which we build the counter. Whilst in our previous epiTOC2 work, we did try to estimate the total number of stem-cell divisions, this is not the focus of this study, and importantly it is also not needed, as a relative measure of mitotic age is sufficient to demonstrate potential for cancer risk prediction. In response to the reviewer’s point, we have now improved the clarity of the stemTOC’s feature selection procedure in Fig.1a-b (see below), as well as in the main text of the Results section:

Construction of the stochastic epigenetic mitotic timer of cancer (stemTOC)

New Figure-1a-b: Construction of stemTOC and estimation of mitotic age. *a)* We first identify CpGs ($n=30,257$) mapping to within 200bp upstream of the TSS of genes that are unmethylated (defined by DNAm beta-value < 0.2) across 86 fetal-tissue samples from 13 different fetal-tissues (including post-natal cord-blood tissue). These are then filtered further by the requirements that they display hypermethylation as a function of population doublings (PDs) in 6 cell-lines representing fibroblasts, endothelial and smooth muscle cell-types. To avoid confounding by passage of time/chronological age, we also demand that they don’t display such hypermethylation when cell-lines are deprived from growth-promoting serum or when treated with mitomycin (MMC, a cell-cycle inhibitor), resulting in 629 “vitro-mitCpGs” satisfying these conditions. *b)* To exclude cell-culture specific effects, we then ask if DNAm at each of these 629 CpGs displays significant hypermethylation with chronological age, as assessed in 3 separate whole blood cohorts, after adjusting for variations in 12 immune-cell type fractions. Because blood has a high rate of stem-cell divisions, correlations with chronological age will capture the subset of vitro-mitCpGs that also change with cell-division in-vivo. In the heatmaps, rows label samples, which have been ordered according to increasing age, so that older samples appear towards the bottom of the heatmap. Columns label the CpGs which have been ordered according to hierarchical clustering. This meta-analysis over the 3 separate whole blood cohorts results in 371 “vivo-mitCpGs”.

Comment: Overall, the model assumed constant methylation change over mitosis, among cell lineages and developmental stage. No evidence is provided to support this key assumption. In fact, constant methylation change over mitosis is unlikely and would be affected by the turnover rate, different biochemical environments (DNMTs/TETs) etc.

Response: We appreciate the reviewer raising these important points. According to Occam's Razor (or the principle of parsimony), we should only dismiss simple models if they fail to describe salient features of the data. In this particular instance, we already demonstrated in our previous publication (Teschendorff Genome Med 2020) that the epiTOC2 estimates correlate reasonably well with the average lifetime intrinsic rates of stem-cell division as published in the literature (using a broad categorization of tissues into those with high, medium and low rates of stem-cell division). In epiTOC2, as well as in stemTOC, we also assumed that different loci may have different probabilities of undergoing DNAm-changes per cell-division step, but that these probabilities per cell-division step are largely independent of tissue and cell-type, with ensuing DNAm differences between cell-types and tissues arising because of the different cell and tissue-specific turnover and stem-cell division rates, as well as tissue-specific exposures and biochemical environments. Thus, although we acknowledge that we have a simple model, it is a sensible model that, so far, works reasonably well. In order to improve upon these models we first need improvements in the estimated stem-cell division and turnover rates of tissues, as well as their population sizes etc. But even from one of the recent state-of-the-art publications (see Sender R et al *The distribution of cellular turnover in the human body, Nat Med 2021*), we can see that for most tissues and organs in the human body (e.g. kidney, lung, liver) we know relatively little. We also agree that in future it would be interesting to incorporate e.g. expression data on DNMTs/TETs, but this would go well beyond the scope of this manuscript and it is unclear in our mind whether this would be necessarily helpful in the pragmatic context of developing cancer risk prediction strategies. In response to the reviewer's point we have now included a paragraph in Discussion describing the limitations of our stemTOC model.

Comment: The authors claimed that the ticking of the clock is only due to division of stem cells (in contrast to a combined effect including expansions?). But no direct evidence is given to support this claim (hence the name stemTOC). Only a heavily confounded analysis (see comment below) of lifetime stem cell division rate was used to show correlation with tumor stemTOC.

Response: The reviewer has raised an excellent point. Let us first clarify that "st" in stemTOC stands for "stochastic" not "stem-cell". Indeed, in the Introduction we wrote "Here we build and validate a pan-tissue epigenetic counter of mitotic age called stemTOC (Stochastic Epigenetic Mitotic Timer of Cancer)". Thus, we were not claiming that stemTOC is only measuring the cumulative number of stem-cell divisions. However, we acknowledge that we had inadvertently caused confusion, as we had used the terms stem-cell division and turnover rates interchangeably, when these are indeed distinct, as correctly pointed out by the reviewer. Reason we had used both terms interchangeably is precisely to

indicate that stemTOC is measuring a combined or total effect from cumulative stem-cell divisions as well as the effect from the more temporary and dynamic effects of amplifying progenitor expansions. In other words, stemTOC is not only measuring the underlying cumulative number of stem-cell divisions, but also the divisions associated with more temporary progenitor expansions. It would be a major task that would go beyond the scope of this MS, to dissect the two, although we note that taking the upper-quantile over the 371 CpGs as an estimate of total mitotic-age may in fact help capture the effect of such progenitor expansions. Taking an upper-quantile over the 371 stemTOC CpG DNAm values would also be important to better capture stochastic subclonal expansions that we know exist in normal aging tissues (as these have been shown to be genetic and epigenetic mosaics), or in the precancer or cancer context, as these expansions in question could reflect the emergence of dominant subclones. Indeed, in SI fig.S18, we show that taking the upper quantile improves discrimination accuracy. In response to the reviewer's point, we have now extensively revised our terminology, explaining in both Introduction and Discussion that we are measuring a "total mitotic age" that not only reflects the cumulative number of stem-cell divisions but also the more recent temporary divisions of progenitor expansions.

Comment: The 371 vivo-mitCpGs was unmethylated in 86 fetal samples of different tissues but defined for methylation gain based solely on blood data. Whether these CG methylation tracks mitotic age in other cell types remain unvalidated.

Response: Based on the reviewer's comment, we fear that the reviewer may have misunderstood the stemTOC feature selection procedure, as outlined in Fig.1a-b. Indeed, it should be clear from Fig.1a-b, as well as from the Results and Methods parts, that the 371 CpGs of stemTOC were selected to undergo hypermethylation with population doublings in 6 separate cell-lines (AG06561, AG11182, AG11546, AG16146, AG21837, AD21859) (Fig.1a), and that only in a secondary step they were also required to undergo hypermethylation with age in whole blood tissue (Fig.1b). The 6 cell-lines are representative of different cell-types including various types of fibroblast, endothelial cell and smooth muscle, suggesting that the stemTOC CpGs change with population doublings in a largely cell-type independent manner. Moreover, we also required the stemTOC CpGs to *not* display DNAm changes with time when a cell-line is treated with a cell-cycle inhibitor or when placed in serum-free culture, thus allowing dissection of DNAm changes associated with chronological age from those associated with cell-division. We note that this is the cell-line data and strategy proposed by Endicott et al Nat Comm 2022. In contrast to Endicott et al, we require that the 371 stemTOC CpGs simultaneously undergo hypermethylation with cell-divisions in-vitro across 6 different cell-lines and in three large in-vivo whole blood cohorts (after adjustment for 12 immune-cell fractions), which very likely selects for CpGs that change with cell-division in immune-cells too, specially given that blood stem-cells (LT-HSCs + ST-HSCs) divide quite frequently. Moreover, our very initial selection to focus on CpGs mapping to within 200bp of the TSS of genes that are constitutively unmethylated across so many different fetal tissue-types selects for CpGs that display age-associated hypermethylation *independent* of tissue-type, as

demonstrated by Neiman and Howard Cedar's beautiful work published in Cancer Res 2014, as explained at the beginning of the Results section.

In response to the reviewer's point, we have now revised Fig1a-b to make it clearer that stemTOC CpGs are being selected to undergo gains in DNAm not only in blood but also in each of 6 different cell-lines. For convenience, we display the revised Fig.1a-b below:

Construction of the stochastic epigenetic mitotic timer of cancer (stemTOC)

New Figure-1a-b: Construction of stemTOC and estimation of mitotic age. **a)** We first identify CpGs (n=30,257) mapping to within 200bp upstream of the TSS of genes that are unmethylated (defined by DNAm beta-value < 0.2) across 86 fetal-tissue samples from 13 different fetal-tissues (including neonatal cord-blood tissue). These are then filtered further by the requirements that they display hypermethylation as a function of population doublings (PDs) in 6 cell-lines representing fibroblasts, endothelial and smooth muscle cell-types. To avoid confounding by passage of time/chronological age, we also demand that they don't display such hypermethylation when cell-lines are deprived from growth-promoting serum or when treated with mitomycin (MMC, a cell-cycle inhibitor), resulting in 629 "vitro-mitCpGs" satisfying these conditions. **b)** To exclude cell-culture specific effects, we then ask if DNAm at each of these 629 CpGs displays significant hypermethylation with chronological age, as assessed in 3 separate whole blood cohorts, after adjusting for variations in 12 immune-cell type fractions. Because blood has a high rate of stem-cell divisions, correlations with chronological age will capture the subset of vitro-mitCpGs that also change with cell-division in-vivo. In the heatmaps, rows label samples, which have been ordered according to increasing age, so that older samples appear towards the bottom of the heatmap. Columns label the CpGs which have been ordered according to hierarchical clustering. This meta-analysis over the 3 separate whole blood cohorts results in 371 "vivo-mitCpGs".

Comment: There is also a significant difference between fetal and postnatal tissues for some lineages.

Response: The reviewer has raised an excellent point. Let us first clarify that it is extremely important for our epigenetic mitotic clock models to use a set of CpGs that are in the same unmethylated state across as many different cell-types as possible in a suitably defined "ground state". We support the view by Howard Cedar (see Neiman D, Cedar H et al Cancer Res 2014) that a suitably defined ground-state is the fetal-stage, and following this work as well as our previous publications on epiTOC1/2 (Yang et al Genome Biol 2016 & Teschendorff Genome Med 2020), we focus on CpGs mapping to within

200bp upstream of the TSS of genes that are constitutively unmethylated across as many fetal tissue types as possible because these sites are enriched for CpGs that are *****not***** involved in the development and differentiation of postnatal tissues. Thus, to address the reviewer's point, we have decided to present data that conclusively shows that the 371 stemTOC CpGs are ***not*** markers of differentiation. First, we show this for neonatal tissue. We note that although at the neonatal stage (i.e. newborns) there will already be intrinsic differences in mitotic age between tissues, that these will be relatively small. Moreover, if stemTOC CpGs truly track mitotic age and are not markers of differentiation, we would expect the overwhelming majority to retain ultra-low DNAm levels (Beta DNAm < 0.2) in neonatal tissue. To this end, we have now obtained and analyzed EPIC DNAm data of buccal swabs from newborns (n=44, GSE229463) and an independent set of cord-blood samples (n=128, GSE195595). We note that buccal swabs and cord-blood are effectively the only two neonatal tissue-types available for this type of analysis. Heatmaps of DNAm-values of our 371 stemTOC CpGs in these two datasets, as well as density distributions of the average DNAm values of these CpGs, are displayed below for the reviewer's convenience.

This clearly shows that the overwhelming majority of our 371 stemTOC CpGs retain low DNAm values (ie < 0.2) in buccal swabs, clearly indicating that they are not markers of differentiation stage in squamous epithelium. Similarly, the very low DNAm-values in the independent cord-blood samples demonstrates the effectiveness of our original selection of 371 stemTOC as being unmethylated in cord-blood. In response to the reviewer's point we have now added the above figure as a new SI fig.S1.

To further demonstrate that the 371 stemTOC CpGs are not markers of differentiation in the context of adult cell and tissue-types, we have performed a similar analysis, computing the fraction of 371 CpGs that display relatively big differences in DNAm between age-matched sorted cell-types and age-matched normal tissue-types using published data from BLUEPRINT (Chen & Soranzo Cell 2016, sorted naiveCD4T-cells, monocytes and neutrophils), Paul D et al (Nat Commun 2016, sorted B-cells, monocytes and CD4T-cells), DNAm-atlas from Moss J et al (Nat Commun 2018, sorted non-immune cell-types) and from eGTEX (Oliva M et al Nat Genet 2023, normal tissue-types). As a benchmark we compare these fractions with those obtained with the solo-CpGs from HypoClock (as we had previously already shown that these solo-CpGs are more likely to be markers of differentiation, see Teschendorff Genome Med 2020). Results, depicted below for the reviewer's convenience, clearly demonstrate that stemTOC CpGs are depleted for differentiation markers, which means that any DNAm differences

between adult cell and tissue-types that arise at these sites are likely due to the cell-type and tissue-type specific differences in mitotic age.

The figure above depicts the fraction of HypoClock (purple) and stemTOC (orange) CpGs that display absolute differences in DNAm between immune cell/tissue-types greater than specific thresholds. Focusing on the immune-cell subsets (panels a+b), we can see that the fraction of stemTOC CpGs that display >0.5 DNAm differences is effectively zero, and that this fraction is clearly much higher for HypoClock. Similar patterns are observed across the multi-tissue eGTEX datasets (panel-c), even for less stringent thresholds. Of note, the only tissue displaying some notable differences for stemTOC is colon, the most proliferative tissue. We note that the age-matched nature of all these datasets is really important as this adjusts for the confounding effect of chronological age, so clearly what this figure therefore demonstrates is that our stemTOC CpGs are not cell-type specific, and that the small differences seen between colon and other tissues is due to the much higher stem-cell division rate of colon. Cell-type specific markers would be displaying >0.5 DNAm changes between cell-types and >0.25 changes between tissue-types (the mixed nature of tissues means that absolute differences in DNAm would be diluted down).

To further demonstrate this, we repeated the analysis for 8 sorted non-immune cell-types (hepatocytes, adipocytes, colon-epithelial, lung epithelial, neurons, pancreas_endocrine, pancreas_exocrine, endothelial) from the Moss et al DNAm atlas (Illumina 450k, Moss J et al Nat Commun 2018), and once again results are unequivocal in demonstrating that stemTOC CpGs are not cell-type specific, and that it is only when comparing colon (the most proliferative tissue) to other much less proliferative tissues, that we see some differences, as expected:

Thus, we conclude from all of this that our stemTOC CpGs are not cell-type specific markers of adult tissues or cell-types. In response to the reviewer's point, we have now added the above two figures as new SI figs.S2-S3.

Comment: In Figure 3b, the cell of origin correspondence is not relevant here since one can replace the X-axis label of all the panels by simply tumor purity (as estimated by EpiSCORE). To really show stemTOC tracking tumor cell of origin (if it does), one should show how one can distinguish tumors of different cells of origin purely based on stemTOC readouts. This analysis is also significantly confounded by the mitotic age of other cell types in the TCGA samples.

Response: We appreciate the reviewer's three points but have to disagree with the last two. We agree

that EpiSCORE's tumor cell of origin fraction correlates with tumor purity and indeed, as mentioned in our response to a previous reviewer's point, this was one of the key results of our EpiSCORE DNAm-Atlas Nat Methods 2022 paper. However, EpiSCORE's tumor cell of origin fraction and tumor purity are not equivalent for the simple reason that using EpiSCORE's tumor cell of origin fraction we would be able to distinguish liver hepatocellular carcinomas from liver cholangiocarcinomas (see Zhu T et al Nat Methods 2022), or basal breast cancers from luminal breast cancers. In contrast to EpiSCORE, there is no clear rationale for discriminating such tumor types using a traditional tumor purity estimate, for instance as obtained using a CNV-based method like ABSOLUTE. Indeed, there is no logical basis why a tumor purity index like ABSOLUTE should discriminate say liver hepatocellular carcinoma from liver cholangiocarcinoma. Hence, we think that it is better to use EpiSCORE's tumor cell of origin fraction on the x-axis of Fig.3b than to use a tumor purity index that is uninformative of the cell of origin e.g. ABSOLUTE CNV-based estimate. Nevertheless, in response to the reviewer's point we have now computed the correlations of stemTOC's mitotic age with other measures of tumor purity, with the results shown in SI fig.S10c.

As far as the second point is concerned, we disagree with the reviewer's suggestion that stemTOC, a pure readout of mitotic age, should distinguish tumors derived from different cells of origin, for the same reason that an mRNA expression-based proliferation index would not be able to distinguish highly proliferative lung cancers from say highly proliferative esophageal cancers. By using EpiSCORE's tumor cell of origin fraction estimates in Fig.3b, we are pushing boundaries by demonstrating an explicit correlation between mitotic age and the tumor cell of origin fraction. Although biologically intuitive, we are not aware of any previous study demonstrating this important result.

Regarding the third and final point: although we agree that the mitotic age estimates in the TCGA samples are an average over the mitotic ages of each cell-type in the tissue, the mitotic age of the cancer cell population likely dominates the average because cancer cells are the most proliferative cells. Indeed, this is the most likely explanation as to why we observe such consistent *positive* correlations between mitotic age and tumor cell of origin fraction in each of 15 TCGA cancer-types. The reviewer's suggestion that these strong positive correlations in each of 15 TCGA cancer-types are somehow the result of confounding by immune or stromal cells seems very implausible to us, specially in light of the fact that these stromal and immune-cells are not proliferating at the rate of cancer cells. In response to the reviewer's point, we have now changed the title of the subsection, removing the term "tracking" which seems to have caused confusion, to simply indicate that there is a correlation, whilst also making it clear that the tumor cell of origin fraction is, in effect, a measure of tumor purity.

Comment: The authors claim that stemTOC addressed the issue of CTH. But from the presented data, stemTOC does not nominate mitotic age of each cell components. Instead, for most of the presented analysis, the authors merely estimate the cell composition (the finding of which is known) and mitotic age of the bulk tissue separately. This is the case for the smoking and NAFLD studies in Fig 4.

Response: We thank the reviewer for raising this important point and the reviewer is absolutely right. In

responding to the reviewer, it would be important to clarify that broadly speaking CTH can confound mitotic-age estimates in two distinct ways. First, it can confound mitotic age estimates if the procedure of selecting clock/counter-CpGs inadvertently selects for cell-type specific markers. For instance, it would be bad if our clock/counter CpGs include cell-type specific markers (i.e. CpGs that display big >50% changes in DNAm between cell-types), because then cell-type compositional changes between two bulk samples would result in their mitotic-ages being different even if all underlying cell-types are of the same mitotic age. The way we build stemTOC advances the field by ensuring that our 371 stemTOC CpGs are not cell-type specific, and therefore that these CpGs are not confounded by CTH. This is what we claimed and what we meant by “stemTOC not being confounded by CTH”. As mentioned earlier, we have now added new SI figs.S1,S2,S3 in support of our claim that the stemTOC CpGs are not cell-type specific markers or markers of differentiation.

Now, there is a second way in which CTH can “cause confounding” and that refers to the actual mitotic-age (or epigenetic clock) estimate obtained in a bulk-tissue. In an unsorted bulk tissue, the mitotic age estimate will be a weighted average over the mitotic ages of each underlying cell-type, with the weights being the underlying cell-type fractions. In other words, even if the CpGs making up the mitotic-counter are not cell-type specific, the mitotic age estimate in a bulk sample will only represent a weighted average over the mitotic ages of each cell-type. At present, although we have explored strategies to develop cell-type specific mitotic clocks, these have not yet been successful enough, partly because we lack the appropriate data to substantially validate such clocks/counters. However, does this type of confounding prevent the use of stemTOC to measure mitotic age in the tumor cell-of-origin? The answer as clearly shown in Fig.3 and Fig.4 is *no*. In other words, the 2nd type of confounding that the reviewer is concerned about does not present a major practical limitation. In response to this excellent point, we have now dedicated a whole paragraph in Discussion to describe the different ways in which CTH can confound or bias mitotic age estimates, and to also emphasize how stemTOC pushes boundaries in avoiding confounding by CTH in the CpG selection procedure.

Comment: Fig 5e: This analysis is founded on rough (and simplistic) estimates of annual stem cell division rates from prior studies. It is known that stem cells do not divide at a constant rate throughout the life span. Within each cell type/cancer type, very little correlation is visible, suggesting mostly a cell type-specific effect. Further, any age-associated metric would manifest an association as age is explicitly formulated in the X-axis.

Response: The reviewer is right that the intrinsic rates of stem-cell division we are using are lifetime averages. This limitation also applies to the Vogelstein/Tomasetti papers, from where these estimates are derived. It is precisely for this reason that plots such as Fig.5e or 5f should be interpreted in terms of tissues belonging to 3 or 5 broad stem-cell division rate categories, as for instance shown in Fig.2g. Within each cancer-type, a correlation between stemTOC and TNSC would be driven purely by age since $TNSC=IR*Age$, and because our IR value only differs between cancer-types and not between samples of a given cancer-type. However, for any cancer sample, its mitotic age will be heavily

influenced by its proliferative potential and much less so by the age at diagnosis, so we would not expect there to be a correlation between stemTOC and IR*Age (TNSC) within cancer-types, except for those whose normal-tissues have high stem-cell division rates. Indeed, for colorectal cancer (COAD) and stomach cancer (STAD), two tissues with the highest stem-cell division rates, we do observe a correlation with TNSC (age): COAD (PCC=0.20, P=0.0004) & STAD (PCC=0.18, P=0.0003). Regarding the final point, in Fig.5f we display the same data as Fig.5e but now collapsing all cancer samples of a given cancer-type into one median value and also adjusting for the effect of chronological age, with the x-axis now representing the annual intrinsic rate of stem-cell division. Thus, the correlation statistics and P-value shown in Fig.5f are automatically adjusted for chronological age. This means that the association with IR is non-trivial.

Minor Point:

Comment: Why were only a subset of TCGA cancers presented? PCC is also missing.

Response: PCCs arise from the adrenal gland and were not included because our EpiSCORE DNAm-atlas does not include a DNAm reference matrix for the adrenal gland, meaning that we can't estimate underlying cell-type fractions for these samples.

Reviewers' Comments:

Reviewer #1:

Remarks to the Author:

I have now reviewed again the manuscript and rebuttal letter provided by Zhu and colleagues. I have noticed that, despite the manuscript being rejected with major concerns, the current manuscript remains rather similar to the previous version. All the main figures remain the same without novel data or analyses. The majority of the points raised by reviewers were addressed with the same data, without novel data, analyses or references. In the fewer points where they present new data or analyses, I do not see compelling biological evidence demonstrating the novelty or superiority of stemTOC vs previous mitotic clocks. In others, the new data seems discrepant with previous published data (discrepant correlations (R2) between epiTOC2 and epiCMIT and PD from Endicott 2022 study, Fig 4 and Supplementary Fig 11 from Endicott manuscript). In other points, the authors directly refused to perform the analyses I proposed, without any clear rationale or explanation, or claiming low number of samples without apparently exploring all data from literature. Finally, despite the authors recognizing in the rebuttal their initial misleading interpretation about the tumor cell of origin, they seem to maintain the same interpretation throughout the current version of the manuscript and highlighted in the title.

Reviewer #2:

Remarks to the Author:

I am satisfied with the clarifications and revisions in response to the concerns of the other two reviewers - as far as I understand them.

However, I have a lingering concern wrt how the authors view the confounding of their mitotic counter with selective clonal processes and subsequent changes in tumor-cell of origin fractions.

To the point: there is considerable evidence in tumor biology that stochastic clonal expansions, in particular in cancer precursors, are more driven by decreases in stem cell terminal differentiation rates and death rather than by increases in the rate of stem cell divisions (i.e. mitotic rates). The authors may consider the idea that the CpGs they identify in the upper 95% tile are basically mitotic in normal tissues, but stand out as being selected across individuals and tissues in precancers and cancers, NOT because they are better markers of mitotic age, but because they lead to subtle quenching of transcription (if not silencing) of otherwise active genes resulting in a survival benefit if not loss of homeostasis. For example, such an effect was conjectured in the Barrett's Esophagus, the precursor to esophageal adenocarcinoma (Luebeck et al. Clin Epi 2017).

The explanation that the select group of StemTOC CpGs in the upper 95% tile is a reflection of clonal and mitotic mosaicism is NOT entirely plausible. It is fraught with the fact that distinct stochastic clonal expansions may well occur due to changes in cell death/differentiation and not due to differences in the number of mitotic events along lineages although it does not preclude the latter. Note, in a homogenous clone, all cells have the same mean mitotic 'length' (cell division number) along their phylogenetic trajectory, although they may differ stochastically. Even if clones differ in their mitotic rates, it cannot be stated that they are necessarily the ones with the highest cancer risks since it is ultimately the net cell proliferation rate that matters and produces tumor bulk. Admittedly, this may come across as semantics, but clarity matters.

A discussion (and perhaps investigation) of the potential selection of a particular subgroup of the 371 stemTOCs, not by changing the mitotic rate and/or tumor cell of origin fraction but by subtle increases

in cell survival and decreased terminal differentiation, perhaps would further the relevance of this particular finding.

Reviewer #3:

Remarks to the Author:

After reviewing the author's response (only my part), I acknowledge some improvements in the manuscript. However, my primary concern, centered on the methodology used to estimate and validate mitotic ages in bulk tumors (hence the significance of reporting any correlation between purity), remains.

The authors claim a correlation between the mitotic age of a tumor and the tumor cell of origin fraction ("Now, the real significance of Fig.3 is not in using EpiSCORE's tumor cell of origin fraction as an improved estimate of tumor purity, but in demonstrating that the mitotic age of a tumor correlates so well with the tumor cell of origin fraction or, as the reviewer likes to put it, tumor purity."). This assertion raises the question: Are they referring to mitotic age estimates derived from pure tumor cells or those based on bulk tumor analysis? Clarification on this point is crucial. Based on the presented results, it seems the latter is true – the estimates are made on bulk tumors. If so, reporting a correlation between the bulk tumor's attenuated mitotic age and the tumor cell of origin fraction (essentially tumor purity) appears trivial. Since tumor cells inherently have a high mitotic age, and the bulk tumor tissue's mitotic age is lowered due to tumor purity, this correlation seems expected.

The authors differentiate between tumor cell of origin fractions and tumor purity. They illustrate this with the example of distinguishing liver hepatocellular carcinomas from liver cholangiocarcinomas using EpiSCORE's tumor cell of origin fraction ("However, EpiSCORE's tumor cell of origin fraction and tumor purity are not equivalent for the simple reason that using EpiSCORE's tumor cell of origin fraction we would be able to distinguish liver hepatocellular carcinomas from liver cholangiocarcinomas"). However, I find this mixing up of tumor cell of origin and tumor purity unclear and disagree that there is any ambiguity here. Tumor purity, in my understanding, should be defined as the proportion of tumor cells, all of which originate from a single specific cell of origin through clonal expansion. If the authors are applying a unique definition of tumor purity in their study (such as classifying cholangiocarcinoma purity based on hepatocyte fraction, which doesn't make sense anyway), this needs to be explicitly stated and justified.

Detailed Response to Reviewer Points:

Reviewer #1 (Remarks to the Author):

Comment: I have now reviewed again the manuscript and rebuttal letter provided by Zhu and colleagues. I have noticed that, despite the manuscript being rejected with major concerns, the current manuscript remains rather similar to the previous version. All the main figures remain the same without novel data or analyses.

Response: We thank the reviewer for taking time to re-evaluate our MS but strongly disagree with the statement that we did not provide novel data or analyses. The fact is, we analyzed an additional 19 DNAm datasets, encompassing over 14,500 samples, adding 10 new Supplementary Figures (figs.S1, S2, S3, S7, S8, S9, S10c, oldS12 (newS13), oldS13 (newS14) and oldS14(newS15)) to address many of the reviewer's key concerns. We also made changes to Fig.1 and Fig.2 to further clarify key points which had caused some confusion. Whilst we acknowledge that the format of the main figures did not change, this only reflects the nature of the criticism raised by the reviewers, which in our opinion did not warrant us to make changes to the main figures. However, we are certainly happy to increase the number of main figures if the editor so wishes, by turning some of the Supplementary Figures into main figures. It is our opinion however, that the points raised by the reviewer were largely technical in nature, and that these technical points are best addressed with Supplementary Figure items, so as to maintain the logical flow of the MS.

Comment: The majority of the points raised by reviewers were addressed with the same data, without novel data, analyses or references.

Response: We respectfully disagree. In the revised version of the paper we analyzed an additional 19 DNAm datasets, encompassing over 14,500 samples, adding 10 new Supplementary Figures (figs.S1, S2, S3, S7, S8, S9, S10c, oldS12, oldS13 and oldS14), addressing many of the key issues raised by the reviewer. Alongside the addition of these datasets we have added corresponding references to the associated datasets.

Comment: In the fewer points where they present new data or analyses, I do not see compelling biological evidence demonstrating the novelty or superiority of stemTOC vs previous mitotic clocks.

Response: The new data and analyses were primarily aimed at addressing the other concerns from this reviewer. As far as the direct comparison of stemTOC to the other clocks is concerned, the one key novel biological insight we gain from this MS is that clocks based on CpGs that gain DNAm with cell-division ("hypermethylated clocks") outperform those based on CpGs that lose

DNAm with cell-division (“hypomethylated clocks”). The evidence for this is compelling. Once again, we display a figure below summarizing all the lines of evidence, clearly supporting this finding:

For the 3 heatmap figures above displaying paired Wilcoxon rank sum test P-values, comparing each of the 7 mitotic counters to each other, the convention is that a significant P-value means that the counter labelling the row displays stronger correlations than the counter labelled by the column. Thus, overall, the evidence that clocks based on hypomethylated CpGs (specially HypoClock and epiCMIT-hypo) perform worse than hypermethylated ones is strong in the context of (1) correlations with independent stem-cell division rates in normal tissue (Fig.2h), (2) correlations with tumor cell of origin fraction (Fig.3e), correlations with chronological age in normal tissues with high or moderate stem-cell division rates (SI Fig.S5c & SI Fig.S6c), as well as in predicting tumor cell of origin (SuppFig.10d). The heatmaps shown in Fig.3d, SuppFig.5b, SuppFig.6b further attest that the improved performance of hypermethylated clocks is robust across studies. As far as the comparison of stemTOC to the other hypermethylated mitotic clocks is concerned, we agree that the performance is only marginally better. Whilst this may seem disappointing, the benchmarking or comparison to existing hypermethylated clocks is not an essential or critical part of this MS. The global comparison of hypermethylated vs hypomethylated clocks does however push boundaries in the field, clearly demonstrating for the first time, that hypermethylated clocks perform better in the context of cancer risk prediction.

We further note that in this day and age, honest methodological comparisons in bioinformatics often demonstrate that there is no clear “winner”, which only reflects the complexity of these analyses. As a concrete example, Saelens W et al Nat Biotech compared single-cell lineage-trajectory inference algorithms, reporting that no single method ever performs better than others. Purpose of mentioning

this here is that any methodological comparison is ***valuable*** regardless of whether it points to a winning method or not. Hence, we do not agree with this reviewer dismissing our work on grounds that stemTOC is not conclusively better than epiTOC2 or epiCMIT-hyper. The insights gained by our comparative analysis are (1) that “hypermethylated” clocks outperform “hypomethylated” ones, and (2) that the different procedures and methods used to build the various “hypermethylated” clocks lead to similar performance on real datasets, which is very valuable knowledge, giving us hints as to how we should proceed in future to improve these mitotic clocks.

Comment: In others, the new data seems discrepant with previous published data (discrepant correlations (R2) between epiTOC2 and epiCMIT and PD from Endicott 2022 study, Fig 4 and Supplementary Fig 11 from Endicott manuscript).

Response: Concerning the Endicott et al paper, there is absolutely no conflict or discrepancy whatsoever. First of all, figure Fig.4b,c,d of Endicott et al demonstrate that the hypermethylated clock epiTOC2 also does reasonably well. In fact, I personally corresponded with Peter Laird and Hui Shen and showed them the data below, and they agreed to include epiTOC2 in Fig.4b,c,d of their paper whilst their work was still a preprint on bioRxiv. Indeed, we re-analyzed Endicott et al’s cell-line data very extensively using our own normalization procedure and below is the side-by-side comparison of epiTOC2 to epiCMIT in the 6 cell-lines analyzed here, which I shared with Peter Laird and Hui Shen:

As the reviewer will see, the correlations of epiTOC2 with population doublings (PD) are excellent, whilst epiCMIT fails to correlate in one of the cell-lines, and displays lower R^2 values in 4 of the 6 cell-lines. **Any differences in R^2 values between this figure and Fig.4 of Endicott et al are likely due to the different normalization procedures used, as Sesame (the method used by Peter Laird and Hui Shen) is overly stringent throwing away lots of CpGs that are actually part of EpiTOC2 and EpiTOC1.**

Second, Endicott et al did not perform an extensive and comprehensive comparison of hyper and hypo-clocks in the context of correlations with chronological age, nor with tumor cell of origin, nor with precancerous states, and it is precisely in these contexts where we see the hypermethylated clocks do much better, so again there is absolutely no contradiction with Endicott et al.

Comment: In other points, the authors directly refused to perform the analyses I proposed, without any clear rationale or explanation, or claiming low number of samples without apparently exploring all data from literature.

Response: The reviewer raised a very large number of points (41 points) and we addressed 38 of them by adding new data or following the reviewer's request. For a few other points we provided a clear scientific rationale why the specific suggestions of the reviewer were not helpful. First, the reviewer wanted us to include the combined epiCMIT-score alongside its components epiCMIT-hyper and epiCMIT-hypo, and yet we provided a very clear and scientifically sound explanation why epiCMIT is not a valid clock. Once again, we opine that epiCMIT violates a central dogma of machine-learning science: the same rule and same set of CpGs should be applied to every sample. However, epiCMIT chooses a different set of CpGs depending on the sample based on an arbitrary criterion that could be influenced by a plethora of confounding factors. epiCMIT computes two separates scores evaluated over two different mutually exclusive sets of CpGs (epiCMIT-hyper & epiCMIT-hypo) and only after computing these two scores, it then decides which one to assign to a given sample. To put this in context, imagine taking Horvath's clock, and computing different age-estimates say by choosing different subsets of CpGs, and then selecting the estimate that best fits your data. As such, we opine that epiCMIT suffers from "selection bias". Confirming this selection bias, epiCMIT-hypo underperforms, very clearly so, in all the analysis in this work. In any case, displaying epiCMIT-hyper and epiCMIT-hypo separately makes more sense because (1) of the huge difference in performance between them, which needs emphasizing, and (2) also because epiCMIT-hyper outperforms epiCMIT, as shown by our new analysis below:

a) Normal-Adjacent Tissue (TCGA)

b) Sorted Immune Cells

c)

Correlations of mitotic-age with cell-of-origin fraction

d)

New SuppFig.S11: a) Pearson Correlation Coefficient (PCC) heatmap of correlations by 4 mitotic clocks (stemTOC, epiCMIT-hyper, epiCMIT-hypo, epiCMIT) with chronological in the normal-adjacent samples of TCGA cancer-types. **b)** As a) but for sorted immune-cell datasets. **c)** As a) but now for correlations of the mitotic clocks with tumor cell of origin fraction (as a proxy for tumor purity). **d)** Wilcoxon rank sum paired test one-tailed P-values comparing the PCC values displayed in c) of one clock to another. Convention is that a significant P-value means that the mitotic clock labelled by the row outperforms the one displayed in the column.

Indeed, as we can see from panel-d), stemTOC outperforms epiCMIT-hyper and epiCMIT, and epiCMIT-hyper outperforms epiCMIT. That epiCMIT-hyper outperforms epiCMIT is also clear from panels a)+b). Thus, it makes more sense for us to present the results for epiCMIT-hyper, because

epiCMIT-hyper performs better than epiCMIT. Nevertheless, we have included the above figure into the revised MS as a new SuppFig.S11.

Concerning the sorted data from the Nat Cancer 2020 paper, we have re-checked the sample numbers of the sorted normal naïve and memory B-cell datasets (the ones proposed by the reviewer that we should analyze) and these were indeed very low (n=10 peripheral blood naïve B-cells (NBC) and 10 PB memory B-cells (MBCs)). For everyone's convenience, we display the corresponding figure panel from the paper below:

Panel a from Duran-Ferrer M et al Nat Cancer 2020 paper: PB=peripheral blood, NBC=naïve B-cell, MBC=memory B-cell.

Age-range of these samples is also not specified, so in our opinion it is futile to attempt to analyze this type of data in relation to the data presented in Fig.2c where we used the much larger BLUEPRINT dataset encompassing 139 naïve CD4T-cells, 139 monocyte and 139 neutrophil samples from individuals encompassing a wide age-range:

Fig2c-d: c) Scatterplots of stemTOC's mitotic age (y-axis) vs chronological age (x-axis), for three sorted immune-cell populations as profiled by BLUEPRINT. Number of samples is given above

each plot. The Pearson Correlation Coefficient (PCC) and P-value from a linear regression is given. **d) Heatmap of PCC values between mitotic and chronological age for 7 mitotic clocks across 9 independent sorted immune-cell populations. Sorted immune-cell samples labeled by cell-type and study it derives from. BP=BLUEPRINT.**

Surely, using a larger high quality dataset from the BLUEPRINT consortium encompassing 139 sorted naïve CD4T cells, 139 monocyte and 139 neutrophil samples should be much more meaningful than analyzing a dataset of only 10 naive B-cell samples. Indeed, what is the point of analyzing such a small dataset if power calculations indicate that any false negative result is more likely down to a lack of power or conversely that the false positive rate is so high that any positive result is unreliable?

The reviewer then states that we did not explore the literature, **but panel-d) above clearly shows that we analysed 9 independent sorted immune-cell datasets, most of these containing over 100 samples or close to 100 samples.**

Comment: Finally, despite the authors recognizing in the rebuttal their initial misleading interpretation about the tumor cell of origin, they seem to maintain the same interpretation throughout the current version of the manuscript and highlighted in the title.

Response: We respectfully disagree. We actually felt it was crystal clear throughout, that the EpiSCORE's tumor cell of origin fraction is a measure of tumor purity as indeed we had already shown this in our Nat Methods 2022 paper. In our previous rebuttal letter we never recognized any misleading interpretation, but emphasized the point above that EpiSCORE's cell of origin fraction is a measure of tumor purity. In any case, whether in Fig.3 we use EpiSCORE or another measure of tumor purity is largely irrelevant, because Fig.3 is meant to demonstrate a validation and benchmarking of stemTOC on real data. We sincerely feel that this reviewer may be unfamiliar with the cell-type deconvolution field and has not read our Nat Methods 2022 paper. Nevertheless, we have now made further changes to the title of the MS as well as one of the subsections, removing the wording "tumor cell of origin fraction".

Reviewer #2 (Remarks to the Author):

Comment: I am satisfied with the clarifications and revisions in response to the concerns of the other two reviewers - as far as I understand them.

Response: We thank the reviewer for re-evaluating our MS and specially also for pointing out that we have satisfactorily addressed the points from the other two reviewers.

Comment: However, I have a lingering concern wrt how the authors view the confounding of their mitotic counter with selective clonal processes and subsequent changes in tumor-cell of origin fractions. To the point: there is considerable evidence in tumor biology that stochastic clonal expansions, in particular in cancer precursors, are more driven by decreases in stem cell terminal differentiation rates and death rather than by increases in the rate of stem cell divisions (i.e. mitotic rates). The authors may consider the idea that the CpGs they identify in the upper 95% tile are basically mitotic in normal tissues, but stand out as being selected across individuals and tissues in precancers and cancers, NOT because they are better markers of mitotic age, but because they lead to subtle quenching of transcription (if not silencing) of otherwise active genes resulting in a survival benefit if not loss of homeostasis. For example, such an effect was conjectured in the Barrett's Esophagus, the precursor to esophageal adenocarcinoma (Luebeck et al. Clin Epi 2017)

Response: We thank the reviewer for raising this excellent point. We agree with what the reviewer is indicating, that the preneoplastic clone more likely to turn cancerous is the one characterized by silencing of key developmental genes, as pointed out by Luebeck et al Clin Epi 2017. Indeed, our own work (Teschendorff et al Genome Med 2016) where we showed that this epigenetically associated silencing of tissue-specific genes is not unique to any given cancer-type but a hallmark shared by many tumor-types, further supports this. As such, we agree that taking the upper-quantile of the mitotic-CpGs is not necessarily capturing the clone with higher mitotic age. However, given that evidence points towards (1) the promoters of these developmental genes acquiring DNA hypermethylation in cancer and precancerous lesions as well as aged normal tissue, and (2) that these same loci are also prone to acquire DNA hypermethylation as a consequence of cell-division, this would suggest that the preneoplastic clone at highest cancer-risk is also potentially the one that has undergone most cell-division changes. In response to this point, we have now added one more sentence in Discussion to explain the reviewer's point.

Comment: The explanation that the select group of StemTOC CpGs in the upper 95% tile is a reflection of clonal and mitotic mosaicism is NOT entirely plausible. It is fraught with the fact that distinct stochastic clonal expansions may well occur due to changes in cell death/differentiation and not due to differences in the number of mitotic events along lineages although it does not

preclude the latter. Note, in a homogenous clone, all cells have the same mean mitotic 'length' (cell division number) along their phylogenetic trajectory, although they may differ stochastically. Even if clones differ in their mitotic rates, it cannot be stated that they are necessarily the ones with the highest cancer risks since it is ultimately the net cell proliferation rate that matters and produces tumor bulk. Admittedly, this may come across as semantics, but clarity matters. A discussion (and perhaps investigation) of the potential selection of a particular subgroup of the 371 stemTOCs, not by changing the mitotic rate and/or tumor cell of origin fraction but by subtle increases in cell survival and decreased terminal differentiation, perhaps would further the relevance of this particular finding.

Response: As above, we agree with the reviewer that taking the upper-quantile of the 371 stemTOC CpGs may well be capturing the subclone that is least differentiated. Indeed, our group has already provided preliminary single-cell RNA-Seq data in support of this, i.e that the least differentiated preneoplastic cells are the most likely ones to give rise to cancer (Liu T et al Cancer Res 2022, PMID: 35536873 DOI: 10.1158/0008-5472.CAN-22-0668). The reviewer's suggestion to further explore the 371 stemTOC CpGs to see if a particular subset may mark e.g. developmental genes that undergo silencing in precancerous lesions is a very interesting one. However, the 371 stemTOC CpGs were selected to be cell-type independent in the fetal stage. Consequently, this subset is not really enriched for developmental TFs which may already display DNAm differences across different fetal tissue-types. In any case, exploring this hypothesis that specific developmental TFs are hypermethylated in precancerous lesions is a somewhat separate question to the one we are exploring here. We firmly believe though that those preneoplastic clones that display the silencing of tissue-specific TFs are also more likely to be those display a higher mitotic age, because the silencing of these tissue-specific TFs appears to be mostly mediated by DNA hypermethylation and these specific loci are certainly also those that undergo DNAm gains following cell-division. We would love to write a lengthy Discussion on this, but this MS is no longer the appropriate place to do this.

Reviewer #3 (Remarks to the Author):

Comment: After reviewing the author's response (only my part), I acknowledge some improvements in the manuscript. However, my primary concern, centered on the methodology used to estimate and validate mitotic ages in bulk tumors (hence the significance of reporting any correlation between purity), remains.

Response: We thank the reviewer for taking time again to re-evaluate our MS and for acknowledging that the MS has improved. Below we respond in detail to the reviewer's remaining concern. Here, we just wanted to point out that we have made changes to the title, abstract and also changed the title of one of the subsections to remove the wording "tumor cell of origin fraction" as this seems to have caused confusion.

Comment: The authors claim a correlation between the mitotic age of a tumor and the tumor cell of origin fraction ("Now, the real significance of Fig.3 is not in using EpiSCORE's tumor cell of origin fraction as an improved estimate of tumor purity, but in demonstrating that the mitotic age of a tumor correlates so well with the tumor cell of origin fraction or, as the reviewer likes to put it, tumor purity."). This assertion raises the question: Are they referring to mitotic age estimates derived from pure tumor cells or those based on bulk tumor analysis? Clarification on this point is crucial. Based on the presented results, it seems the latter is true – the estimates are made on bulk tumors. If so, reporting a correlation between the bulk tumor's attenuated mitotic age and the tumor cell of origin fraction (essentially tumor purity) appears trivial. Since tumor cells inherently have a high mitotic age, and the bulk tumor tissue's mitotic age is lowered due to tumor purity, this correlation seems expected.

Response: The reviewer is right that the mitotic age estimate is derived from bulk-tissue tumors, and that the observed correlation between mitotic age and tumor cell of origin fraction (or tumor purity) is likely the result of the increased mitotic age of the tumor cells relative to the surrounding stroma, so that tumors with higher purity should on average display a higher mitotic age. However, we wholeheartedly disagree with the reviewer in calling this "a trivial result". Consider for sake of argument a kidney cancer (sample-1) that has 90% purity but with tumor cells displaying a mitotic age of 0.5 (defined on a normalized scale between 0 and 1), with the surrounding stroma having a mitotic age of 0. Consider now another kidney cancer (sample-2) that has only 70% purity but with a mitotic age score of 0.9, again with surrounding stroma having a value of 0. Tumor sample-1 would then have a mitotic age of $0.9 \times 0.5 = 0.45$, whilst Tumor sample-2 would have a mitotic age score of $0.7 \times 0.9 = 0.63$. In other words, Tumor sample-1 would have a lower mitotic age than Tumor sample-2 despite the former having a much higher purity. This simple "Gedankenexperiment"

clearly demonstrates that the observed correlation between mitotic age and tumor purity is not automatic or trivial.

Nevertheless, on average, one would expect a correlation between the two measures, and as we tried to explain in our previous response letter, testing for this expected correlation between mitotic age and tumor purity is a powerful way to compare mitotic clocks to each other, and so **Fig.3 should be interpreted as a further validation of stemTOC.** Indeed, it is critically important that different methods (in our case these are the different mitotic clocks) should be compared in real-data scenarios where we can objectively define a “ground truth”. **In this particular instance, the ground truth is precisely the expected correlation between mitotic age and tumor purity.**

Another way to understand this: assume for a moment that stemTOC’s mitotic age did not correlate with tumor purity. This would then be a strong indication that stemTOC is a failure, because, on average, mitotic age should be higher in tumor samples that contain more tumor cells, compared to those with fewer tumor cells.

Hence, the purpose of Fig.3 is to further validate stemTOC and benchmark it against other mitotic clocks. The important biological insights are then shown in Fig.4b-d, where we demonstrate that stemTOC’s sensitivity is high enough to detect subtle increases in mitotic age within the normal lung tissue and normal buccal swabs of smokers and that this mitotic age is also higher in samples with a higher squamous epithelial content. Once again, let us emphasize here that in this application to normal-tissue, we can’t define a “tumor purity” index using traditional methods like ABSOLUTE (these sample do not have genotype or CNV data), but we can use EpiSCORE’s cell-type fraction estimates, highlighting one key advantage of EpiSCORE over conventional “tumor purity indices”.

Last but not least, the reviewer’s comment suggests that he/she wants us to develop a cell-type specific mitotic clock in order for this work to be publishable in Nature Communications. We think that this is unrealistic because it is already very difficult to validate a mitotic clock in real human datasets, and to do so at the level of individual cell-types would be a formidable challenge. In fact, we have an ongoing project and grant to develop such cell-type specific mitotic clocks and the current versions we have developed do not outperform stemTOC. In this sense, presenting a cell-type specific mitotic clock may not yet be even necessary, since as demonstrated in Fig.4, stemTOC is already sensitive enough to detect subtle mitotic age increases in preneoplastic lesions as well as in normal-tissues exposed to cancer-risk factors like smoking and obesity.

We would like to end this point with some critically important clarifications regarding cell-type heterogeneity and epigenetic clocks, which may help the reviewer better understand the formidable challenge of developing a cell-type specific mitotic clock. A clock should ideally be built from CpGs that are not cell-type specific in an appropriate ground-state, in other words, these CpGs should not differ between cell-types of the same mitotic age. Because tissues display different mitotic

rates, as they age there is an increased divergence in their mitotic ages, hence the ground-state needs to be chosen early on in development (e.g. fetal or neonatal stage is the best we can do given data availability). stemTOC advances the field by very carefully picking CpGs that satisfy all the requirements, including that they are not cell-type specific. Nevertheless, even if we have a clock that is built from such CpGs, when we compute the clock in a bulk-tissue, the mitotic age is still (inevitably so) an average over the mitotic ages of the underlying cell-types. As mentioned earlier, we are currently working on trying to infer cell-type specific mitotic ages, and whilst we have a number of mathematical models to do this, the resulting estimates are very hard to validate at the level of individual cell-types. We simply do not yet have the experimental data to validate mitotic age at cell-type resolution in real data. Such an endeavour, if overcome, would in our opinion then merit publication in a higher-ranked journal

Comment: The authors differentiate between tumor cell of origin fractions and tumor purity. They illustrate this with the example of distinguishing liver hepatocellular carcinomas from liver cholangiocarcinomas using EpiSCORE's tumor cell of origin fraction ("However, EpiSCORE's tumor cell of origin fraction and tumor purity are not equivalent for the simple reason that using EpiSCORE's tumor cell of origin fraction we would be able to distinguish liver hepatocellular carcinomas from liver cholangiocarcinomas"). However, I find this mixing up of tumor cell of origin and tumor purity unclear and disagree that there is any ambiguity here. Tumor purity, in my understanding, should be defined as the proportion of tumor cells, all of which originate from a single specific cell of origin through clonal expansion. If the authors are applying a unique definition of tumor purity in their study (such as classifying cholangiocarcinoma purity based on hepatocyte fraction, which doesn't make sense anyway), this needs to be explicitly stated and justified.

*Response: We thank the reviewer for raising this point. To clarify, we are certainly *not* computing cholangiocarcinoma purity with the hepatocyte fraction. We are using the estimated cholangiocyte fraction to estimate cholangiocarcinoma purity. And we are using the estimated hepatocyte fraction to estimate hepatocellular carcinoma purity. We agree with the reviewer that using the estimated fraction of the putative cell of origin is not directly equivalent to tumor purity, and yet the two measures should strongly correlate with each other, as indeed already demonstrated by us in our previous Zhu T et al Nat Methods 2022 publication. For the reviewer's convenience, we now display the corresponding panel from this paper below, which demonstrates a fairly good correlation between tumor purity as estimated using EpiSCORE and independent tumor purity estimation methods, including ABSOLUTE and CPE:*

Figure legend: a-b) Systematic validation of the DNAm-atlas tissue-specific DNAm reference matrices in the corresponding Illumina 450k DNAm datasets from the TCGA. **a)** Heatmap depicts the Pearson Correlation Coefficients (PCCs) between the tumor purity, as estimated using our EpiSCORE DNAm-atlas, and the tumor purity estimated with different methods, including ESTIMATE, ABSOLUTE (CNV-based), IHC (immuno-histo-chemistry) and CPE (a method that combines all previous three). **b)** As a), but now for the PCCs between the total immune-cell fraction, as estimated using DNAm-atlas, and the corresponding total immune-cell fraction obtained by other methods (ESTIMATE & LUMP). Number of * in panels a and b indicate statistical significance level (P-value thresholds), as shown in panel-b. P-values were derived from a one-tailed correlation test.

We note that this Nat Methods MS was assessed by 3 independent reviewers, and that they all unanimously accepted this panel as a validation of the EpiSCORE tumor purity estimation procedure. We are hence puzzled why this reviewer keeps dwelling on this issue of tumor purity, when it is also actually not a major point of this paper anyway. Fig.3 is meant to provide further validation of stemTOC, not to highlight novel biological insight. The novel biological insights are shown in Figs.4 and 5.

Finally, we note that the only reason for using EpiSCORE's cell of origin fraction estimate as a measure of tumor purity in Fig.3, and not some other measure of tumor purity, is because later in Fig.4b-d, where we display data for normal-tissues, we have no option but to use EpiSCORE. It therefore makes sense to use EpiSCORE in both Figs.3 and 4.

Reviewers' Comments:

Reviewer #4:

Remarks to the Author:

I have been asked specifically to comment on whether the authors have adequately addressed reviewer comments (specifically Reviewer 1 and 3). I thus assessed: 1) did the authors directly address comments with new data, 2) if not, is it reasonable to simply clarify the methods or reasoning to address the comment, and 3) for points where there remains reasonable disagreement or ambiguity, does the uncertainty compromise the novelty or main point of the paper.

Given the large number of reviewer comments, the answer varies based on which reviewer comment we are talking about. I find the authors' arguments to be compelling concerning two major issues: correlation with chronological age (for both underlying CpGs, and for the overall mitotic clock), and correlation with tumor cell of origin. The same is true for some smaller technical issues. The new figures provided in the Supplement are sufficient, and I do not believe the authors need to provide any further justification or data on these issues. Even where there could be some reasonable disagreement, I do not think they compromise any central claim of the paper.

However, I find two of Reviewer 1's comments to not be adequately addressed, and these are on critical points. First, I am concerned about the broad statements concerning the superiority of hypermethylation vs. hypomethylation counters and fear they are overstated. While I am hesitant to raise an issue that has not yet been raised in review, it is generally related to Reviewer 1's point so it seems pertinent. Some of the key datasets used to demonstrate superiority of hypermethylation – e.g. TCGA, BLUEPRINT- are based on 450K data. As the authors state in Methods, "Of the 87 RepliTali CpGs, only 30 are present on the Illumina 450k array." I do not see discussion of how authors account for this in their 450K analyses to make a fair comparison (though it is possible I missed this), and I am concerned much of the relevant RepliTali signal on the EPIC array may be simply ignored by the authors' analysis. Also, although the authors cite Zhou for HypoClock, HypoClock was not developed by Zhou – they use WGBS data there to investigate solo-WCGWs. Instead, the 450K-adapted HypoClock seems to have been developed by the authors themselves in their 2020 Genome Medicine paper. As Endicott et al discuss in their paper, solo-WCGWs are highly underrepresented on the 450K array (1.5% on the array vs. 11% in the genome), and those selected on the 450K array are likely to be unusual for a solo-WCGW (selected because they overlap an enhancer, etc). In the 2020 Genome Medicine paper, the authors dramatically restrict this already-limited subset of solo-WCGWs based on their own criteria. Thus, any hypomethylation mitotic signal remaining in HypoClock is inherently limited, and this does not seem to be a fair comparison to the hypermethylation counters. This 450K limitation applies to the TCGA Vogelstein-Tomasetti stem-cell division rate analysis (Fig 2h), TCGA age correlations (Fig 2B), TCGA cell-of-origin analysis (Fig S10), and sorted cell type age correlations (Fig 2d). Of note, in Fig S16 (precancerous lesions) – RepliTali performs as well as stemTOC, even though RepliTali suffers from the fact that multiple datasets (at least the ones shown in FigS16A) are 450K, raising the possibility RepliTali would perform even better using EPIC data. In the CTH analysis that includes EPIC data (Fig S7), RepliTali and EpiCMIT_hypo do not appear confounded by CTH and so are not outperformed by hypermethylation counters. The remaining EPIC analysis that clearly shows a higher metric for hypermethylation counters is the age correlation in the eGTEx dataset, but it is unclear to me what we expect to be the "correct" correlation of mitotic age with chronological age in various tissues (while the authors interpret that RepliTali/EpiCMIT's correlations are too low, it is also possible that stemTOC's correlations are too high). This 450K/EPIC issue thus appears to affect many of the authors' main conclusions about hyper- and hypo-methylation counters (which the authors state is a main advance of the paper). I invite the authors here to correct any misperceptions I have about their analysis or the underlying 450K/EPIC issue, or point me to an explanation of their RepliTali calculation that I missed.

Second, I remain as puzzled as Reviewer 1 about the sensitivity of MS1 vs. stemTOC, and how the saturation effect plays into this. Yes, stemTOC saturates in highly proliferative cancer cells, but that could simply indicate a smaller dynamic range for stemTOC due to a ceiling effect. The extrapolation to stemTOC vs. MS1 in normal-tissue data is not convincing in my view. I do find more compelling the authors' observation that MS1 does not correlate nearly as well with tumor cell of origin as stemTOC does, though this does not apply to precancerous tissues.

These two issues impact central claims made by the authors in the abstract. Thus these issues must be addressed and/or these statements in the abstract need to be dramatically toned down.

Finally, a perhaps naïve question: why do we need mitotic counters that only use DNAm gain or only use DNAm loss? Wouldn't it be more informative to ask what signals are shared between the two types and what signals are unique to one type, and then use that insight to build mitotic counters to leverage both?

Below are my notes on specific comments that I believe the authors have adequately addressed.
Reviewer 1

(Reviewer 1's comments are numbered based on the original review).

Major Points Comments 1-4, 7, 11, 15: Clearly has addressed reviewer comments through a mix of new data and clarification, mostly technical issues/questions

Major Points Comments 5, 6, 8-10: The general issue seems to be whether selecting CpGs correlated with age is reasonable, and whether mitotic clocks should generally increase with chronological age. I think the authors' strategy of CpG selection is reasonable – essentially they are assuming that CpGs that change with mitotic divisions should be a subset of those that change with age (but this is not the only criterion). It is also reasonable to assume that mitotic clocks should generally increase with chronological age especially in more proliferative tissues. The fact that chronological age is not the *only* factor that would influence mitotic age does not make this assumption unreasonable.

Major Point Comment 13: As far as I can tell, all issues are cleared up with clarification that the authors are referring to *tumor* cell of origin, as well as noting that the 2022 Nature Methods paper already performed the analyses the reviewer asked for.

Reviewer 3

(I am addressing Reviewer 3's remaining concern)

A true mitotic clock should be increased in a bulk tumor sample with more tumor cells. Yes, the correlation is "expected"- but that is the point. stemTOC is behaving as expected. Isn't that a necessary step in the validation of stemTOC? I also agree with the authors that use of EpiSCORE is reasonable, to ensure it is the expected cell type driving the correlation (the cell type that gave rise to the tumor, which is now overrepresented because the tumor cells are highly related to the original cell type). I agree with the authors here.

Reviewer #4 (Remarks to the Author):

Comment: I have been asked specifically to comment on whether the authors have adequately addressed reviewer comments (specifically Reviewer 1 and 3). I thus assessed: 1) did the authors directly address comments with new data, 2) if not, is it reasonable to simply clarify the methods or reasoning to address the comment, and 3) for points where there remains reasonable disagreement or ambiguity, does the uncertainty compromise the novelty or main point of the paper. Given the large number of reviewer comments, the answer varies based on which reviewer comment we are talking about. I find the authors' arguments to be compelling concerning two major issues: correlation with chronological age (for both underlying CpGs, and for the overall mitotic clock), and correlation with tumor cell of origin. The same is true for some smaller technical issues. The new figures provided in the Supplement are sufficient, and I do not believe the authors need to provide any further justification or data on these issues. Even where there could be some reasonable disagreement, I do not think they compromise any central claim of the paper.

Response: We sincerely thank the reviewer for taking time to evaluate our MS and for the professional assessment and positive feedback.

Comment: However, I find two of Reviewer 1's comments to not be adequately addressed, and these are on critical points. First, I am concerned about the broad statements concerning the superiority of hypermethylation vs. hypomethylation counters and fear they are overstated. While I am hesitant to raise an issue that has not yet been raised in review, it is generally related to Reviewer 1's point so it seems pertinent. Some of the key datasets used to demonstrate superiority of hypermethylation – e.g. TCGA, BLUEPRINT- are based on 450K data. As the authors state in Methods, "Of the 87 RepliTali CpGs, only 30 are present on the Illumina 450k array." I do not see discussion of how authors account for this in their 450K analyses to make a fair comparison (though it is possible I missed this), and I am concerned much of the relevant RepliTali signal on the EPIC array may be simply ignored by the authors' analysis. Also, although the authors cite Zhou for HypoClock, HypoClock was not developed by Zhou – they use WGBS data there to investigate solo-WCGWs. Instead, the 450K-adapted HypoClock seems to have been developed by the authors themselves in their 2020 Genome Medicine paper. As Endicott et al discuss in their paper, solo-WCGWs are highly underrepresented on the 450K array (1.5% on the array vs. 11% in the genome), and those selected on the 450K array are likely to be unusual for a solo-WCGW (selected because they overlap an enhancer, etc). In the 2020 Genome Medicine paper, the authors dramatically restrict this already-limited subset of solo-WCGWs based on their own criteria. Thus, any hypomethylation mitotic signal remaining in HypoClock is inherently limited, and this does not seem to be a fair comparison to the hypermethylation counters. This 450K limitation applies to the TCGA Vogelstein-Tomasetti stem-cell division rate analysis (Fig 2h), TCGA age correlations (Fig 2B), TCGA cell-of-origin analysis (Fig S10), and sorted cell type age correlations (Fig 2d). Of note, in Fig S16 (precancerous lesions) – RepliTali performs as well as stemTOC, even though RepliTali suffers from the fact that multiple datasets (at least the ones shown in FigS16A) are 450K, raising the possibility RepliTali would perform even better using EPIC data. In the CTH analysis that includes EPIC data (Fig S7), RepliTali and EpiCMIT_hypo do not appear confounded by CTH and so are not outperformed by hypermethylation counters. The remaining EPIC analysis that clearly shows a higher metric for hypermethylation counters is the age correlation in the eGTEX dataset, but it is unclear to me what we expect to be the "correct" correlation of mitotic age with chronological age is in various tissues (while the authors interpret

that RepliTali/EpiCMIT's correlations are too low, it is also possible that stemTOC's correlations are too high). This 450K/EPIC issue thus appears to affect many of the authors' main conclusions about hyper- and hypo-methylation counters (which the authors state is a main advance of the paper). I invite the authors here to correct any misperceptions I have about their analysis or the underlying 450K/EPIC issue, or point me to an explanation of their RepliTali calculation that I missed.

Response: The reviewer has raised a number of excellent points. We completely agree that the technology used to build a clock is a very important consideration, specially in so far as to how similar it is to the technologies of the validation and benchmarking datasets. In order to answer the reviewer's question it will be important to consider in turn each of the three "hypomethylation-based" clocks ("HypoClock", "RepliTali" and "epiCMIT-hypo"), as they were all derived from different technologies: the HypoClock sites were derived from WGBS, RepliTali from EPIC and epiCMIT-hypo from 450k. It is also very important to immediately point out that epiCMIT-hyper was built from exactly the same data (hence also same 450k technology) as epiCMIT-hypo. Hence, as far as the comparison of epiCMIT-hyper to epiCMIT-hypo is concerned, it should be clear that this comparison is fair. And in relation to this particular comparison our data is unequivocal in demonstrating that epiCMIT-hyper is a much more reliable mitotic clock than epiCMIT-hypo. This is not necessarily because of any inherent limitation of CpGs that lose DNAm being less faithful markers of cell-division, but more likely because of confounding by cell-type heterogeneity. The comparison of epiTOC1/2+stemTOC (all 450k based) to epiCMIT-hypo is also a fair comparison, and our data clearly shows that effectively all hypermethylated clocks are an improvement over epiCMIT-hypo (see e.g. Fig.2b,2d,2h, Fig.3d-e, SI fig.S5, S6 and S11).

As far as the comparison to "HypoClock" is concerned, we agree that the restriction to 450k/EPIC data is a limitation since the solo-CpGs were derived from WGBS data, so that the HypoClock assessed on array-data only contains a relatively small fraction (although still a very large number >6,000 CpGs) of the original solo-CpGs. However, we note that in our previous publication (Genome Med 2020) we analyzed WGBS data too, demonstrating that solo-CpGs were more likely to be immune cell-type specific than randomly selected CpGs. For the reviewer's convenience, we display below the relevant figure from that paper:

This figure shows that for any comparison of immune cell subtypes, the probability of a solo-CpG (red curve) displaying a bigger difference in average DNAm between the two cell subtypes (x-axis) is greater than for a randomly selected CpG (grey curve). We stress that the analysis above was done for all solo-CpGs (several million) and not just for the 6214 that mapped to 450k arrays. Thus, we are inclined to think that it is not just the 6214 solo-CpGs that map to 450k arrays that are enriched for cell-type specific enhancer regions, but that there is a larger set of solo-CpGs that are also cell-type specific. However, we also opine that there will be a fairly large subset of solo-CpGs that are not cell-type specific and indeed we think that in this regard RepliTali is a significant improvement over HypoClock (or PMDsoloCpG-clock if we use all PMD solo-CpGs) in honing in on this specific subset. Indeed, the data presented in Endicott et al, as well as the data presented in this MS clearly suggest that RepliTali (EPIC-based) is an improvement over the previous clocks built from solo-CpGs. This improvement in itself does not mean that the RepliTali solo-CpGs are optimized at measuring mitotic age since there could be other solo-CpG subsets not represented on the EPIC array that are better. Thus, likewise, our observation and statements that hypermethylated based clocks outperform HypoClock is merely a statement about “clocks”, and not a statement about whether the optimal subset only includes hypermethylated sites. We have now clarified this important point in Discussion.

As far as the comparison of hypermethylated-based clocks to RepliTali is concerned, we agree that comparing RepliTali (EPIC-based) to 450k-based hypermethylated clocks on 450k data may unfairly favour the 450k-clocks. Conversely, a comparison on EPIC data may slightly favour RepliTali, as not all 450k probes are on the EPIC array. Based on the EPIC data analyzed in this work (i.e. eGTEX and some of the whole blood datasets where we assessed correlations of mitotic age with chronological age), stemTOC does seem to attain stronger correlations than RepliTali. Whilst we fully agree with the reviewer that fully maximizing a correlation between mitotic and chronological age is not a reasonable evaluation metric (as indeed, a maximum correlation of 1 is clearly not optimal since biologically mitotic age and chronological age are distinct), nevertheless there is some evidence that the better performance of stemTOC over RepliTali on EPIC data is biologically genuine: for instance, let us take a high turnover-rate tissue like colon or blood, where there should be a significant correlation between mitotic and chronological age, and consider first the eGTEX EPIC colon dataset. If we don't adjust for CTH, RepliTali's correlation between mitotic and chronological age is not significant, but it becomes significant after adjustment for CTH. This was shown in SuppFig.S8 which for convenience we re-display again further below, now adding a new panel-b) to facilitate the direct comparison between stemTOC and RepliTali:

new SI fig.S8: Associations of mitotic age with chronological age in the solid normal tissue EPIC datasets from eGTEX: a) For each mitotic counter, scatterplots of the linear regression t -statistics of association between mitotic age and chronological age, before (NotAdjCTF) and after (AdjCTF) adjustment for cell-type fractions. Each datapoint represent an eGTEX normal tissue EPIC dataset, for which the underlying cell-type fractions in the tissue could be estimated using our EpiSCORE DNAm-atlas algorithm. The red dashed-lines indicate the threshold of statistical significance ($P=0.05$). **b)** Boxplots compare the same regression t -statistics for two clocks (stemTOC and RepliTali), not adjusting for CTH and adjusting for CTH. P -values are from a one-tailed paired t -test.

In contrast, for stemTOC, the correlation is significant and not dependent on whether we adjust for CTH.

The same is true for the EPIC whole blood datasets (shown in SuppFig.S7), which for the reviewer's convenience we redisplay again below, now stratified by EPIC and 450k technologies:

SI fig.S7(EPIC datasets only): Associations of mitotic age with chronological age in blood are not confounded by CTH. For each mitotic clock, boxplots compare the linear regression t -statistics of association between mitotic age and chronological age, not adjusting for immune cell-type fractions (NotAdjForCTF) and adjusting for 12 immune-cell type fractions (AdjForCTF). Each datapoint corresponds to one whole blood cohort, and there are a total of 5 whole blood EPIC cohorts, as indicated. P -values derive from two-tailed paired Wilcoxon rank sum tests comparing the t -statistics before and after adjustment for CTFs.

We can see that stemTOC and all other hypermethylated clocks attain stronger associations (larger regression t -statistics between mitotic and chronological age) than RepliTali, and importantly that for one WB dataset (Barturen), RepliTali does not display a significant correlation, although it becomes significant once we adjust for CTH. For completeness, we also display below the corresponding figure restricting to 450k datasets, which shows that now there are multiple cohorts where there is no significant correlation for RepliTali, even after adjustment for CTH, probably because of the lower number of RepliTali probes on the 450k arrays.

SI fig.S7 (450k datasets only): Associations of mitotic age with chronological age in blood are not confounded by CTH. For each mitotic clock, boxplots compare the linear regression t -statistics of

association between mitotic age and chronological age, not adjusting for immune cell-type fractions (NotAdjForCTF) and adjusting for 12 immune-cell type fractions (AdjForCTF). Each datapoint corresponds to one whole blood cohort, and there are a total of 13 whole blood 450k cohorts, as indicated. P-values derive from two-tailed paired Wilcoxon rank sum tests comparing the t-statistics before and after adjustment for CTFs.

To make all of this clearer in relation to the direct comparison of stemTOC to RepliTali, we re-display all of the above data in a different format:

The P-values are from a one-tailed paired t-test comparing stemTOC to RepliTali, and we can observe how (i) stemTOC does indeed display stronger correlations between mitotic and chronological age, (ii) how for a few cohorts, the RepliTali estimates don't pass statistical significance, and (iii) that adjustment for CTH improves the correlations displayed by RepliTali in particular. Hence, we suspect that some of the RepliTali CpGs are potentially still affected by CTH issues. Indeed, we note that in building stemTOC we do restrict to CpGs that are all in the same methylation state within a suitably defined ground-state whereas the use of multiple cell-lines to build RepliTali does not ensure that underlying CpGs are not cell-type specific. Thus, given that on EPIC data we do see an improvement of stemTOC over RepliTali, we are inclined to think that the other improvements seen on 450k datasets (e.g. TCGA) are not the result of an intrinsic (technological) bias.

In summary, although we agree with the reviewer that the data is not conclusive of stemTOC outperforming RepliTali, the data in this MS is supportive of the hypermethylated clocks+RepliTali outperforming epiCMIT-hypo and HypoClock. And the results on EPIC data shown above is suggestive of an improvement of stemTOC over RepliTali. So, overall, and given that there is relatively little CpG overlap between the 4 hypermethylated clocks, it does seem to point towards hypermethylated clocks being more reliable mitotic age estimators. But we also agree that there could be subsets of solo-CpGs that may yield even better mitotic clocks, and these could be developed in future if sufficiently large and relevant WGBS datasets are generated. In response to the reviewer's points we have now significantly revised the Abstract and Discussion, toning down any statements about hypermethylated sites outperforming hypomethylated ones, and adding the above new panels to the Supplementary Figures.

Comment: Second, I remain as puzzled as Reviewer 1 about the sensitivity of MS1 vs. stemTOC, and how the saturation effect plays into this. Yes, stemTOC saturates in highly proliferative cancer cells, but that could simply indicate a smaller dynamic range for stemTOC due to a ceiling effect. The extrapolation to stemTOC vs. MS1 in normal-tissue data is not convincing in my view. I do find more compelling the authors' observation that MS1 does not correlate nearly as well with tumor cell of origin as stemTOC does, though this does not apply to precancerous tissues.

Response: We thank the reviewer for raising this good point. We agree that the saturation effect depicted in Fig.5e (displayed again for convenience in panel-a) below) could merely reflect a "ceiling effect" (i.e. that DNAm values are bounded by 1). In particular, the reviewer is correct in pointing out that the saturation effect could be a technical artefact from taking the 95% upper-quantile in the definition of stemTOC. To explore this, we have studied what happens if we take an average DNAm over the stemTOC CpGs (instead of an upper 95% quantile), and the result is displayed in panel-b) below:

As we can see from the lower left panel, there is no longer a ceiling effect, because the average DNAm values are all < 0.8. However, the plot on the lower right (which collapses the data for each tissue-type using the median), still displays a saturation-profile, as evident from a non-linear least squares fit, which shows that the best fit is an exponential. Thus, it seems that the saturation effect in panel-a) above (or in Fig.5f) is not a technical artefact of our way of defining stemTOC as an upper quantile.

As far as the mutational loads are concerned, we agree with the reviewer that because these loads are defined as averages, that this may be the reason why we don't observe a saturation effect in tumors (Fig.5c). However, as shown by our analysis above, taking an average over the 371 stemTOC CpGs still suggests that there is a saturation effect (without a ceiling effect). It is also worth noting that we are not aware of any somatic mutational studies analogous to the DNAm-studies we performed back in 2012 & 2016 (Teschendorff et al Genome Med 2012 & Nat Commun 2016) that demonstrate explicit discrimination of normal healthy tissues from normal-tissues "at-cancer-risk". The greater difficulty of confidently detecting somatic mutations in normal-tissue is one underlying reason for this, and yet, this in itself already points towards DNAm changes in

normal-tissues being more easily detectable and providing a more sensitive marker of molecular changes that could be relevant for cancer risk prediction. Moreover, our understanding is that the rate of DNAm change per cell division in the human genome is greater than that of somatic mutations, although recent preprints indicate that this may not be the case in relation to cell-division.

In summary, we agree with the reviewer that we have not conclusively demonstrated any increased sensitivity of stemTOC over mutational loads in the context of normal tissues, but point out that it is highly plausible given the evidence presented here and the previous literature. In response to this, we have now toned down the abstract, the relevant subsection in Results and also clarified these points in Discussion. We have also added the above figure as a new SuppFig.S21.

Comment: These two issues impact central claims made by the authors in the abstract. Thus these issues must be addressed and/or these statements in the abstract need to be dramatically toned down.

Response: In response to this, we have now toned down our statements in the abstract.

Comment: Finally, a perhaps naïve question: why do we need mitotic counters that only use DNAm gain or only use DNAm loss? Wouldn't it be more informative to ask what signals are shared between the two types and what signals are unique to one type, and then use that insight to build mitotic counters to leverage both?

Response: We thank the reviewer for asking this good question. We think that it could turn out to be extremely important to distinguish CpGs that gain DNAm from those that lose DNAm in the context of cell-division. First of all, our understanding is that the loss of DNAm associated with cell-division occurs primarily in late-replicating PMDs, and there is a clear mechanistic interpretation for this incomplete maintenance, as for instance nicely explained by Zhou W et al Nat Genet 2018. Clearly, the mechanism(s) that lead to aberrant hypermethylation associated with cell-division are very distinct. Hence, in our opinion, it is important to establish which of these different epimutation mechanisms seems to associate better with mitotic age, and which patterns of DNAm change display stronger associations with cancer-risk. Indeed, the underlying mechanisms that lead to hypermethylation and hypomethylation in the context of cell-division may operate in a context dependent manner. For instance, based on all the data we have accumulated loss of DNAm in late-replicating PMDs seems to be more prominent when cells are under high replicative stress (cancer, early development, high tissue-turnover) and that would make a lot of biological sense. Indeed, the solo-CpGs within late-replicating PMDs were to our understanding identified by comparing highly-replicative cancer-tissue to normal tissue, so this would select for CpGs whose DNAm changes in the context of high-replicative stress. In contrast, the aberrant gains in DNAm do not seem to depend on the degree of replicative stress. We therefore think that understanding these differences is an extremely important question for the cancer-risk, early detection and even cancer-prevention field, a question that we can't answer as well if we consider clocks that merge these distinct mechanisms together.

Reviewers' Comments:

Reviewer #4:

Remarks to the Author:

I appreciate the authors' response, in particular toning down the statements in the abstract. After reviewing the literature again, I do believe the paper's novelty warrants publication in Nature Communications. The extensive benchmarking of multiple mitotic counters for multiple properties that would be expected for a mitotic clock is noteworthy, as is stemTOC's multiple innovations upon prior iterations. There are two remaining minor issues:

First, I appreciate the extensive discussion of the 450K/EPIC issue in both the results and discussion section. However, the burden is currently on the reader to keep track of what datasets are 450K and which ones are EPIC, which is difficult given the large number of datasets. Given that moving forward all data will be collected on EPIC (in fact, EPIC v2), it is important that it is crystal clear to readers which comparisons were done using which arrays when looking at the figures alone. For clarity, the authors should re-label RepliTali in all figures where 450K data is used as RepliTali-450K. This labeling should also be used in the text.

Related to the above point: using EPIC arrays, what is the correlation between the full RepliTali metric and RepliTali when limiting to just the 450K/EPIC overlap? This correlation should be reported, ideally for multiple datasets.

Second, review of the literature shows that the link between mitotic age and the Tomasetti-Vogelstein hypothesis was previously shown by PMID 37467337. This should be cited in the introduction and discussion.

Reviewer #4 (Remarks to the Author):

Comment: I appreciate the authors' response, in particular toning down the statements in the abstract. After reviewing the literature again, I do believe the paper's novelty warrants publication in Nature Communications. The extensive benchmarking of multiple mitotic counters for multiple properties that would be expected for a mitotic clock is noteworthy, as is stemTOC's multiple innovations upon prior iterations. There are two remaining minor issues.

Response: We sincerely thank the reviewer for taking time to re-evaluate our MS and for the positive feedback.

Comment: First, I appreciate the extensive discussion of the 450K/EPIC issue in both the results and discussion section. However, the burden is currently on the reader to keep track of what datasets are 450K and which ones are EPIC, which is difficult given the large number of datasets. Given that moving forward all data will be collected on EPIC (in fact, EPIC v2), it is important that it is crystal clear to readers which comparisons were done using which arrays when looking at the figures alone. For clarity, the authors should re-label RepliTali in all figures where 450K data is used as RepliTali-450K. This labeling should also be used in the text.

Response: We thank the reviewer for appreciating the extensive work done. As the reviewer points out we discuss this issue at length within Results and Discussion, and so we feel that this is sufficient to alert the reader to this point. Changing names now in figures would in our opinion only overcomplicate matters. For instance, this would not only apply to RepliTali, but when applying 450k-trained clocks to EPIC data we would also need to alert the reader to this fact. This is not in the interest of the general readership. As a compromise, we now state in the figure legends if RepliTali was restricted to 450k probes.

Comment: Related to the above point: using EPIC arrays, what is the correlation between the full RepliTali metric and RepliTali when limiting to just the 450K/EPIC overlap? This correlation should be reported, ideally for multiple datasets.

Response: Below is the result for the various normal tissues of the eGTEX dataset (all EPIC data), and as the reviewer can see correlations are strong, as expected.

We have decided however not to include this in the MS, because it does not really add much value and we already have a very long MS.

Comment: Second, review of the literature shows that the link between mitotic age and the Tomasetti-Vogelstein hypothesis was previously shown by PMID 37467337. This should be cited in the introduction and discussion.

Response: The first manuscript to demonstrate a correlation between mitotic age and the Tomasetti-Vogelstein estimates was a 2016 publication by our own group (epiTOC, Yang Z et al Genome Biol 2016). We also re-analyzed this in our 2020 publication (epiTOC2, Teschendorff et al Genome Med 2020). The paper the reviewer is referring to is a 2023 publication, which we presume performed the comparison motivated by these earlier papers. Nevertheless, we have now cited the above paper in the Introduction to our manuscript.